# For high-dimensional hierarchical models, consider exchangeability of effects across covariates instead of across datasets

**Brian L. Trippe**
MIT CSAIL
btrippe@mit.edu

**Hilary K. Finucane**
Broad Institute
finucane@broadinstitute.org

**Tamara Broderick**
MIT CSAIL
tbroderick@csail.mit.edu

## Abstract

Hierarchical Bayesian methods enable information sharing across regression problems on multiple groups of data. While standard practice is to model regression parameters (effects) as (1) exchangeable across the groups and (2) correlated to differing degrees across covariates, we show that this approach exhibits poor statistical performance when the number of covariates exceeds the number of groups. For instance, in statistical genetics, we might regress dozens of traits (defining groups) for thousands of individuals (responses) on up to millions of genetic variants (covariates). When an analyst has more covariates than groups, we argue that it is often preferable to instead model effects as (1) exchangeable across *covariates* and (2) correlated to differing degrees across *groups*. To this end, we propose a hierarchical model expressing our alternative perspective. We devise an empirical Bayes estimator for learning the degree of correlation between groups. We develop theory that demonstrates that our method outperforms the classic approach when the number of covariates dominates the number of groups, and corroborate this result empirically on several high-dimensional multiple regression and classification problems.

## 1 Introduction

Hierarchical modeling is a mainstay of Bayesian inference. For instance, in (generalized) linear models, the unknown parameters are *effects*, each of which describes the association of a particular covariate with a response of interest. Often covariates are shared across multiple related groups, but the effects are typically allowed to vary both by group and by covariate. A classic methodology, dating back to Lindley and Smith (1972) [44], models the effects as conditionally independent across groups, with a latent (and learnable) degree of relatedness across covariates. From a practical standpoint, the model is motivated by the understanding that it "borrows strength" across different groups [24, Chapter 5.6]. Mathematically, the model is motivated by assuming effects are exchangeable across groups and applying a de Finetti theorem [44, 35]. The methodology of Lindley and Smith is ubiquitous when the number of groups is larger than the number of covariates. It is a standard component of Bayesian pedagogy [[23, Chapter 13.3]; [24, Chapter 15.4]] and software; e.g. it is used in the mixed modeling package `lme4` [5], which has over 16 million downloads at the time of writing.

35th Conference on Neural Information Processing Systems (NeurIPS 2021).

Despite its resounding success when there are more groups than covariates, we show in the present work that this standard methodology performs poorly when there are more covariates than groups. To address the many-covariates case, we turn for inspiration to statistical genetics, where scientists commonly learn linear models relating genetic variants (covariates) to traits (corresponding to different groups) across individuals (which each exhibit a response). These applications may exhibit millions of covariates, thousands of responses, and just a handful of groups. In these cases, [39, 12, 56, 69, 46, 53] use a multivariate Gaussian prior akin to that of Lindley and Smith, but assume conditional independence across *covariates* and prior parameters that encode correlations across *groups*, rather than than the other way around.

As we will see, this alternative modeling approach may be motivated from a Bayesian perspective when one begins from an assumption of a priori exchangeability of the effects across covariates (rather than across groups). This exchangeability assumption is reasonable in statistical genetics, where we have little knowledge to distinguish our expectations about the effects of different genetic variants; we argue this modeling approach can be effective other modern high-dimensional analyses of multiple groups of data (beyond statistical genetics) in which large collections of covariates are frequently treated monolithically, e.g. by applying ridge regression. Namely, when there are more covariates than groups, we propose to model the effects as exchangeable across *covariates* (rather than groups) and learn the degree of relatedness of effects across *groups* (rather than covariates). In what follows, we refer to this framework as *ECov*, for *e*xchangeable effects across *cov*ariates, and distinguish it from *e*xchangeable effects across *groups* or *EGroup*.

While the existing methods in statistical genetics for modeling multiple traits obtain as a special case of ECov, to the best of our knowledge this approach is absent from existing literature on hierarchical Bayesian regression. Brown and Zidek (1980) [10] and Haitovsky (1987) [28] form two exceptions, but these two papers (1) consider only the situation in which a single covariate matrix is shared across all groups (or equivalently, for each data point all responses are observed) and (2) include only theory and no empirics. While Lindley and Smith (and others) discuss a priori exchangeability across covariates in the context of analysis of a single group, to our knowledge no other work has pushed this idea forward to share strength across multiple groups.

We suspect that the historical origins of the methodology in statistical genetics may have hindered earlier expansion of this class of models to a wider audience. In particular, this literature traces back to mixed effects modeling for cattle breeding [57]; here, an even-earlier notion of the genetic contribution of trait correlation (i.e. "genetic correlation;" see Hazel (1943) [29]) informs the covariance structure of random effects. Although genetic correlation is now commonly understood to describe the correlation of effects of DNA sequence changes on different traits [12], its provenance predates even the first identification of DNA as the genetic material in 1944 [3]. As such, this older motivation obviated the need for a more general justification grounded in exchangeability. See Appendix A for further discussion of related work, including more recent works from within the machine learning community on sharing strength across multiple groups of data.

In the present work, we propose ECov as a general framework for hierarchical regression when the number of covariates exceeds the number of groups. We show that the classic model structure from statistical genetics can be seen as an instance of this framework, much as Lindley and Smith give a (complementary) instance of an EGroup framework. To make the ECov approach generally practical, we devise an accurate and efficient algorithm for learning the matrix of correlations between groups. We demonstrate with theory and empirics that ECov is preferred when the number of covariates exceeds the number of groups, while EGroup is preferred when the number of groups exceeds the number of covariates. Our experiments analyze three real, non-genetic groups in regression and classification, including an application to transfer learning with pre-trained neural network embeddings. We provide proofs of theoretical results in the appendix.

## 2   Exchangeability and its applications to hierarchical linear modeling

We start by establishing the data and model, motivating exchangeability among covariate effects (ECov), and motivating our Bayesian generative model.

**Setup and notation.** Consider $Q$ groups with $D$ covariates. Let $N^q$ be the number of data points in group $q$. For the $q$th group, the $N^q \times D$ real design matrix $X^q$ collects the covariates, and $Y^q$ is the $N^q$-vector of responses. The $n$th datapoint in group $q$ consists of covariate $D$-vector $X_n^q$ and

scalar response $Y_n^q$. We let $\mathcal{D} := \{(X^q, Y^q)\}_{q=1}^Q$ denote the collection of data from all $Q$ groups.

We consider the generalized linear model $Y_n^q | X_n^q, \beta^q \overset{indep}{\sim} p(\cdot | X_n^{q\top} \beta^q)$ with unknown $D$-vector of real effects $\beta^q$. We collect all effects in a $D \times Q$ matrix $\beta$ with $(d, q)$ entry $\beta_d^q$. The linear form of the likelihood allows interpretation of $\beta_d^q$ as the association between the $d$th covariate and the response in group $q$. In linear regression, the responses are real-valued and the conditional distribution is Gaussian. In logistic regression, the responses are binary, and we use the logit link. The independence assumption conflicts with some models that one might use, for example in some cases when the different groups partially overlap.

**Example.** As a motivating non-genetics example, consider a study of the efficacy of microcredit. There are seven famous randomized controlled trials of microcredit, each in a different country [48]. We might be interested in the association between various aspects of small businesses (covariates), including whether or not they received microcredit, and their business profit (response). In this case, the $d$th element of $X_n^q$ would be the $d$th characteristic of the $n$th small business in the $q$th country, and $Y_n^q$ is the profit of this business. See the experiments for additional examples in rates of policing, web analytics, and transfer learning.

**Exchangeable effects across groups (EGroup).** To fully specify a Bayesian model, we need to choose a prior over the parameters $\beta$. Lindley and Smith assume the effects are exchangeable across groups. Namely, for every $Q$-permutation $\sigma$, $p(\beta^1, \beta^2, \ldots, \beta^Q) = p(\beta^{\sigma(1)}, \beta^{\sigma(2)}, \ldots, \beta^{\sigma(Q)})$. Assuming exchangeability holds for an imagined growing $Q$ and applying de Finetti's theorem motivates a conditionally independent prior. Concretely, Lindley and Smith take $\beta^q \overset{i.i.d.}{\sim} \mathcal{N}(\xi, \Gamma)$, for $D$-vector $\xi$ and $D \times D$ covariance matrix $\Gamma$. The $(d, d')$ entry of $\Gamma$ captures the degree of relatedness between the effects for covariates $d$ and $d'$. Both $\xi$ and $\Gamma$ may be learned in an empirical Bayes procedure. However, when $D$ is large relative to $Q$, learning these parameters can present both computational and inferential challenges, as the $O(D^2)$ degrees of freedom in $\Gamma$ outnumber the $O(DQ)$ effects.

**Exchangeable effects across covariates (ECov).** We here argue for a complementary approach in settings where $D > Q$. In the microcredit example, notice that $D > Q$ will arise whenever the experimenter records more characteristics of a small business than there are locations with microcredit experiments; that is, $D > 7$ in this particular case. Concretely, let $\beta_d$ be the $Q$-vector of effects for covariate $d$ across groups. Then, in the ECov approach, we will assume that effects are exchangeable across covariates instead of across groups. Namely, for every $D$-permutation $\sigma$, $p(\beta_1, \beta_2, \ldots, \beta_D) = p(\beta_{\sigma(1)}, \beta_{\sigma(2)}, \ldots, \beta_{\sigma(D)})$. We will see theoretical and empirical benefits to ECov in later sections, but note that the ECov assumption is often consistent with prior beliefs in high dimensional settings. For instance, regarding microcredit, we may have no prior knowledge about how effects differ for distinct small-business characteristics. And we may a priori believe that different countries could exhibit more similar effects – and wish to learn the degree of relatedness across those countries.

We may apply a similar rationale as Lindley and Smith to motivate a conditionally independent model. Analogous to Lindley and Smith, we propose a Gaussian prior: $\beta_d \overset{i.i.d.}{\sim} \mathcal{N}(0, \Sigma)$. $\Sigma$ is now a $Q \times Q$ covariance matrix whose $(q, q')$ entry captures the similarity between the effects in the $q$ and $q'$ groups. For simplicity, we restrict to $\mathbb{E}[\beta_d] = 0$; see Appendix E.3 for discussion. Another potential benefit to ECov relative to EGroup is that we might expect a statistically easier problem, with $O(Q^2)$ rather than $O(D^2)$ values to learn in the relatedness matrix. We provide a rigorous theoretical analysis in Sections 4 and 5.

## 3 Our method

We next describe our inference method for specific instances of the exchangeable covariate effects model of Section 2. We compute the $\beta$ posterior and take an empirical Bayes approach to estimate $\Sigma$. We find that an expectation maximization (EM) algorithm estimates $\Sigma$ effectively; Appendix A.2 compares our approach to existing methods for the related problem of estimating $\Gamma$ for EGroup.

**Notation.** We identify estimates of $\beta$ and $\Sigma$ with hats. For instance, $\hat{\beta}_{LS}$ is the least squares estimate, with $\hat{\beta}_{LS}^q := (X^{q\top} X^q)^{-1} X^{q\top} Y^q$. We will sometimes find it useful to stack the columns of $\beta$ or its estimates into a length $DQ$ vector; we denote such vectors with an arrow; for example,

**Algorithm 1** Expectation Maximization for Exchangeability Among Covariate Effects

1: // Initialize covariance
2: $\Sigma^{(0)} \leftarrow I_Q$
3: // Run EM algorithm
4: **for** $t = 0, 1, \ldots$ **do**
5:     // Expectation step
6:     $\mu_1, \ldots, \mu_D, V_1, \ldots, V_D \leftarrow \texttt{E\_Step}(\Sigma^{(t)})$
7:
8:     // Maximization step
9:     $\Sigma^{(t+1)} \leftarrow D^{-1} \sum_{d=1}^{D} (\mu_d \mu_d^\top + V_d)$
10:
11: Return $\Sigma^{(t+1)}$

---

**Algorithm 2** E-Step: Linear Regression

1: $\vec{\mu}, V \leftarrow \mathbb{E}[\vec{\beta}|\mathcal{D}, \Sigma], \mathrm{Var}[\vec{\beta}|\mathcal{D}, \Sigma]$
2: **for** $d = 1, \ldots, D$ **do**
3:     $\mu_d \leftarrow (e_d \otimes I_Q)^\top \vec{\mu}$
4:     $V_d \leftarrow (e_d \otimes I_Q)^\top V (e_d \otimes I_Q)$
5: Return $\mu_1, \ldots, \mu_D, V_1, \ldots, V_D$

---

**Algorithm 3** E-Step: Logistic Regression

1: $\vec{\mu}^* \leftarrow \arg\max_{\vec{\beta}} \log p(\vec{\beta}|\mathcal{D}, \Sigma)$
2: $V \leftarrow -[\nabla_{\vec{\beta}}^2 \log p(\vec{\beta}|\mathcal{D}, \Sigma)\big|_{\vec{\beta}=\vec{\mu}^*}]^{-1}$
3: **for** $d = 1, \ldots, D$ **do**
4:     $\mu_d \leftarrow (e_d \otimes I_Q)^\top \vec{\mu}^*$
5:     $V_d \leftarrow (e_d \otimes I_Q)^\top V (e_d \otimes I_Q)$
6: Return $\mu_1, \ldots, \mu_D, V_1, \ldots, V_D$

$\vec{\beta} := [\beta^{1\top}, \beta^{2\top}, \ldots, \beta^{Q\top}]^\top$. For a natural number $N$, we use $I_N, \mathbf{1}_N$, and $e_N$ to denote the $N \times N$ identity matrix, $N$-vector of ones, and $N$th basis vector, respectively. We use $\otimes$ to denote the Kronecker product.

### 3.1 Posterior inference with a Gaussian likelihood

We first consider a Gaussian likelihood: for each group $q$ and observation $n$, we take $Y_n^q | X_n^q, \beta^q \overset{indep}{\sim} \mathcal{N}(X_n^{q\top}\beta^q, \sigma_q^2)$ where $\sigma_q^2$ is a group-specific variance. When the relatedness matrix $\Sigma$ is known, a natural estimate of $\beta$ is its posterior mean. We obtain the full posterior, including its mean, via a standard conjugacy argument; see Appendix B.1:

**Proposition 3.1.** *For each covariate $d$, let $\beta_d \overset{i.i.d.}{\sim} \mathcal{N}(0, \Sigma)$ a priori. For each group $q$ and data point $n$, let $Y_n^q | X_n^q, \beta^q \overset{indep}{\sim} \mathcal{N}(X_n^{q\top}\beta^q, \sigma_q^2)$. Then $\vec{\beta}|\mathcal{D}, \Sigma \sim \mathcal{N}(\vec{\mu}, V)$ for $\vec{\mu} = V[\sigma_1^{-2}Y^{1\top}X^1, \ldots, \sigma_Q^{-2}Y^{Q\top}X^Q]^\top$ and $V^{-1} = \Sigma^{-1} \otimes I_D + \mathrm{diag}(\sigma_1^{-2}X^{1\top}X^1, \ldots, \sigma_Q^{-2}X^{Q\top}X^Q)$, where $\mathrm{diag}(\sigma_1^{-2}X^{1\top}X^1, \ldots, \sigma_Q^{-2}X^{Q\top}X^Q)$ denotes a $DQ \times DQ$ block-diagonal matrix.*

At first glance, the posterior mean $\vec{\mu}$ for this model might seem to introduce a computational challenge because exact computation of $V$ involves an $O(D^3Q^3)$-time matrix inversion. Our experiments (Section 6), however, involve on the order of $DQ \approx 1,000$ parameters, so direct inversion of $V$ demands less than a single second. Moreover, in much larger problems $\vec{\mu}$ may still be computed very efficiently using the conjugate gradient algorithm [49, Chapter 5], with convergence in a small number of $O(D^2Q)$ time iterations; see Appendix B.2.

### 3.2 Empirical Bayes estimation of $\Sigma$ by expectation maximization

The posterior mean of $\beta$ in Proposition 3.1 requires $\Sigma$, which is typically unknown. Accordingly, we propose an empirical Bayes approach of estimating $\Sigma$ by maximum marginal likelihood:

$$\hat{\beta}_{\mathrm{ECov}} := \mathbb{E}[\beta \mid \mathcal{D}, \hat{\Sigma}] \text{ where } \hat{\Sigma} := \arg\max_{\Sigma \succeq 0} p(\mathcal{D} \mid \Sigma). \tag{1}$$

Equation (1) defines a two step procedure. In the first step, we learn the similarity between groups via estimation of $\Sigma$. In the second step, we use this similarity to compute an estimate, $\hat{\beta}_{\mathrm{ECov}}$, that correspondingly shares strength. Though we have been unable to identify a general analytic form for $\hat{\Sigma}$, we can compute it with an expectation maximization (EM) algorithm [47, Chapter 1.5]. Algorithm 1 summarizes this procedure; see Appendix B.3 for details.

## 3.3 Classification with logistic regression

We can extend the approach above to inference for multiple related classification problems. We assume a logistic likelihood; for each $q$ and $n$, $Y_n^q | X_n^q, \beta^q \overset{indep}{\sim} \text{Bern}[(1 + \exp\{-X_n^{q\top}\beta^q\})^{-1}]$. In the classification case, we cannot use Gaussian conjugacy directly, so we apply an approximation. Specifically, we adapt the original E-step in Algorithm 3 by using a Laplace approximation to the posterior [7, Chapter 4.4]. We approximate the posterior mean of $\beta$ by the maximum a posteriori value. We leave extensions to other generalized linear models to future work.

# 4 Theoretical comparison of frequentist risk

In this section, we prove theory that suggests ECov has better frequentist risk than EGroup when $D$ is large relative to $Q$. Analyzing $\hat{\beta}_{\text{ECov}}$ directly is challenging due to its non-differentiability as a function of the data, so we take a multipart approach. First, in Theorem 4.3, we show that an ECov estimate based on moment-matching (MM), $\hat{\beta}_{\text{ECov}}^{\text{MM}}$, dominates least squares, $\hat{\beta}_{\text{LS}}$, when $D$ is large relative to $Q$; $\hat{\beta}_{\text{LS}}$ in turn dominates $\hat{\beta}_{\text{EGroup}}^{\text{MM}}$ (a similar estimator for EGroup). Second, in Theorem 4.5, we show that $\hat{\beta}_{\text{ECov}}$ uniformly improves on $\hat{\beta}_{\text{ECov}}^{\text{MM}}$.

**Setup.** Take a fixed value of $\beta$ and an estimator $\hat{\beta}$. We use squared error risk, $\text{R}(\beta, \hat{\beta}) := \mathbb{E}\left[\|\hat{\beta} - \beta\|_F^2 \mid \beta\right]$, as our measure of performance. $\|\cdot\|_F$ is the Frobenius norm of a matrix, and the expectation is over all observations $Y^1, \ldots, Y^Q$ jointly. We require the following orthogonal design condition.

**Condition 4.1.** *For each group $q$, $\sigma_q^{-2} X^{q\top} X^q = \sigma^{-2} I_D$ for some shared variance $\sigma^2$.*

Though restrictive, this condition is useful for theory, as other authors have found; see Appendix C.1. We empirically demonstrate that our theoretical conclusions apply more broadly in Section 6.

**ECov vs. EGroup when using moment matching in high dimensions.** For ECov, the following estimate for $\Sigma$ is unbiased under correct prior specification: $\hat{\Sigma}^{\text{MM}} := D^{-1}\hat{\beta}_{\text{LS}}^{\top}\hat{\beta}_{\text{LS}} - D^{-1}\text{diag}(\sigma_1^2\|X^{1\dagger}\|_F^2, \ldots, \sigma_Q^2\|X^{Q\dagger}\|_F^2)$, where $\dagger$ denotes the Moore-Penrose pseudoinverse of a matrix and $\hat{\beta}_{\text{LS}}$ is the least squares estimate. We define $\hat{\beta}_{\text{ECov}}^{\text{MM}} := \mathbb{E}[\beta | \mathcal{D}, \hat{\Sigma}^{\text{MM}}]$ to be the resulting parameter estimate, and define $\hat{\beta}_{\text{EGroup}}^{\text{MM}}$ analogously for EGroup; see Appendix C.2 for details. While $\hat{\beta}_{\text{ECov}}^{\text{MM}}$ and $\hat{\beta}_{\text{EGroup}}^{\text{MM}}$ are naturally defined only when $D \geq Q$ and $D \leq Q$, respectively, we find it informative to compare how their performances depend on $D$ and $Q$ nonetheless.

Before our theorem, a lemma provides concise expressions for the risks of $\hat{\beta}_{\text{ECov}}^{\text{MM}}$ and $\hat{\beta}_{\text{EGroup}}^{\text{MM}}$.

**Lemma 4.2.** *Under Condition 4.1 and when $D \geq Q$, $\text{R}(\beta, \hat{\beta}_{\text{ECov}}^{\text{MM}}) = \sigma^2 DQ - \sigma^4 D(D - 2 - 2Q)\mathbb{E}[\|\hat{\beta}_{\text{LS}}^{\dagger}\|_F^2 \mid \beta]$. Additionally, when $D \leq Q$, $\text{R}(\beta, \hat{\beta}_{\text{EGroup}}^{\text{MM}}) = \sigma^2 DQ - \sigma^4 Q(Q - 2 - 2D)\mathbb{E}[\|\hat{\beta}_{\text{LS}}^{\dagger}\|_F^2 \mid \beta]$.*

Lemma 4.2 reveals forms for the risks of $\hat{\beta}_{\text{ECov}}^{\text{MM}}$ and $\hat{\beta}_{\text{EGroup}}^{\text{MM}}$ that are surprisingly simple. The symmetry between the forms and risks of these estimators, however, is intuitive; under Condition 4.1, $\hat{\beta}_{\text{ECov}}^{\text{MM}}$ and $\hat{\beta}_{\text{EGroup}}^{\text{MM}}$ can be seen as respectively arising from the same procedure applied to $\hat{\beta}_{\text{LS}}$ and its transpose.

With Lemma 4.2 in hand, we can now compare the risk of $\hat{\beta}_{\text{ECov}}^{\text{MM}}$, $\hat{\beta}_{\text{LS}}$, and $\hat{\beta}_{\text{EGroup}}^{\text{MM}}$.

**Theorem 4.3.** *Let Condition 4.1 hold. Then (1) if $D > 2Q + 2$, $\hat{\beta}_{\text{ECov}}^{\text{MM}}$ dominates $\hat{\beta}_{\text{LS}}$ with respect to squared error risk. In particular, for any $\beta$, $\text{R}(\beta, \hat{\beta}_{\text{ECov}}^{\text{MM}}) < \text{R}(\beta, \hat{\beta}_{\text{LS}})$. Additionally, (2) if $D > Q/2 - 1$, $\hat{\beta}_{\text{EGroup}}^{\text{MM}}$ is dominated by $\hat{\beta}_{\text{LS}}$.*

Since $\hat{\beta}_{\text{LS}}$ is minimax [41, Chapter 5], Theorem 4.3 implies that $\hat{\beta}_{\text{ECov}}^{\text{MM}}$ has minimax risk in the high-dimensional setting. It follows that, regardless of how well the ECov prior assumptions hold, $\hat{\beta}_{\text{ECov}}^{\text{MM}}$ will not perform very poorly.

**Further improvement with maximum marginal likelihood.** The moment based approach analyzed above has a limitation: with positive probability, $\hat{\Sigma}^{\mathrm{MM}}$ is not positive semi-definite (PSD). Though our expression for $\hat{\beta}_{\mathrm{ECov}}^{\mathrm{MM}}$ remains well-defined in this case, this non-positive definiteness obscures the interpretation of $\hat{\beta}_{\mathrm{ECov}}^{\mathrm{MM}}$ as a Bayes estimate. We next show that performance further improves if $\Sigma$ is instead estimated by maximum marginal likelihood (Equation (1)) and is thereby constrained to be PSD.

Our next lemma characterizes the form of the resulting estimator, $\hat{\beta}_{\mathrm{ECov}}$, and establishes a connection to the positive part James-Stein estimator [4].

**Lemma 4.4.** *Assume $D > Q$ and consider the singular value decomposition $\hat{\beta}_{\mathrm{LS}} = V \operatorname{diag}(\lambda^{\frac{1}{2}}) U^\top$ where $V$ and $U$ satisfy $V^\top V = U^\top U = I_Q$, and $\lambda$ is a $Q$-vector of non-negative reals. Under Condition 4.1, Equation (1) reduces to $\hat{\Sigma} = U \operatorname{diag}\left[(D^{-1}\lambda - \sigma^2 \mathbf{1}_Q)_+\right] U^\top$ and $\hat{\beta}_{\mathrm{ECov}} = V \operatorname{diag}\left[\lambda^{\frac{1}{2}} \odot (\mathbf{1}_Q - \sigma^2 D\lambda^{-1})_+\right] U^\top$, where $(\cdot)_+$ is shorthand for $\max(\cdot, 0)$ element-wise, $\odot$ is the Hadamard (i.e. element-wise) product, and the powers in $\lambda^{\frac{1}{2}}$ and $\lambda^{-1}$ are applied element-wise.*

Lemma 4.4 allows us to see $\hat{\beta}_{\mathrm{ECov}}$ as shrinking $\hat{\beta}_{\mathrm{LS}}$ toward 0 in the direction of each singular vector to an extent proportional to the inverse of the associated singular value. The transition from $\hat{\beta}_{\mathrm{ECov}}^{\mathrm{MM}}$ to $\hat{\beta}_{\mathrm{ECov}}$ is then analogous to the taking the "positive part" of the James-Stein estimator in vector estimation, which provides a uniform improvement in risk [4]. Though $\mathrm{R}(\beta, \hat{\beta}_{\mathrm{ECov}})$ is not easily available analytically, we nevertheless find that it dominates its moment-based counterpart.

**Theorem 4.5.** *Assume $D > Q + 1$. Under Condition 4.1 $\hat{\beta}_{\mathrm{ECov}}$ dominates $\hat{\beta}_{\mathrm{ECov}}^{\mathrm{MM}}$ with respect to squared error loss, achieving strictly lower risk for every value of $\beta$.*

We establish Theorem 4.5 using a proof technique adapted from Baranchik [4]; see also Lehmann and Casella [41][Thm. 5.5.4]. The standard approach we build upon is complicated by the fact that the directions in which we apply shrinkage are themselves random.

Theorem 4.5 provides a strong line of support for using $\hat{\beta}_{\mathrm{ECov}}$ over $\hat{\beta}_{\mathrm{ECov}}^{\mathrm{MM}}$ that does not rely on any assumption of "correct" prior specification; in particular the risk improves without any subjective assumptions on $\beta$. We discuss related earlier work in Appendix A.4.

## 5 Gains from ECov in the high-dimensional limit

The results of Section 4 give a promising endorsement of ECov but face two important limitations. First, the domination results relative to least squares do not directly demonstrate that $\hat{\beta}_{\mathrm{ECov}}$ attains improvements by leveraging similarities across groups in a meaningful way; indeed for a single group (i.e. $Q = 1$) $\hat{\beta}_{\mathrm{ECov}}$ can be understood as a ridge regression estimate [31], and Theorems 4.3 and 4.5 provide that $\hat{\beta}_{\mathrm{ECov}}$ dominates $\hat{\beta}_{\mathrm{LS}}$ for $D > 3$. Second, domination results reveal nothing about the size of the improvement or how it depends on any structure of $\beta$; intuitively, we should expect better performance when $\beta$ is in some way representative of the assumed prior. To address these limitations, we analyze the size of the gap between the risk of (1) $\hat{\beta}_{\mathrm{ECov}}$ and (2) our method applied to each group independently (ID), which we denote by $\hat{\beta}_{\mathrm{ID}}$.[1] We will characterize the dependence of this gap on $\beta$.

Reasoning quantitatively about the dependence of the risk on the unknown parameter poses significant analytical challenges. In particular, Lemma 4.2 shows that $\mathrm{R}(\beta, \hat{\beta}_{\mathrm{ECov}}^{\mathrm{MM}})$ depends on $\beta$ through $\mathbb{E}[\|\hat{\beta}_{\mathrm{LS}}^\dagger\|_F^2 | \beta]$; however, $\|\hat{\beta}_{\mathrm{LS}}^\dagger\|_F^2$ is the sum of the eigenvalues of a non-central inverse Wishart matrix, a notoriously challenging quantity to work with; see e.g. [42, 30]. To regain tractability, we (1) develop an analysis asymptotic in the number of covariates $D$ and (2) shift to a Bayesian analysis in order to sensibly consider a growing collection of covariate effects. In particular, we consider a sequence of regression problems, with parameters $\{\beta_d\}_{d=1}^\infty$ distributed as $\beta_d \overset{i.i.d.}{\sim} \pi$ for some distribution $\pi$. Accordingly, instead of using the frequentist risk as in Section 4, we now use the Bayes risk to measure performance. Specifically, for a group with $D$ covariates and an estimator $\hat{\beta}$,

---

[1] Our approach $\hat{\beta}_{\mathrm{ECov}}$ is well defined in the $Q = 1$ single group case; for each group $q$, we obtain $\hat{\beta}_{\mathrm{ID}}^q$ by computing $\hat{\beta}_{\mathrm{ECov}}$ on the group $\mathcal{D} = \{(X^q, Y^q)\}$.

the Bayes risk is $R_\pi^D(\hat\beta) := \mathbb{E}_\pi[R(\beta, \hat\beta)]$ where $R(\beta, \hat\beta)$ is the usual frequentist risk. In the following, we describe the results of this analysis with proofs and additional details left to Appendix D.

For a single metric characterizing the benefits of joint modeling, we will define the *asymptotic gain* as the relative performance between our two estimators of interest here, $\hat\beta_{\mathrm{ECov}}$ and $\hat\beta_{\mathrm{ID}}$.

**Definition 5.1.** *Consider a sequence of datasets of $Q$ regression problems with an increasing number of covariates $D$, $\{\mathcal{D}_D\}_{D=1}^\infty$. Assume that for each group Condition 4.1 is satisfied with variance $\sigma^2$ and that each $\beta_d \overset{i.i.d.}{\sim} \pi$. The asymptotic gain of joint modeling is $\mathrm{Gain}(\pi, \sigma^2) := \lim_{D\to\infty}(\sigma^2 DQ)^{-1}[R_\pi^D(\hat\beta_{\mathrm{ID}}) - R_\pi^D(\hat\beta_{\mathrm{ECov}})]$.*

The factor of $\sigma^2 DQ$ in Definition 5.1 puts $\mathrm{Gain}(\pi, \sigma^2)$ on a scale that is roughly invariant to the size and noise level of the problem; for example, $(\sigma^2 DQ)^{-1}R_\pi^D(\hat\beta_{\mathrm{LS}}) = 1$ for any $\pi$, $D$, and $Q$. In Appendix D.5 we discuss how this asymptotic formulation may allow relaxation of Condition 4.1 if one considers certain random design matrices; for simplicity, the present analysis considers only fixed designs.

Our next lemma gives an analytic expression for $\mathrm{Gain}(\pi, \sigma^2)$ that provides a starting point for understanding its problem dependence.

**Lemma 5.2.** *Assume $\tilde\Sigma := \mathrm{Var}_\pi[\beta_1]$ is finite and has eigenvalues $\lambda_1, \ldots, \lambda_Q$. If Condition 4.1 satisfied asymptotically, $\mathrm{Gain}(\pi, \sigma^2) = \sigma^2 Q^{-1}[\sum_{q=1}^Q (\lambda_q + \sigma^2)^{-1} - \sum_{q=1}^Q (\tilde\Sigma_{q,q} + \sigma^2)^{-1}]$.*

Lemma 5.2 reveals that the diagonals and eigenvalues and $\tilde\Sigma$ are key determinants of $\mathrm{Gain}(\pi, \sigma^2)$, but does not directly provide an interpretation of when $\hat\beta_{\mathrm{ECov}}$ offers benefits over $\hat\beta_{\mathrm{ID}}$. Our next theorem demonstrates when an improvement can be achieved from joint modeling.

**Theorem 5.3.** $\mathrm{Gain}(\pi, \sigma^2) \geq 0$, *with equality only when $\tilde\Sigma = \mathrm{Var}_\pi[\beta_1]$ is diagonal.*

*Proof.* From Lemma 5.2 we see $\mathrm{Gain}(\pi, \sigma^2)$ is the difference between a strictly Schur-convex function applied to the eigenvalues of $\tilde\Sigma$ and to its diagonals (since $(x + \sigma^2)^{-1}$ is convex on $\mathbb{R}_+$). By the Schur-Horn theorem, the eigenvalues of $\tilde\Sigma$ majorize its diagonals, providing the result. $\qquad\square$

Theorem 5.3 tells us that $\hat\beta_{\mathrm{ECov}}$ succeeds at adaptively learning and leveraging similarities among groups in the high-dimensional limit. In particular, $\mathrm{Gain}(\pi, \sigma^2)$ reduces to zero only when the eigenvalues of $\tilde\Sigma$ are arbitrarily close to the entries of its diagonal, which occurs only when the covariate effects are uncorrelated across groups. However, when covariate effects are correlated, we obtain an improvement.

Our next theorem quantifies this relationship through upper and lower bounds.

**Theorem 5.4.** *Let $\lambda^\downarrow$ and $\ell^\downarrow$ denote the eigenvalues and diagonals of $\tilde\Sigma$, respectively, sorted in descending order. Then $\mathrm{Gain}(\pi, \sigma^2) \leq 2\sigma^2 Q^{-1}\|\lambda\|_2\|\ell^\downarrow - \lambda^\downarrow\|_2/(\lambda_{\min} + \sigma^2)^3$ and $\mathrm{Gain}(\pi, \sigma^2) \geq \sigma^2 Q^{-1}\|\ell^\downarrow - \lambda^\downarrow\|_2^2/(\lambda_{\max} + \sigma^2)^3$, where $\lambda_{\max}$ and $\lambda_{\min}$ are the largest and smallest, respectively, eigenvalues of $\tilde\Sigma$.*

Theorem 5.4 allows us to see several aspects of when our method will and will not perform well. First, the presence of $\|\ell^\downarrow - \lambda^\downarrow\|_2^2$ in both the upper and lower bounds demonstrates that $\mathrm{Gain}(\pi, \sigma^2)$ will be small when the eigenvalues are close to the diagonal entries, with Euclidean distance as an informative metric.

As we find in our next corollary, Theorem 5.4 additionally allows us to see that nontrivial gains may be obtained only in an intermediate signal-to-noise regime, where signal is given by the size of the covariate effects and noise is the variance level $\sigma^2$. Notably, under Condition 4.1, $\sigma^2$ relates directly to the variance of $\hat\beta_{\mathrm{LS}}$, and is influenced by both the residual variances and the group sizes; see Appendix C.1. In particular we interpret $\lambda_{\min}$ as a proxy for signal strength since it captures the magnitude of typical $\beta_d$'s along their direction of least variation.

**Corollary 5.5.** $\mathrm{Gain}(\pi, \sigma^2) \leq 4\kappa^2 \lambda_{\min}/\sigma^2$ *and* $\mathrm{Gain}(\pi, \sigma^2) \leq 4\kappa^2(\lambda_{\min}/\sigma^2)^{-1}$, *where $\kappa := \lambda_{\max}/\lambda_{\min}$ is the condition number of $\tilde\Sigma$.*

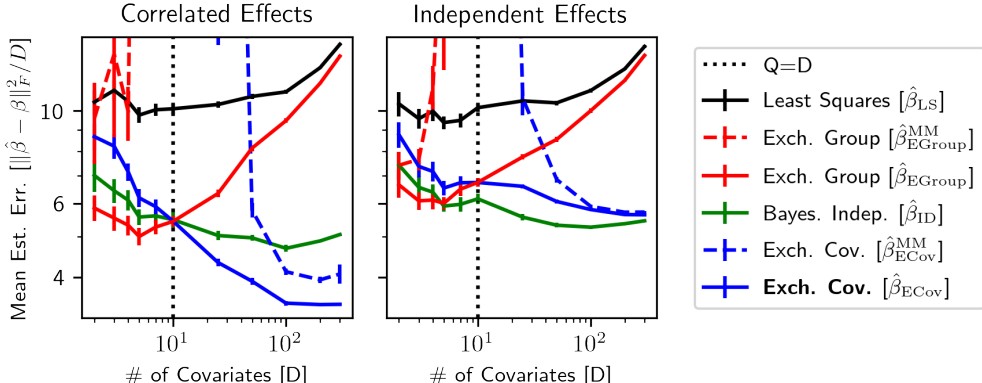

Figure 1: Dimension dependence of parameter estimation error in simulation. Covariate effects are either [Left] correlated or [Right] independent across the $Q = 10$ groups. Each point is the mean $\pm 1$SEM across 20 replicates.

Corollary 5.5 formalizes the intuitive result that with enough noise, the little recoverable signal is insufficient to effectively share strength. And furthermore, in the low-noise and high-signal regime $\hat{\beta}_{\text{ID}}$ is very accurate on its own and there is little need for joint modeling. However, when there is a large gap between the largest and smallest eigenvalues of $\hat{\Sigma}$, leading $\kappa$ to be large, the gain could be larger. $\kappa$ will be large, for example, when the covariate effects are very correlated across groups.

# 6 Experiments

## 6.1 Simulated data

We first conduct simulations, where we can directly control the relatedness among groups and where we know the ground truth values of the parameters. We show that ECov is more accurate than EGroup when covariates outnumber groups, whether effects are correlated across groups or not.

In particular, we simulated covariates, parameters, and responses for $Q = 10$ groups across a range of covariate dimensions. We generated covariate effects as $\beta_d \overset{i.i.d.}{\sim} \mathcal{N}(0, \Sigma)$. We chose $\Sigma$ so that effects were either correlated (Figure 1 Left) or independent (Figure 1 Right) across groups; see Appendix E for details. We compare performance of six estimates on these groups. These are estimates assuming EGroup/ECov using moment matching and maximum marginal likelihood to choose $\Sigma/\Gamma$ ($\hat{\beta}_{\text{EGroup}}^{\text{MM}}/\hat{\beta}_{\text{ECov}}^{\text{MM}}$ and $\hat{\beta}_{\text{EGroup}}/\hat{\beta}_{\text{ECov}}$, respectively), as well as least squares ($\hat{\beta}_{\text{LS}}$), and ECov applied to each group independently ($\hat{\beta}_{\text{ID}}$).

Figure 1 reinforces our theoretical conclusions that (1) $\hat{\beta}_{\text{ECov}}$ is more accurate when covariates outnumber groups and (2) $\hat{\beta}_{\text{EGroup}}$ is more accurate when groups outnumber covariates. Our simulated $X$ matrices are somewhat relaxed from a strict orthogonal design (Appendix E), so these experiments suggest that our conclusions hold beyond Condition 4.1. Additionally, $\hat{\beta}_{\text{ECov}}$ and $\hat{\beta}_{\text{EGroup}}$ both outperform their moment based counterparts, $\hat{\beta}_{\text{ECov}}^{\text{MM}}$ and $\hat{\beta}_{\text{EGroup}}^{\text{MM}}$.

Even for the simulations with independent effects, Theorem 4.3 suggests $\hat{\beta}_{\text{ECov}}$ should still outperform $\hat{\beta}_{\text{LS}}$ and $\hat{\beta}_{\text{EGroup}}$ in the higher dimensional regime, and we see this behavior in the right panel of Figure 1. Additionally, in agreement with Theorem 5.3, $\hat{\beta}_{\text{ECov}}$ does not improve over $\hat{\beta}_{\text{ID}}$ in the presence of independent effects, and the performances of these two estimators converge as $D$ grows.

## 6.2 Real data

We find that ECov beats EGroup, as well as least squares and independent estimation, across three real groups. We describe the datasets (with additional details in Appendix E.4) and then our results.

**Community level law enforcement in the United States.** Policing rates vary dramatically across different communities, mediating disparate impacts of criminal law enforcement across racial and socioeconomic groups [64, 54]. Understanding how demographic and socioeconomic attributes of communities relate to variation in rates of law enforcement is crucial to understanding these impacts. Linear models provide the desired interpretability. We use a dataset [51] consisting of $D = 117$ community characteristics and their rates of law enforcement (per capita) for different crimes. We consider $Q = 4$ group subsets corresponding to distinct (region, crime) pairs: (Midwest, Robbery), (South, Assault), (Northeast, Larceny), and (West, Auto-theft). This data setup illustrates a small $Q$ and accords with the independent residuals assumption in the likelihood shared by ECov and EGroup (Section 2). Across $q$, $N^q$ represents between 400 and 600 communities.

**Blog post popularity.** We regress reader engagement (responses) on $D = 279$ characteristics of blog posts (covariates) [13]. We divided the corpus based on an included length attribute into $Q = 3$ groups, corresponding to (1) long posts, (2) short posts, and (3) posts from an earlier corpus with missing length attribute. We hypothesized that the relationships between the characteristics of posts and engagement would differ across these three groups. We randomly downsampled to $N^q = 500$ posts in each group to mimic a low sample-size regime, in which sharing strength is crucial.

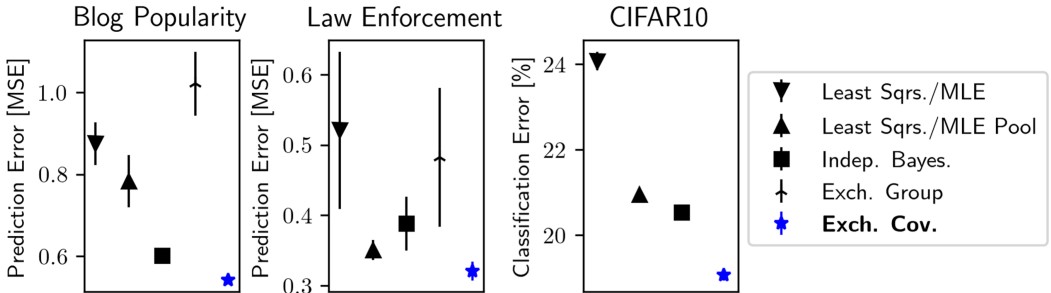

Figure 2: Prediction performance on held out data in three applications (mean $\pm 1$SEM across 5-fold cross-validation splits).

**Multiple binary classifications using pre-trained neural network embeddings on CIFAR10.** Modern machine learning methods have proved very successful on large datasets. Translating this success to smaller datasets is one of the most actively pursued algorithmic challenges in machine learning. It has spurred the development of frameworks from transfer learning [65] to one-shot learning [62] to meta-learning [21]. One common and simple strategy starts with a learned representation (or "embedding") from an expressive neural network fit to a large group. Then one can use this embedding as a covariate vector for classification tasks with few labeled data points.

We take a $D = 128$ dimensional embedding of the CIFAR10 image group [37, 60]. We create $Q = 8$ different binary classification tasks using the classes in CIFAR10 (Appendix E.4). We downsampled to $N^q$ varying from 100 to 1000 to mimic a setting in which we hope to share strength from large groups to improve performance on smaller datasets.

**Discussion of evaluation and results.** In previous sections we have focused on parameter estimation. Here we instead evaluate with prediction error on held-out data since the true parameters are not observed. Specifically we perform 5-fold cross-validation and report the mean squared errors and classification errors on test splits. To reduce variance of out-of-sample error estimates on the applications in which we downsampled, we also evaluate on the additional held-out data. Because the residual variances were unknown, we estimated these for each application and group as $\hat{\sigma}_q^2 :=$ $\|P_{X^q}^{\perp} Y^q\|^2 / (N_q - D)$, where $P_{X^q}^{\perp} := I_{N_q} - X^q (X^{q\top} X^q)^{-1} X^{q\top}$ (see e.g. [23, Chapter 18.1]). All methods ran quickly on a 36 CPU machine; computation of $\hat{\beta}_{\text{ECov}}$, including the EM algorithm, required $2.04 \pm 0.64$, $6.89 \pm 3.19$ and $37.14 \pm 3.39$ seconds (`mean ± st-dev` across splits) on the law enforcement, blog, and CIFAR10 tasks, respectively.

Our results further reinforce the main aspects of our theory. $\hat{\beta}_{\text{ECov}}$ outperformed $\hat{\beta}_{\text{EGroup}}$, independent Bayes estimates ($\hat{\beta}_{\text{ID}}$), and least squares ($\hat{\beta}_{\text{LS}}$) in all applications (at $> 95\%$ nominal confidence

with a paired t-test).[2] Additionally, $\hat{\beta}_{\text{ECov}}$ outperformed the baseline of ignoring heterogeneity, pooling groups together, and using the same effect estimates for every group ("Least Sqrs./MLE Pool").

Appendix E includes additional results and comparisons. In particular, we provide the performance of the estimators on each component group for each application. Additionally, we report the performances of (1) stable and computationally efficient moment based alternatives to $\hat{\beta}_{\text{ECov}}$ and $\hat{\beta}_{\text{EGroup}}$ and (2) variants of $\hat{\beta}_{\text{ECov}}$ and $\hat{\beta}_{\text{EGroup}}$ that include a learned (rather than zero) prior mean. Appendix E.5 reports the licenses of software we used.

## 7 Discussion

The Bayesian community has long used hierarchical modeling with priors encoding exchangeability of effects across groups of data (EGroup). In the present work, we have made a case for instead using priors that encode exchangeability across *covariates* (ECov) – in particular, when the number of covariates exceeds the number of groups. We have presented a corresponding concrete model and inference method. We have shown that ECov outperforms EGroup in theory and practice when the number of covariates exceeds the number of groups.

Our approach is, of course, not a panacea. In some settings, a priori exchangeability among covariate effects will be inconsistent with prior beliefs. For example, imagine in the CIFAR10 application if meta-data covariates (such as geo-location and date) were available, in addition to embeddings. Then we might achieve better performance by treating meta-data covariates as distinct from embedding covariates. Additionally, we focused on a Gaussian prior for convenience. In cases where practitioners have more specific prior beliefs about effects, alternative priors and likelihoods may be warranted, though they may be more computationally challenging. Moreover, while relatively interpretable, linear models have their downsides. The linear assumption can be overly simplistic in many applications. It is common to misinterpret effects as causal rather than associative. Both the linear model and squared error loss lend themselves naturally to reporting means, but in many applications a median or other summary is more appropriate; so using a mean for convenience can be misleading.

Many exciting directions for further investigation remain. For example, the covariance $\Sigma$ may provide an informative measure of task similarity; this similarity measure can be useful in, e.g., meta learning [34] and statistical genetics [12]. Additionally, we here explored two approaches to choosing the covariance matrices in the empirical Bayes step; more sophisticated approaches to covariance estimation may provided improved performance. It also remains to extend our methodology to other generalized linear models.

## Acknowledgments and Disclosure of Funding

The authors thank Sameer K. Deshpande, Ryan Giordano, Alex Bloemendal, Lorenzo Masoero, and Diana Cai for insightful discussions on the manuscript, and the anonymous reviewers for their constructive suggestions. This work was supported in part by ONR Award N00014-18-S-F006 and an NSF CAREER Award. BLT is supported by NSF GRFP.

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
