# A  Additional Related Work

## A.1  Brown and Zidek details

As discussed in Section 1, the papers of Brown and Zidek [10] and Haitovsky [28] carry the only references of which we are aware of the idea of exchangeability of effects across covariates for sharing strength among multiple groups of data. We here provide additional discussion on this related prior work. To aid our comparison, we slightly modify their notation to match ours.

In their paper, "Adaptive Multivariate Ridge Regression", Brown and Zidek [10] consider multiple related regression regression problems with a shared design (i.e. $X := X^1 = X^2 = \cdots = X^Q$) and seek to extend the univariate ridge regression estimator of Hoerl and Kennard [31] to the multivariate setting. Specifically, the authors propose a class of estimators of the form

$$\hat{\vec{\beta}} = (I_Q \otimes X^\top X + K \otimes I_D)^{-1}(I_Q \otimes X^\top)\vec{Y},$$

where $\vec{Y} := [Y^{1\top}, Y^{2\top}, \cdots, Y^{Q\top}]^\top$, $\otimes$ denotes the Kronecker product, and $K$ is a $Q \times Q$ ridge matrix which they suggest be chosen by some "adaptive rule" (i.e. that $K$ be a function of the observed data). Notably, this functional form closely resembles our expression for $\mathbb{E}[\vec{\beta}|\mathcal{D}, \Sigma]$ in Proposition 3.1, if we take $K = \Sigma^{-1}$.

The authors do not explicitly discuss the interpretation of $K^{-1}$ as the covariance of a Gaussian prior, nor any interpretation for this quantity as capturing any notion of a priori similarity of the regression problems. However, they do point to Bayesian motivations at the outset of the paper. In particular, Brown and Zidek [10] narrow their consideration of possible methods for choosing $K$ to those which satisfy two criteria:

1. For any $K$, $\hat{\vec{\beta}}$ correspond to a Bayes estimate.

2. In the case that $X^\top X = I_D$, $\hat{\vec{\beta}}$ correspond to the Efron and Morris [19] extension of the James and Stein [33] estimator to vector observations.[3]

They present four such estimators (derived from existing estimators of a multivariate normal means that dominate the sample mean) and demonstrate conditions under which each of these estimators dominates the least squares estimator for $\beta$.

As a further point of connection, the authors claim in the their abstract that their "result is implicitly in the work of Lindley and Smith [44] although not actually developed there." However, the authors give little support for, or clarification of this claim. In particular, their analysis is entirely frequentist and they provide no explanation for how their proposed estimators for $K$ might be interpreted as reasonable empirical Bayes estimates.

In their short follow-up paper, Haitovsky [28] elaborates on this Bayesian motivation. The primary focus of Haitovsky [28] is a matrix normal prior [16] that captures structure in effects across both groups and covariates. Though this prior is not exchangeable across covariates in general, they note that the special case of where effects are uncorrelated across different covariates satisfies the notion of exchangeability for which we have advocated in this paper.

## A.2  Methods of inference for $\Gamma$ in existing work assuming exchangeability of effects across groups.

We here describe several existing approaches for estimating the covariance matrix $\Gamma$ in the exchangeability of effects among groups model. These existing methods do not translate directly to the exchangeability of effects among covariates model proposed in this paper. However, in principle, one could likely adapt any of them to our setting. We have chosen to use the EM algorithm described in Section 3 for its simplicity, efficiency, and stability. We leave the investigation of alternative estimation approaches to future work.

In their initial paper, Lindley and Smith (1972) [44] suggest that a fully Bayesian approach would be ideal. They advocate for placing a subjectively specified, conjugate Wishart prior on $\Gamma$, and remark

---

[3] See Appendix A.4 for further discussion of connections to Efron and Morris [19].

that one should ideally consider the posterior of $\Gamma$ rather than relying on a point estimate. However, in the face of analytic intractability, they propose returning MAP estimates for $\Gamma$ and $\beta$ and provide an iterative optimization scheme that they show is stationary at $\hat{\Gamma}, \hat{\beta} = \arg\max \log p(\Gamma, \beta | \mathcal{D})$.

Advances in computational methods since 1972 have given rise to other ways of estimating $\Gamma$ in this model. Gelfand et al. [22] describe a Gibbs sampling algorithm for posterior inference. Gelman et al. [24, Chapter 15 sections 4-5] describe an EM algorithm which returns a maximum a posteriori estimate marginalizing over $\beta$, $\hat{\Gamma} = \arg\max p(\Gamma|\mathcal{D}) = \int p(\Gamma, \beta|\mathcal{D})d\beta$; notably, though the updates in our EM algorithm for the case of exchangeability in effects across covariates differ from those in the case of exchangeability among groups, one can see the two algorithms as closely related through their shared dependence on Gaussian conjugacy. Finally, in the software package `lme4`, Bates et al. [5] use the maximum marginal likelihood estimate, $\hat{\Gamma} = \arg\max p(\mathcal{D}|\Gamma)$, which they compute using gradient based optimization.

## A.3 Details on connections to `lme4`

In the notation of `lme4` [5], our paper considers only random effects and no fixed effects. In that work, each vector of random effects, denoted $\mathcal{B}$, corresponds to a length $D$ ($q$ in their notation) column of $\beta$ (in our notation). Bates et al. [5, Equation 3] states the prior derived from Lindley and Smith [44] that reflects the assumption of exchangeability across groups and captures correlation structure across covariates. This correlation structure is modeled whenever two or more random effects are specified and allowed to vary across groups. In the high dimensional setting (when $D > Q$), however, `lme4` fails to run because the optimization problem associated with empirical Bayes step is ill-conditioned.

## A.4 Related work on estimation of normal means

As we discuss in Appendix C.1, under Condition 4.1 and when $\sigma^2 = 1$, we have that

$$\hat{\beta}_{\text{LS}}^q \overset{indep}{\sim} \mathcal{N}(\beta^q, I_D).$$

As such, inference reduces to the "normal means problem", with a matrix valued parameter. Specifically, we can equivalently write

$$\hat{\beta}_{\text{LS}} = \beta + \epsilon,$$

for a random $D \times Q$ matrix $\epsilon$ with i.i.d. standard normal entries.

This problem has been studied closely outside of the context of regression. Notably, Efron and Morris [18] approach the problem from an empirical Bayesian perspective and recommend an approach analogous to estimating $\Sigma$ by

$$\hat{\Sigma}^{\text{Ef}} := (D - Q - 1)^{-1}\hat{\beta}_{\text{LS}}^{\top}\hat{\beta}_{\text{LS}} - I_Q.$$

Efron and Morris [18] argue for this estimate because it is unbiased for a transformation of the parameter. In particular, $\hat{\Sigma}^{\text{Ef}}$ satisfies $\mathbb{E}[(I_Q + \hat{\Sigma}^{\text{Ef}})^{-1}] = (I_Q + \Sigma)^{-1}$ when each $\beta_d \overset{i.i.d.}{\sim} \mathcal{N}(0, \Sigma)$. They show that, among all estimates of the form $\alpha\hat{\beta}_{\text{LS}}^{\top}\hat{\beta}_{\text{LS}} - I_Q$ with real valued $\alpha$, this factor $\alpha = (D - Q - 1)^{-1}$ is optimal in terms of squared error risk. Notably, this includes the moment estimate $\hat{\Sigma}^{\text{MM}}$ we describe in Section 4, which corresponds to $\alpha = D^{-1}$. However, this optimality result does not translate to the associated positive part estimators. In fact, in experiments not shown, we have found that $\hat{\beta}_{\text{ECov}}$ reliably outperforms an analogous positive part variant that estimates $\Sigma$ by $\hat{\Sigma}^{\text{Ef}}$.

*Remark* A.1. Efron and Morris [18, Theorem 5] prove that an analogous positive part estimator is superior to their original estimator in term of "relative savings loss" (RSL). Our domination result in Theorem 4.3 is strictly stronger and implies an improvement in RSL as well. Furthermore our proof technique immediately applies to their estimator.

Several other works have noted the dependence of the risk of estimators for the matrix variate normal means problem on the expectations of the eigenvalues of inverse non-central Wishart matrices [18, 70, 59]. In all of these cases, the authors did not document attempts to interpret or approximate these difficult expectations.

More recently, Tsukuma [58] explores a large class of estimators for the matrix variate normal means problems that shrink $\hat{\beta}_{\text{LS}}$ along the directions of its singular vectors in different ways. For subclass

of these estimators, Tsukuma [58][Corollary 3.1] proves a domination result for associated positive part estimators. In the orthogonal design case, $\hat{\beta}_{\mathrm{ECov}}$ can be shown to be a member of this subclass of estimators, providing an alternative route to proving Theorem 4.5.

### A.5 Additional related work on multiple related regressions

Methods for simultaneously estimating the parameters of multiple related regression problems have a long history in statistics and machine learning, with different assumptions and analysis goals leading to a diversity of inferential approaches. Perhaps the most famous is Zellner's landmark paper on seemingly unrelated regressions (SUR) [67]. Zellner [67] addresses the situation where apparent independence of regression problems is confounded by covariance in the errors across $Q$ problems (i.e. 'groups' in our language). In the presence of such correlation in residuals, the parameter may be identified with greater asymptotic statistical efficiency by considering all $Q$ problems together [67, 68]. While most work on SUR has taken a purely frequentist perspective in which $\beta$ is assumed fixed, some more recent works on SUR have considered Bayesian approaches to inference [8, 15, 55, 27, 2]. However these do not address the scenario of interest here, in which we believe *a priori* that there may be some covariance structure in the effects of covariates *across* the regressions, or that some regression problems are more related than others. The setting of the present paper further differs from SUR in that we do not consider correlation in residuals as a possible mechanism for sharing strength between groups, but instead explicitly assume independence in the noise.

Breiman and Friedman [9] present a distinct, largely heuristic approach to multiple related regression problems where all $Q$ responses are observed for each group, or equivalently each group has the same design. The authors focus entirely on prediction and obviate the need share information across regression problems when forming an initial estimate of $\beta$ by proposing to predict new responses in each regression with a linear combination of the predictions of linear models defined by the independently computed least squares estimate of each regression problem. However this approach does not consider the problem of estimating parameters, which is a primary concern of the present work.

Reinsel [52]'s paper, "Mean Squared Error Properties of Empirical Bayes Estimators in a Multivariate Random Effects General Linear Model", considers a mixed effects model in which a linear model for regression coefficients is specified $\beta^q = Ba_q + \lambda_q$ where $a := [a_1, a_2, \ldots, a_Q]$ is a $K \times Q$ known design matrix associated with the regression problems,[4] $B$ is a $D \times K$ matrix of unknown parameters and $[\lambda_1, \lambda_2, \ldots, \lambda_Q]$ is a $D \times Q$ matrix of error terms. These error terms are assumed exchangeable across groups. In contrast to the present work, Reinsel [52] requires the relatedness between groups to be known a priori through the known design matrix $a$.

Laird and Ware [38] consider a random effects model for longitudinal data in which different individuals correspond to different regression problems with distinct parameters. In their construction, covariance structure in the noise is allowed across the observations for each individual, but not across individuals. Additionally, as in [44], the authors model the covariance in effects of different covariates a priori within each regression, but not covariance across regressions.

Brown et al. [11] propose to use sparse prior for $\beta$ which encourages a shared sparsity pattern. Conditioned on a binary $D-$vector $\gamma \in \{0, 1\}^D$, $\beta$ is supposed to follow a multivariate normal prior as

$$\vec{\beta} \overset{i.i.d.}{\sim} \mathcal{N}(0, \Sigma \otimes H_\gamma)$$

where $H_\gamma$ is a $D \times D$ covariance matrix which expresses that for $d$ such that $\gamma_d = 0$ we expect each $\beta_{d,q}$ to be close to zero. Notably, this is equivalent to the assumption that $\beta$ follows a matrix-variate multivariate normal distributed as $\beta \sim \mathcal{MN}(0, H_\gamma, \Sigma)$ [16]. Curiously, and without stated justification, the same $\Sigma$ is also taken to parameterize the covariance of the residual errors, as well as of an additional bias term. We suspect this restriction is made for the sake of computational tractability. Indeed, [56] makes similar modeling assumptions for tractability in the context of statistical genetics. In contrast to the present work, the premise of Brown et al. [11] is sharing strength through similar sparsity patterns and covariance in the residuals, rather than learning and leveraging patterns of similarity in effects of covariates across groups.

---

[4] Notably, though Reinsel [52] refers to $a$ as a design matrix, it has little relation of the design matrices $X^q$ to which we frequently refer in the present work.

Other more recent papers have considered alternative approaches for multiple regression with sparse priors [6, 43, 17]. As one example, Obozinski et al. [50] estimate parameters across multiple groups with a mixed $\ell_1/\ell_2$ regularized objective that induces sparsity. Yang et al. [66], Lee et al. [40] build on this work by Obozinski et al. [50] with a focus on applications in genetics. These latter methods may be understood as returning the maximum a posteriori estimate under a Bayesian model. However, in contrast to our approach, the corresponding prior distributions implicit in such perspectives do not capture a priori correlation of effects across groups. Moreover, these methods are of course inappropriate when we do not expect sparsity a priori.

**Meta-Learning** The popular "Model Agnostic Meta-Learning" (MAML) approach [21] can be understood as a hierarchical Bayesian method that treats tasks / groups exchangeably [26]. As such, MAML and its variations do not allow tasks to be related to different extents (as our approach does). A few recent works on meta-learning are exceptions; for example, Jerfel et al. [34] model tasks as grouped into clusters by using a Dirichlet process prior, and Cai et al. [14] consider a weighted variant of MAML that allows, for a given task of interest, the contribution of data from other tasks to vary. However these works differ from the present paper in their focus on prediction with flexible black-box models, whereas the primary concern of the present is parameter estimation in linear models.

**Exchangeability of effects across covariates in the single group context.** In the context of regression problems consisting of only a single group (i.e. corresponding to the special case of $Q = 1$) Lindley and Smith [44] suggest modeling the $D$ scalar covariate effects exchangeable. In particular, they suggest modeling scalar covariate effects as i.i.d. from a univariate Gaussian prior when this exchangeability assumption is appropriate. However, because this development is restricted to analyses of data in a single group, it does not relate to the problem of sharing strength across multiple groups, which is the subject of the present work.

# B    Section 3 supplementary proofs and discussion

## B.1    Proof of Proposition 3.1

*Proof.* First note that the least squares estimates $\hat{\beta}_{\text{LS}} := [(X^{1\top}X^1)^{-1}X^{1\top}Y^1, \ldots, (X^{Q\top}X^Q)^{-1}X^{Q\top}Y^Q]$ are a sufficient statistic of $\mathcal{D}$ for $\beta$, and so $\beta|\mathcal{D}, \Sigma \sim \beta|\hat{\beta}_{\text{LS}}, \Sigma$. As such, it is sufficient to consider the likelihood of $\hat{\beta}_{\text{LS}}$. Let $\vec{\hat{\beta}}_{\text{LS}} := [Y^{1\top}X^1(X^{1\top}X^1)^{-1}, \ldots, Y^{Q\top}X^Q(X^{Q\top}X^Q)^{-1}]$ be the $DQ$-vector defined by stacking the least squares estimates for each group. Since for each $q$, we have $\hat{\beta}_{\text{LS}}^q|\beta \overset{indep.}{\sim} \mathcal{N}(\beta^q, \sigma_q^2(X^{q\top}X^q)^{-1})$, we can write $\vec{\hat{\beta}}_{\text{LS}}|\beta \sim \mathcal{N}\left[\vec{\beta}, \text{diag}\left(\sigma_1^2(X^{1\top}X^1)^{-1}, \ldots, \sigma_Q^2(X^{Q\top}X^Q)^{-1}\right)\right]$. Next, that each $\beta_d \overset{i.i.d.}{\sim} \mathcal{N}(0, \Sigma)$ a priori implies that we may write $\vec{\beta} \sim \mathcal{N}(0, \Sigma \otimes I_D)$ a priori, where $\otimes$ is the Kronecker product. Then, by Gaussian conjugacy (see e.g. Bishop [7, Chapter 2.3]), we have that $\vec{\beta}|\mathcal{D} \sim \mathcal{N}(\vec{\mu}, V)$, where $\vec{\mu} = V\left[(\Sigma \otimes I_D)^{-1}0 + \text{diag}\left(\sigma_1^2(X^{1\top}X^1)^{-1}, \ldots, \sigma_Q^2(X^{Q\top}X^Q)^{-1}\right)^{-1} \vec{\hat{\beta}}_{\text{LS}}\right]$ for $V^{-1} = (\Sigma \otimes I_D)^{-1} + \text{diag}\left(\sigma_1^2(X^{1\top}X^1)^{-1}, \ldots, \sigma_Q^2(X^{Q\top}X^Q)^{-1}\right)^{-1}$. Due to the block structure of the matrices above, these simplify to $\vec{\mu} = V\left[\frac{Y^{1\top}X^1}{\sigma_1^2}, \ldots, \frac{Y^{Q\top}X^Q}{\sigma_Q^2}\right]$ and $V^{-1} = \Sigma^{-1} \otimes I_D + \text{diag}(\frac{X^{1\top}X^1}{\sigma_1^2}, \ldots, \frac{X^{Q\top}X^Q}{\sigma_Q^2})$, as desired. □

## B.2    Efficient computation with the conjugate gradient algorithm

As mentioned in Section 3.1, $\vec{\mu} = \mathbb{E}[\vec{\beta}|\mathcal{D}, \Sigma]$ in Proposition 3.1 may be computed efficiently using the conjugate gradient algorithm (CG) for solving linear systems. We here describe several properties of CG that make it surprisingly well-suited to this application.

We first note that Proposition 3.1 allows us to frame computation of $\vec{\mu}$ as the solution to the linear system

$$A\vec{\mu} = b$$

for $b = \left[Y^{1\top}X^1/\sigma_1^2, \ldots, Y^{Q\top}X^Q/\sigma_Q^2\right]^\top$ and $A = \Sigma^{-1} \otimes I_D +$ $\operatorname{diag}\left(\sigma_1^{-2}X^{1\top}X^1, \ldots, \sigma_Q^{-2}X^{Q\top}X^Q\right)$. A naive approach to computing $\vec{\mu}$ could then be to explicitly compute $A^{-1}$ and report the matrix vector product, $A^{-1}b$. However, as mentioned in Section 3.1, since $A$ is a $DQ \times DQ$ matrix, explicitly computing its inverse would require roughly $O(D^3Q^3)$ time. This operation becomes very cumbersome when $D$ and $Q$ are too large; for instance if $D$ and $Q$ are in the hundreds the, $DQ$ is is the tens of thousands.

CG provides an exact solution to linear systems in at most $DQ$ iterations, with each iteration requiring only a small constant number of matrix vector multiplications by $A$. This characteristic does not provide a complexity improvement for solving general linear systems because for dense, unstructured $DQ \times DQ$ matrices, matrix vector multiplies require $O(D^2Q^2)$ time, and CG still demands $O(D^3Q^3)$ time overall. However this property provides a substantial benefit in our setting. In particular, the special form of $A$ allows computation of matrix vector multiplications in $O(D^2Q)$ rather than $O(D^2Q^2)$ time, and storage of this matrix with $O(D^2Q)$ rather than $O(D^2Q^2)$ memory. Specifically, if $v = [v_1, v_2, \ldots, v_Q]$ is a $D \times Q$ matrix with $D$-vector columns $v_q$, for the $DQ$-vector $\vec{v} = [v_1^\top, v_2^\top, \ldots, v_Q^\top]^\top$ we can compute $A\vec{v}$ as $\operatorname{vec}\left(v\Sigma^{-1}\right) + [\sigma_1^{-2}X^{1\top}X^1v_1, \ldots, \sigma_Q^{-2}X^{Q\top}X^Qv_Q]^\top$, where $\operatorname{vec}(\cdot)$ represents the operation of reshaping an $D \times Q$ matrix into a $DQ$-vector by stacking its columns. When $D > Q$, this operation is dominated by the $Q$ $O(D^2)$ matrix-vector multiplications to compute the second term. As such, CG provides an order $Q$ improvement in both time and memory.

Next, CG may be viewed as an iterative optimization method. At each step it provides an iterate which is the closest to the $\vec{\mu}$ on a Krylov subspace of expanding dimension. As such, the algorithm may be terminated after fewer than $DQ$ steps to provide an approximation of the solution. Moreover, the algorithm may be provided with an initial estimate, and improves upon that estimate in each successive iteration. In our case we may readily compute a good initialization. For example, we can initialize with the posterior mean of the parameter for each group when conditioning on that group alone, i.e. $\vec{\mu}^{(0)} := \left[\mathbb{E}[\beta^1|Y^1]^\top, \ldots, \mathbb{E}[\beta^Q|Y^Q]^\top\right]^\top$.

Finally, the convergence properties of the conjugate gradient algorithm are well understood. Notably the $i$th iterate of conjugate gradient $\vec{\mu}^{(i)}$ when initialized at $\vec{\mu}^{(0)}$ satisfies

$$\|\vec{\mu}^{(i+1)} - \vec{\mu}\|_A \leq 2\left(\frac{\kappa - 1}{\kappa + 1}\right)^i \|\vec{\mu}^{(0)} - \vec{\mu}\|_A,$$

where $\kappa = \sqrt{\frac{\lambda_{\max}(A)}{\lambda_{\min}(A)}}$ is the square root of the condition number of $A$, and $\|\cdot\|_A$ is the $A-$quadratic norm [49, Chapter 5.1], [45]. Since $A$ will often be reasonably well conditioned (note, for example, that $\lambda_{\min}(A) \geq \lambda_{\min}(\Sigma)$), convergence can be rapid. Notably, in an unpublished application the authors encountered (not described in this work) involving $D \approx 20,000$ covariates and $Q \approx 50$ groups, the approximately million dimensional estimate $\vec{\mu}$ was computed in roughly 10 minutes on a 16 core machine.

### B.3  Expectation maximization algorithm further details

In Sections 3.2 and 3.3 we introduced EM algorithms for estimating $\Sigma$ for both linear and logistic regression models. In this subsection we provide a derivation of the updates in Algorithm 1 and discuss computational details of our fast implementation.

**Derivations of EM updates for linear regression.**  Our notation inherits directly from [47, Chapter 1.5], to which we refer the reader for context. In our application of the EM algorithm, we take the collection of all covariate effects $\beta$ as the 'missing data.' For the expectation (E) step, we therefore

require

$$Q(\Sigma, \Sigma^{(i)}) := \mathbb{E}[\log p(\beta|\Sigma)|\mathcal{D}, \Sigma^{(i)}]$$

$$= c + \frac{D}{2}\log|\Sigma^{-1}| - \frac{1}{2}\sum_{d=1}^{D}\mathbb{E}[\beta_d^\top \Sigma^{-1}\beta_d|\mathcal{D}, \Sigma^{(i)}]$$

$$= c + \frac{D}{2}\log|\Sigma^{-1}| - \frac{1}{2}\sum_{d=1}^{D}\mathrm{tr}\left(\Sigma^{-1}\mathbb{E}[\beta_d\beta_d^\top|\mathcal{D}, \Sigma^{(i)}]\right) \qquad (2)$$

$$= c + \frac{D}{2}\log|\Sigma^{-1}| - \frac{1}{2}\sum_{d=1}^{D}\mathrm{tr}\left(\Sigma^{-1}(\mu_d\mu_d^\top + V_d)\right),$$

where $c$ is a constant that does not depend on $\Sigma$, $\mu = [\mu_1 \dots, \mu_D]^\top := \mathbb{E}[\beta|\mathcal{D}, \Sigma^{(i)}]$ and for each $d$ $V_d := (I_Q \otimes e_d)^\top \mathrm{Var}[\vec{\beta}|\mathcal{D}, \Sigma^{(i)}](I_Q \otimes e_d)$. From the last line of Equation (2) we may see that $\mu$ and $\{V_d\}_{d=1}^{D}$, comprise the required posterior expectations.

The solution to the maximization step may then be found by considering a first order condition for maximizing over $\Sigma^{-1}$ rather than $\Sigma$. Observe that $\frac{\partial}{\partial\Sigma^{-1}}Q(\Sigma, \Sigma^{(i)}) = \frac{D}{2}\Sigma - \frac{1}{2}\sum_{d=1}^{D}(\mu_d\mu_d^\top + V_d)$. Setting this to zero we obtain $\Sigma^{(i+1)} = D^{-1}\sum(\mu_d\mu_d^\top + V_d)$. This is the desired update for the M-step provided in Algorithm 2.

**Logistic regression EM updates.** The updates for the approximate EM algorithm described in Section 3 are derived from a Gaussian approximation to the posterior under which the expectation of log prior is taken. In particular we approximate the first line of Equation (2) as

$$Q(\Sigma, \Sigma^{(i)}) := \mathbb{E}[\log p(\beta|\Sigma)|\mathcal{D}, \Sigma^{(i)}]$$

$$= \int p(\beta|\mathcal{D}, \Sigma^{(i)})\log p(\beta|\Sigma)d\beta \qquad (3)$$

$$\approx \int q^{(i)}(\beta)\log p(\beta|\Sigma)d\beta$$

where $q^{(i)}$ denotes the Laplace approximation to $p(\beta|\mathcal{D}, \Sigma^{(i)})$. Specifically, as we summarized in Algorithm 3, we approximate the posterior mean by the maximum a posteriori estimate, $\vec{\mu}^* := \arg\max_{\vec{\beta}}\log p(\vec{\beta}|\mathcal{D}, \Sigma^{(i)})$, and the posterior variance by $V := -[\nabla_{\vec{\beta}}^2\log p(\vec{\beta}|\mathcal{D}, \Sigma^{(i)})|_{\vec{\beta}=\vec{\mu}^*}]^{-1}$. We the let $q^{(i)}$ be the Gaussian density with these moments. This renders the integral in the last line of Equation (3) tractable, and updates are derived in the same way as in the linear case.

Naively, the approximate EM algorithm for logistic regression could be much more demanding than its counterpart in the linear case. In particular, at each iteration we need to solve a convex optimization problem, rather than linear system. However, in practice the algorithm is only little more demanding because, by using the maximum a posteriori estimate from the previous iteration to initialize the optimization, we can solve the optimization problem very easily. In particular, after the first few EM iterations, only one or two additional Newton steps from this initialization are required.

To simplify our implementation, we used automatic differentiation in `Tensorflow` to compute gradients and Hessians when computing the maximum a posteriori values and Laplace approximations.

**Computational efficiency.** We have employed several tricks to provide a fast implementation of our EM algorithms. The M-Steps for both linear and logistic regression involve a series of expensive matrix operations. To accelerate this, we used `Tensorflow`[1] to optimize these steps by way of a computational graph representation generated using the `@tf.function` decorator in python. Additionally, we initialize EM with a moment based estimate (see Appendix E.2).

## C Frequentist properties of exchangeability among covariate effects – supplementary proofs and discussion

### C.1 Discussion of Condition 4.1

The restriction on the design matrices in Condition 4.1 places strong limits the immediate scope of our theoretical results. However, as with many statistical assumptions such as Gaussianity of residuals, this condition lends considerable tractability to the problem that enables us to build insights that we can see hold in more relaxed settings in experiments (see Section 6).

Under Condition 4.1 estimation of the parameter $\beta$ may be reduced to a special matrix valued case of the normal means problem with each $\hat{\beta}^q_{\text{LS},d} \sim \mathcal{N}(\beta^q_d, \sigma^2)$. Accordingly, we may recognize $\sigma^2$ as a reflection of both the residual variances $\sigma^2_q$ and sample sizes $N_q$. In particular, if within each group $q$ the covariates have sample second moment $N_q^{-1} \sum_{n=1}^{N_q} X^q_n X^{q\top}_n = I_D$, and the residual variances and sample sizes are equal (i.e. $\sigma^2_1 = \sigma^2_2 = \cdots = \sigma^2_Q$ and $N^1 = N^2 = \cdots = N^Q$), then $\sigma^2 = \sigma^2_1/N^1$. Additionally, because $\hat{\beta}_{\text{LS}}$ is a sufficient statistic of $\mathcal{D}$ for $\beta$, it suffices to consider $\hat{\beta}_{\text{LS}}$ alone, without needing to consider other aspects of $\mathcal{D}$. For these reasons, conditions of this sort are commonly assumed by other authors in related settings (e.g. van Wieringen [61, Chapters 1.4 and 6.2] and Fan and Li [20], Golan and Perloff [25]).

That the trends predicted by our theoretical results persist beyond the limits of Condition 4.1 should not be surprising. The likelihood, our estimators and their risks are all continuous in the $X^q$, and so domination results may be seen to extends via continuity to settings with well-conditioned designs. On the other hand, problems with design matrices that are more poorly conditioned are more challenging for both theory and estimation in practice (see e.g. Brown and Zidek [10][Example 4.2]).

### C.2 A proposition on analytic forms of the risks of moment estimators

The following proposition characterizes analytic expressions for the moment based estimators. These expressions provide a starting point for the theory in Section 4

**Proposition C.1.** *Assume each $Y^q_n|X^q_n, \beta^q \sim \mathcal{N}(X^{q\top}_n \beta^q, \sigma^2_q)$ and define $\hat{\Sigma}^{\text{MM}} := D^{-1}\hat{\beta}^\top_{\text{LS}}\hat{\beta}_{\text{LS}} - D^{-1}\text{diag}(\sigma^2_1\|X^{1\dagger}\|^2_F, \ldots, \sigma^2_Q\|X^{Q\dagger}\|^2_F)$. Then*

    *1. if each $\beta_d \overset{i.i.d.}{\sim} \mathcal{N}(0, \Sigma)$, $\mathbb{E}[\hat{\Sigma}^{\text{MM}}] = \Sigma$.*

*Furthermore, under Condition 4.1*

    *2. when $D \geq Q$, $\hat{\beta}^{\text{MM}}_{\text{ECov}} = \hat{\beta}_{\text{LS}} - \sigma^2 D\hat{\beta}^{\dagger\top}_{\text{LS}}$ and*

    *3. when $D \leq Q$, $\hat{\beta}^{\text{MM}}_{\text{EGroup}} = \hat{\beta}_{\text{LS}} - \sigma^2 Q\hat{\beta}^{\dagger\top}_{\text{LS}}$,*

*where $\dagger$ denotes the Moore-Penrose pseudoinverse of a matrix.*

*Proof.* We begin with statement (1), that under Condition 4.1 and correct prior specification, $\mathbb{E}[\hat{\Sigma}^{\text{MM}}] = \Sigma$. Recall that $\hat{\Sigma}^{\text{MM}} := D^{-1}\hat{\beta}^\top_{\text{LS}}\hat{\beta}_{\text{LS}} - D^{-1}\text{diag}(\sigma^2_1\|X^{1\dagger}\|^2_F, \ldots, \sigma^2_Q\|X^{Q\dagger}\|^2_F)$. For any fixed $\beta$, we have $\mathbb{E}[\hat{\Sigma}^{\text{MM}}|\beta] = D^{-1}\mathbb{E}[\hat{\beta}^\top_{\text{LS}}\hat{\beta}_{\text{LS}}|\beta] - D^{-1}\text{diag}(\sigma^2_1\|X^{1\dagger}\|^2_F, \ldots, \sigma^2_Q\|X^{Q\dagger}\|^2_F)$, and so seek to characterize $\mathbb{E}[\hat{\beta}^\top_{\text{LS}}\hat{\beta}_{\text{LS}}|\beta]$. Note that we may write $\hat{\beta}_{\text{LS}} \overset{d}{=} \beta + \epsilon$ for a random $D \times Q$ matrix $\epsilon$ with each column $q$ distributed as $\epsilon^q \overset{indep.}{\sim} \mathcal{N}\left[0, \sigma^2_q(X^{q\top}X^q)^{-1}\right]$. As such, for each $q$ we have $\mathbb{E}[\hat{\beta}^{q\top}_{\text{LS}}\hat{\beta}^q_{\text{LS}}|\beta] = \beta^{q\top}\beta^q + \mathbb{E}[\epsilon^{q\top}\epsilon^q]$. Next observe that $\mathbb{E}[\epsilon^{q\top}\epsilon^q] = \text{tr}[\sigma^2_q(X^{q\top}X^q)^{-1}] = \sigma^2_q\|X^{q\dagger}\|^2_F$, where $\dagger$ denotes the pseudo-inverse of a matrix and $\|\cdot\|_F$ is the Frobenius norm. Additionally, for $q \neq q'$, we have $\mathbb{E}[\hat{\beta}^{q\top}_{\text{LS}}\hat{\beta}^{q'}_{\text{LS}}|\beta] = \beta^{q\top}\beta^{q'}$. Putting these together into matrix form, we see $\mathbb{E}[\hat{\beta}^\top_{\text{LS}}\hat{\beta}_{\text{LS}}|\beta] = \beta^\top\beta + \text{diag}(\sigma^2_1\|X^{1\dagger}\|^2_F, \ldots, \sigma^2_Q\|X^{Q\dagger}\|^2_F)$, and so $\mathbb{E}[\hat{\Sigma}^{\text{MM}}|\beta] = D^{-1}\beta^\top\beta$. Under the additional assumption that for each $d$, $\beta_d \overset{i.i.d.}{\sim} \mathcal{N}(0, \Sigma)$, we have that $\mathbb{E}[D^{-1}\beta^\top\beta] = \Sigma$, and (1) obtains from the law of iterated expectation.

We next prove statement (2), that $\hat{\beta}_{\text{ECov}}^{\text{MM}} := \mathbb{E}[\beta | \mathcal{D}, \hat{\Sigma}^{\text{MM}}] = \hat{\beta}_{\text{LS}} - \sigma^2 D \hat{\beta}_{\text{LS}}^{\dagger\top}$. Consider the singular value decomposition (SVD), $\hat{\beta}_{\text{LS}} = V \text{diag}(\lambda^{\frac{1}{2}}) U^\top$. Under Condition 4.1 substituting this expression into $\hat{\Sigma}^{\text{MM}}$ provides $\hat{\Sigma}^{\text{MM}} = D^{-1} U \text{diag}(\lambda) U^\top - \sigma^2 I_Q$. Therefore, Lemma C.2 provides that we may write

$$\hat{\beta}_{\text{ECov}}^{\text{MM}} := \mathbb{E}[\beta | \mathcal{D}, \hat{\Sigma}^{\text{MM}}]$$
$$= \hat{\beta}_{\text{LS}} - \hat{\beta}_{\text{LS}} \left[ \sigma^{-2} \hat{\Sigma}^{\text{MM}} + I_Q \right]^{-1}$$
$$= \hat{\beta}_{\text{LS}} - V \text{diag}(\lambda^{\frac{1}{2}}) U^\top \left[ \sigma^{-2}(D^{-1} U \text{diag}(\lambda) U^\top - \sigma^2 I_Q) + I_Q \right]^{-1} U^\top$$
$$= \hat{\beta}_{\text{LS}} - V \text{diag} \left[ \lambda^{\frac{1}{2}} \odot (\sigma^{-2} D^{-1} \lambda)^{-1} \right] U^\top$$
$$= \hat{\beta}_{\text{LS}} - \sigma^2 D V \text{diag}(\lambda^{-\frac{1}{2}}) U^\top$$
$$= \hat{\beta}_{\text{LS}} - \sigma^2 D \hat{\beta}_{\text{LS}}^{\dagger\top},$$

where $\odot$ is the Hadamard (i.e. elementwise) product, as desired.

We lastly prove (3), that the analogous moment based estimator constructed under the assumption of a priori exchangeability among groups is $\hat{\beta}_{\text{EGroup}}^{\text{MM}} = \hat{\beta}_{\text{LS}} - \sigma^2 Q \hat{\beta}_{\text{LS}}^{\dagger\top}$. We begin by making explicit the assumed model and estimate. Specifically we assume each $\beta^q \overset{i.i.d.}{\sim} \mathcal{N}(0, \Gamma)$ a priori, where $\Gamma$ is a $D \times D$ covariance matrix.

In this case, we obtain an unbiased moment based estimate of $\Gamma$ as $\hat{\Gamma}^{\text{MM}} := Q^{-1} \hat{\beta}_{\text{LS}} \hat{\beta}_{\text{LS}}^\top - Q^{-1} \sum_{q=1}^Q \sigma_q^2 (X^{q\top} X^q)^{-1}$. Following an argument exactly parallel to the one in the proof of (1), we find that under the prior $\beta^q \overset{i.i.d.}{\sim} \mathcal{N}(0, \Gamma)$, we have $\mathbb{E}[\hat{\Gamma}^{\text{MM}}] = \Gamma$. Furthermore, following an argument exactly parallel to the one in the proof of (2), we find that under Condition 4.1 the corresponding empirical Bayes estimate $\hat{\beta}_{\text{EGroup}}^{\text{MM}} := \mathbb{E}[\beta | \hat{\Gamma}^{\text{MM}}] = \hat{\beta}_{\text{LS}} - \sigma^2 Q \hat{\beta}_{\text{LS}}^{\dagger\top}$. We omit full details to spare repetition. $\square$

**Lemma C.2.** *Under Condition 4.1* $\mathbb{E}[\beta | \mathcal{D}, \Sigma] = \hat{\beta}_{\text{LS}} - \hat{\beta}_{\text{LS}} \left[ \sigma^{-2} \Sigma + I_Q \right]^{-1}$.

*Proof.* By Proposition 3.1, we have

$$\mathbb{E}[\vec{\beta} | \mathcal{D}, \Sigma] = V \left[ \frac{Y^{1\top} X^1}{\sigma_1^2}, \dots, \frac{Y^{Q\top} X^Q}{\sigma_Q^2} \right] \quad \text{where} \quad V^{-1} = \Sigma^{-1} \otimes I_D + \text{diag}(\frac{X^{1\top} X^1}{\sigma_1^2}, \dots, \frac{X^{Q\top} X^Q}{\sigma_Q^2}).$$

Under Condition 4.1, we can simplify this as

$$\mathbb{E}[\vec{\beta} | \mathcal{D}, \Sigma] = \left[ \Sigma^{-1} \otimes I_D + \text{diag}(\frac{X^{1\top} X^1}{\sigma_1^2}, \dots, \frac{X^{Q\top} X^Q}{\sigma_Q^2}) \right]^{-1} \left[ \frac{Y^{1\top} X^1}{\sigma_1^2}, \dots, \frac{Y^{Q\top} X^Q}{\sigma_Q^2} \right]$$
$$= \left[ \Sigma^{-1} \otimes I_D + \sigma^{-2} I_{DQ} \right]^{-1} \sigma^{-2} \left[ \hat{\beta}_{\text{LS}}^1, \dots, \hat{\beta}_{\text{LS}}^Q \right]$$
$$= \left[ \sigma^2 \Sigma^{-1} \otimes I_D + I_{DQ} \right]^{-1} \left[ \hat{\beta}_{\text{LS}}^1, \dots, \hat{\beta}_{\text{LS}}^Q \right].$$

As a result, for each $d$, $\mathbb{E}[\beta_d | \mathcal{D}, \Sigma] = \left[ \sigma^2 \Sigma^{-1} + I_Q \right]^{-1} \beta_{\text{LS},d}$ and so, in matrix form, we may write

$$\mathbb{E}[\beta | \mathcal{D}, \Sigma] = \hat{\beta}_{\text{LS}} \left[ \sigma^2 \Sigma^{-1} + I_Q \right]^{-1}$$
$$= \hat{\beta}_{\text{LS}} - \hat{\beta}_{\text{LS}} \left[ I_Q + \sigma^{-2} \Sigma \right]^{-1}.$$

$\square$

## C.3  Proof of Lemma 4.2

*Proof.* We prove the lemma in two parts; first for the case that $D > Q + 1$, and then for the case that $Q \le D \le Q + 1$.

Our proof for the case that $D > Q + 1$ relies on an expression for the squared error risk for estimators of the form $\hat{\beta} = \hat{\beta}_{LS} - \sigma^2 c \hat{\beta}_{LS}^{\dagger\top}$ for real $c$. In particular, Lemma C.3 provides that when $D > Q + 1$ and under Condition 4.1,

$$\mathbb{E}[\|\beta - (\hat{\beta}_{LS} - c\hat{\beta}_{LS}^{\dagger\top})\|_F^2 \mid \beta] = DQ + \sigma^4 c(c + 2 + 2Q - 2D)\mathbb{E}[\|\hat{\beta}_{LS}^\dagger\|_F^2 \mid \beta].$$

Notably, since under Condition 4.1, by Proposition C.1 we have that $\hat{\beta}_{ECov}^{MM} = \hat{\beta}_{LS} - \sigma^2 D \hat{\beta}_{LS}^{\dagger\top}$ we obtain $\mathbb{E}[\|\beta - \hat{\beta}_{ECov}^{MM}\|_F^2 \mid \beta] = \sigma^2 DQ - \sigma^4 D(D - 2Q - 2)\mathbb{E}[\|\hat{\beta}_{LS}^\dagger\|_F^2 \mid \beta]$, as desired.

We next consider $Q \le D \le Q + 1$. In this case, both $R(\beta, \hat{\beta}_{ECov}^{MM})$ and $\sigma^2 DQ - \sigma^4 D(D - 2Q - 2)\mathbb{E}[\|\hat{\beta}_{LS}^\dagger\|_F^2 \mid \beta]$ are positive infinity. In particular, observe that $\|\hat{\beta}_{LS}^\dagger\|_F^2 = \mathrm{tr}[(\hat{\beta}_{LS}^\top \hat{\beta}_{LS})^{-1}]$ is the trace of the inverse of a non-central Wishart matrix, which is known to have infinite expectation for $Q \le D \le Q + 1$ (see e.g. Hillier and Kan [30]). Likewise, Lemma C.6 reveals that $R(\beta, \hat{\beta}_{ECov}^{MM}) = \infty$ as well.

The second assertion of Lemma 4.2, that when $D \le Q$ and under Condition 4.1 $\mathbb{E}[\|\beta - \hat{\beta}_{EGroup}^{MM}\|_F^2 \mid \beta] = \sigma^2 DQ - \sigma^4 Q(Q - 2D - 2)\mathbb{E}[\|\hat{\beta}_{LS}^\dagger\|_F^2 \mid \beta]$, obtains similarly. Specifically, under these conditions an identical argument to that provided in Lemma C.3 provides that

$$\mathbb{E}[\|\beta - (\hat{\beta}_{LS} - \sigma^2 c\hat{\beta}_{LS}^{\dagger\top})\|_F^2 \mid \beta] = DQ + \sigma^4 c(c + 2 + 2D - 2Q)\mathbb{E}[\|\hat{\beta}_{LS}^\dagger\|_F^2 \mid \beta]$$

when $D < Q - 1$. The desired expression is then obtained by taking $c = Q$ to reflect $\hat{\beta}_{EGroup}^{MM} = \hat{\beta}_{LS} - \sigma^2 Q\hat{\beta}_{LS}^{\dagger\top}$, again as specified by Proposition C.1. $\qquad\square$

**Lemma C.3.** *Let $D > Q + 1$ and let $\hat{\beta} = \hat{\beta}_{LS} - \sigma^2 c\hat{\beta}_{LS}^{\dagger\top}$. Then under Condition 4.1 $\mathbb{E}[\|\beta - \hat{\beta}\|_F^2 \mid \beta] = \sigma^2 DQ + \sigma^4 c(c + 2 + 2Q - 2D)\mathbb{E}[\|\hat{\beta}_{LS}^\dagger\|_F^2 \mid \beta]$.*

*Proof.* The results follows by considering Stein's unbiased risk estimate (SURE) [41, Chapter 4, Corollary 7.2] (restated as Lemma C.4) and making several algebraic simplifications. In order to apply the lemma, we note that under Condition 4.1 $\vec{\hat{\beta}}_{LS} \sim \mathcal{N}(\vec{\beta}, \sigma^2 I_{DQ})$ and $\vec{\hat{\beta}} = \vec{\hat{\beta}}_{LS} - g(\vec{\hat{\beta}}_{LS})$ for $g(\vec{\hat{\beta}}_{LS}) = -\sigma^2 c \cdot \mathrm{vec}(\hat{\beta}_{LS}^{\dagger\top})$, where $\mathrm{vec}(\cdot)$ represents the operation of reshaping an $D \times Q$ matrix into a $DQ$-vector by stacking its columns.

We first simplify the sum of partial derivatives in Equation (4) of Lemma C.4. Observe that

$$\sum_{n=1}^{DQ} \frac{\partial g_n(\vec{\hat{\beta}}_{LS})}{\partial \vec{\hat{\beta}}_{LS,n}} = -\sigma^2 c \sum_{d=1}^{D} \sum_{q=1}^{Q} \frac{\partial \hat{\beta}_{LS,d}^{\dagger,q}}{\partial \hat{\beta}_{LS,d}^q},$$

where $\hat{\beta}_{LS,d}^{\dagger,q}$ denotes the entry in the $q$th row and $d$th column of $\hat{\beta}_{LS}^\dagger$.

Next, letting $e_q$ be the $q$th basis vector in $\mathbb{R}^Q$, for each $q$ and $d$ we may write

$$\frac{\partial \hat{\beta}_{LS,d}^{\dagger,q}}{\partial \hat{\beta}_{LS,d}^q} = \frac{\partial}{\partial \hat{\beta}_{LS,d}^q} \hat{\beta}_{LS,d}(\hat{\beta}_{LS}^\top \hat{\beta}_{LS})^{-1} e_q$$

$$= e_q^\top (\hat{\beta}_{LS}^\top \hat{\beta}_{LS})^{-1} e_q + \hat{\beta}_{LS,d} \frac{\partial}{\partial \hat{\beta}_{LS,d}^q} (\hat{\beta}_{LS}^\top \hat{\beta}_{LS})^{-1} e_q$$

$$= e_q^\top (\hat{\beta}_{LS}^\top \hat{\beta}_{LS})^{-1} e_q - \hat{\beta}_{LS,d}^\top (\hat{\beta}_{LS}^\top \hat{\beta}_{LS})^{-1} \left[ \frac{\partial}{\partial \hat{\beta}_{LS,d}^q} (\hat{\beta}_{LS}^\top \hat{\beta}_{LS}) \right] (\hat{\beta}_{LS}^\top \hat{\beta}_{LS})^{-1} e_q$$

$$= \|\hat{\beta}_{LS}^{\dagger,q}\|^2 - \hat{\beta}_{LS,d}^{\dagger\top} \left[ e_q \hat{\beta}_{LS,d}^\top + \hat{\beta}_{LS,d} e_q^\top \right] (\hat{\beta}_{LS}^\top \hat{\beta}_{LS})^{-1} e_q$$

$$= \|\hat{\beta}_{LS}^{\dagger,q}\|^2 - \left[ \hat{\beta}_{LS,d}^{\dagger\top} e_q \hat{\beta}_{LS,d}^\top (\hat{\beta}_{LS}^\top \hat{\beta}_{LS})^{-1} e_q + \hat{\beta}_{LS,d}^{\dagger\top} \hat{\beta}_{LS,d} e_q^\top (\hat{\beta}_{LS}^\top \hat{\beta}_{LS})^{-1} e_q \right]$$

$$= \|\hat{\beta}_{LS}^{\dagger,q}\|^2 - (\hat{\beta}_{LS,d}^{\dagger,q})^2 - \hat{\beta}_{LS,d}^{\dagger\top} \hat{\beta}_{LS,d} \|\hat{\beta}_{LS}^{\dagger,q}\|^2,$$

where in the fourth and last lines we have used that $e_q^\top (\hat\beta_{\text{LS}}^\top \hat\beta_{\text{LS}})^{-1} e_q = \|\hat\beta_{\text{LS}}^{\dagger,q}\|^2$, as can be seen by observing that $(\hat\beta_{\text{LS}}^\top \hat\beta_{\text{LS}})^{-1} = \hat\beta_{\text{LS}}^\dagger \hat\beta_{\text{LS}}^{\dagger\top}$.

Adding these terms together we find

$$
\sum_{d=1}^{D} \sum_{q=1}^{Q} \frac{\partial \hat\beta_{\text{LS},d}^{\dagger,q}}{\partial \hat\beta_{\text{LS},d}^q} = \sum_{d=1}^{D} \sum_{q=1}^{Q} \left\{ \|\hat\beta_{\text{LS}}^{\dagger,q}\|^2 - (\hat\beta_{\text{LS},d}^{\dagger,q})^2 - \hat\beta_{\text{LS},d}^{\dagger\top}\hat\beta_{\text{LS},d}\|\hat\beta_{\text{LS}}^{\dagger,q}\|^2 \right\}
$$

$$
= D\|\hat\beta_{\text{LS}}^\dagger\|_F^2 - \|\hat\beta_{\text{LS}}^\dagger\|_F^2 - \|\hat\beta_{\text{LS}}^\dagger\|_F^2 \sum_{d=1}^{D} \hat\beta_{\text{LS},d}^{\dagger\top}\hat\beta_{\text{LS},d}
$$

$$
= D\|\hat\beta_{\text{LS}}^\dagger\|_F^2 - \|\hat\beta_{\text{LS}}^\dagger\|_F^2 - \|\hat\beta_{\text{LS}}^\dagger\|_F^2 \text{tr}(\hat\beta_{\text{LS}}^\dagger \hat\beta_{\text{LS}})
$$

$$
= (D - Q - 1)\|\hat\beta_{\text{LS}}^\dagger\|_F^2.
$$

We next note that the regularity condition required by Lemma C.4 is satisfied, as demonstrated in Lemma C.5, and so we may write

$$
\mathbb{E}[\|\beta - \hat\beta\|_F^2 \mid \beta] = \sigma^2 DQ + \mathbb{E}[\|g(\hat{\vec\beta}_{\text{LS}})\|^2 \mid \beta] - 2\sigma^2 \sum_{d=1}^{D} \sum_{q=1}^{Q} \mathbb{E}[\frac{\partial \hat\beta_{\text{LS},d}^{\dagger,q}}{\partial \hat\beta_{\text{LS},d}^q} \mid \beta]
$$

$$
= \sigma^2 DQ + \sigma^4 c^2 \mathbb{E}[\|\hat\beta_{\text{LS}}^\dagger\|^2 \mid \beta] - 2\sigma^4 c(D - Q - 1)\mathbb{E}[\|\hat\beta_{\text{LS}}^\dagger\|_F^2 \mid \beta]
$$

$$
= \sigma^2 DQ + \sigma^4 c(c + 2 + 2Q - 2D)\mathbb{E}[\|\hat\beta_{\text{LS}}^\dagger\|^2 \mid \beta].
$$

as desired. $\square$

**Lemma C.4** (Stein's Unbiased Risk Estimate – Lehmann and Casella Corollary 7.2). *Let $X \sim \mathcal{N}(\theta, \sigma^2 I_N)$, and let the estimator $\hat\theta$ be of the form $\hat\theta = X - g(X)$ where $g(X) = [g_1(X), g_2(X), \ldots, g_N(X)]$ is differentiable. If $\mathbb{E}[|\frac{\partial}{\partial X_n} g_n(X)|] < \infty$ for each $n = 1, \ldots, N$, then*

$$
\text{R}(\theta, \hat\theta) = \sigma^2 N + \mathbb{E}[\|g(X)\|^2] - 2\sigma^2 \sum_{n=1}^{N} \frac{\partial}{\partial X_n} g_n(X). \tag{4}
$$

**Lemma C.5.** *Let $D > Q + 1$. Then under Condition 4.1 $\mathbb{E}\left[\left|\frac{\partial \hat\beta_{\text{LS},d}^{\dagger,q}}{\partial \hat\beta_{\text{LS},d}^q}\right| \mid \beta\right] \le \infty$ for each $d$ and $q$.*

*Proof.* From our derivation of $\frac{\partial \hat\beta_{\text{LS},d}^{\dagger,q}}{\partial \hat\beta_{\text{LS},d}^q}$ in Lemma C.3 we have that

$$
\frac{\partial \hat\beta_{\text{LS},d}^{\dagger,q}}{\partial \hat\beta_{\text{LS},d}^q} = \|\hat\beta_{\text{LS}}^{\dagger,q}\|^2 - (\hat\beta_{\text{LS},d}^{\dagger,q})^2 - \hat\beta_{\text{LS},d}^{\dagger\top}\hat\beta_{\text{LS},d}\|\hat\beta_{\text{LS}}^{\dagger,q}\|^2
$$

$$
= \|\hat\beta_{\text{LS}}^{\dagger,q}\|^2 - (\hat\beta_{\text{LS},d}^{\dagger,q})^2 - \|\hat\beta_{\text{LS}}^{\dagger,q}\|^2 \text{tr}[(\hat\beta_{\text{LS}}^\top \hat\beta_{\text{LS}})^{-1}\beta_{\text{LS},d}\beta_{\text{LS},d}^\top].
$$

As such we have that

$$
\left|\frac{\partial \hat\beta_{\text{LS},d}^{\dagger,q}}{\partial \hat\beta_{\text{LS},d}^q}\right| \le \|\hat\beta_{\text{LS}}^{\dagger,q}\|^2 + |(\beta_{\text{LS},d}^{\dagger,q})^2| + \|\hat\beta_{\text{LS}}^{\dagger,q}\|^2 |\text{tr}[(\hat\beta_{\text{LS}}^\top \hat\beta_{\text{LS}})^{-1}\beta_{\text{LS},d}\beta_{\text{LS},d}^\top]|
$$

$$
\le \|\hat\beta_{\text{LS}}^{\dagger,q}\|^2 + \left|\sum_{d'=1}^{D}(\beta_{\text{LS},d'}^{\dagger,q})^2\right| + \|\hat\beta_{\text{LS}}^{\dagger,q}\|^2 \left|\text{tr}[(\hat\beta_{\text{LS}}^\top \hat\beta_{\text{LS}})^{-1}\sum_{d'=1}^{D}\beta_{\text{LS},d'}\beta_{\text{LS},d'}^\top]\right|
$$

$$
= \|\hat\beta_{\text{LS}}^{\dagger,q}\|^2 + \|\hat\beta_{\text{LS}}^{\dagger,q}\|^2 + \|\hat\beta_{\text{LS}}^{\dagger,q}\|^2 \text{tr}[(\hat\beta_{\text{LS}}^\top \hat\beta_{\text{LS}})^{-1}\hat\beta_{\text{LS}}^\top \hat\beta_{\text{LS}}]
$$

$$
\le (2 + Q)\|\hat\beta_{\text{LS}}^{\dagger,q}\|^2
$$

$$
\le (2 + Q)\|\hat\beta_{\text{LS}}^\dagger\|_F^2
$$

$$
= (2 + Q)\text{tr}[(\hat\beta_{\text{LS}}^\top \hat\beta_{\text{LS}})^{-1}].
$$

We next recognize that under Condition 4.1, $(\hat{\beta}_{\mathrm{LS}}^\top \hat{\beta}_{\mathrm{LS}})^{-1}$ is the inverse of a non-central Wishart matrix with non-centrality parameter $\beta$. Therefore, from Hillier and Kan [30, Theorem 1], we have that for $D > Q + 1$, $\mathbb{E}\left[\mathrm{tr}\left((\hat{\beta}_{\mathrm{LS}}^\top \hat{\beta}_{\mathrm{LS}})^{-1}\right) \mid \beta\right] < \infty$. Accordingly, we may conclude that

$$\mathbb{E}\left[\left|\frac{\partial \hat{\beta}_{\mathrm{LS},d}^{\dagger;q}}{\partial \hat{\beta}_{\mathrm{LS},d}^q}\right| \mid \beta\right] \leq \infty \text{ as desired.} \qquad \square$$

**Lemma C.6.** *Assume $Q \leq D \leq Q + 1$. For any $\beta$, $\mathrm{R}(\beta, \hat{\beta}_{\mathrm{ECov}}^{\mathrm{MM}}) = \infty$.*

*Proof.* First observe that we may lower bound $L(\beta, \hat{\beta}_{\mathrm{ECov}}^{\mathrm{MM}})$ as

$$
\begin{aligned}
L(\beta, \hat{\beta}_{\mathrm{ECov}}^{\mathrm{MM}}) &= \|\hat{\beta}_{\mathrm{ECov}}^{\mathrm{MM}} - \beta\|_F^2 \\
&= \|\sigma^2 D \hat{\beta}_{\mathrm{LS}}^{\dagger\top} + \beta - \hat{\beta}_{\mathrm{LS}}\|_F^2 \\
&= \sigma^4 D^2 \|\hat{\beta}_{\mathrm{LS}}^\dagger\|_F^2 + \|\beta - \hat{\beta}_{\mathrm{LS}}\|_F^2 - 2\sigma^2 D \mathrm{tr}\left[-\hat{\beta}_{\mathrm{LS}}^\dagger(\beta - \hat{\beta}_{\mathrm{LS}})\right] \\
&\geq \sigma^4 D^2 \|\hat{\beta}_{\mathrm{LS}}^\dagger\|_F^2 + \|\beta - \hat{\beta}_{\mathrm{LS}}\|_F^2 - 2\sigma^2 D \|\hat{\beta}_{\mathrm{LS}}^\dagger\|_F \|\beta - \hat{\beta}_{\mathrm{LS}}\|_F \\
&= (\sigma^2 D \|\hat{\beta}_{\mathrm{LS}}^\dagger\|_F - \|\beta - \hat{\beta}_{\mathrm{LS}}\|_F)^2
\end{aligned}
$$

where the inequality follows from Cauchy-Schwarz. We next consider any constant $c < \sigma^2 D$ and write

$$
\begin{aligned}
\mathrm{R}(\beta, \hat{\beta}_{\mathrm{ECov}}^{\mathrm{MM}}) &= \mathbb{E}[L(\beta, \hat{\beta}_{\mathrm{ECov}}^{\mathrm{MM}})|\beta] \\
&= \mathbb{P}(c\|\hat{\beta}_{\mathrm{LS}}^\dagger\|_F \geq \|\hat{\beta}_{\mathrm{LS}} - \beta\|_F)\mathbb{E}[L(\beta, \hat{\beta}_{\mathrm{EGroup}}^{\mathrm{MM}}) \mid \beta, c\|\hat{\beta}_{\mathrm{LS}}^\dagger\|_F \geq \|\hat{\beta}_{\mathrm{LS}} - \beta\|_F] \\
&\quad + \mathbb{P}(c\|\hat{\beta}_{\mathrm{LS}}^\dagger\|_F < \|\hat{\beta}_{\mathrm{LS}} - \beta\|_F)\mathbb{E}[L(\beta, \hat{\beta}_{\mathrm{EGroup}}^{\mathrm{MM}}) \mid \beta, c\|\hat{\beta}_{\mathrm{LS}}^\dagger\|_F < \|\hat{\beta}_{\mathrm{LS}} - \beta\|_F] \\
&\geq \mathbb{P}(c\|\hat{\beta}_{\mathrm{LS}}^\dagger\|_F \geq \|\hat{\beta}_{\mathrm{LS}} - \beta\|_F)\mathbb{E}[L(\beta, \hat{\beta}_{\mathrm{EGroup}}^{\mathrm{MM}}) \mid \beta, c\|\hat{\beta}_{\mathrm{LS}}^\dagger\|_F \geq \|\hat{\beta}_{\mathrm{LS}} - \beta\|_F] \\
&\geq \mathbb{P}(c\|\hat{\beta}_{\mathrm{LS}}^\dagger\|_F \geq \|\hat{\beta}_{\mathrm{LS}} - \beta\|_F)\mathbb{E}[(\sigma^2 D \|\hat{\beta}_{\mathrm{LS}}^\dagger\|_F - \|\beta - \hat{\beta}_{\mathrm{LS}}\|_F)^2 \mid \beta, c\|\hat{\beta}_{\mathrm{LS}}^\dagger\|_F \geq \|\hat{\beta}_{\mathrm{LS}} - \beta\|_F] \\
&\geq \mathbb{P}(c\|\hat{\beta}_{\mathrm{LS}}^\dagger\|_F \geq \|\hat{\beta}_{\mathrm{LS}} - \beta\|_F)(\sigma^2 D - c)^2 \mathbb{E}[\|\hat{\beta}_{\mathrm{LS}}^\dagger\|_F^2 \mid \beta, c\|\hat{\beta}_{\mathrm{LS}}^\dagger\|_F \geq \|\hat{\beta}_{\mathrm{LS}} - \beta\|_F] \\
&\geq (\sigma^2 D - c)^2 \mathbb{P}(c\|\hat{\beta}_{\mathrm{LS}}^\dagger\|_F \geq \|\hat{\beta}_{\mathrm{LS}} - \beta\|_F)\mathbb{E}[\mathrm{tr}[(\hat{\beta}_{\mathrm{LS}}^\top \hat{\beta}_{\mathrm{LS}})^{-1}] \mid \beta] = \infty
\end{aligned}
$$

where the last line comes from recognizing $(\hat{\beta}_{\mathrm{LS}}^\top \hat{\beta}_{\mathrm{LS}})^{-1}$ as the inverse of a non-central Wishart matrix, the trace of which has infinite expectation for $Q \leq D \leq Q + 1$. $\qquad \square$

### C.4 Proof of Theorem 4.3 and additional details

*Proof.* The first domination result of Theorem 4.3 follows closely from Lemma 4.2. Under Condition 4.1, $\hat{\beta}_{\mathrm{LS}} \overset{d}{=} \beta + \sigma\epsilon$ for a random matrix $\epsilon$ with i.i.d. standard normal entries, and so we can see $\mathrm{R}(\beta, \hat{\beta}_{\mathrm{LS}}) = \sum_{d=1}^D \sum_{q=1}^Q \mathbb{E}[(\sigma\epsilon_d^q)^2] = DQ\sigma^2$. Next, $D > 2Q + 2$ implies that $D - 2 - 2Q > 0$ so that $D(D - 2 - 2Q)\sigma^2\|\hat{\beta}_{\mathrm{LS}}^\dagger\|_F^2$ is almost surely positive, and therefore positive in expectation. We therefore obtain the result from Lemma 4.2.

We next consider the second domination result. The performance of $\hat{\beta}_{\mathrm{EGroup}}^{\mathrm{MM}}$ may be seen to degrade in stages as we transition from a few covariates and many groups regime to a many covariates and few groups regime. When $D < Q/2 - 1$, we can see that $\hat{\beta}_{\mathrm{EGroup}}^{\mathrm{MM}}$ has good performance. In fact, by an argument analogous to our proof of the first part of Theorem 4.3 above, we can see that $\hat{\beta}_{\mathrm{EGroup}}^{\mathrm{MM}}$ dominates $\hat{\beta}_{\mathrm{LS}}$; Specifically, from Lemma 4.2 we can recognize $\mathrm{R}(\beta, \hat{\beta}_{\mathrm{LS}}) - \mathrm{R}(\beta, \hat{\beta}_{\mathrm{EGroup}})$ as the expectation of an almost surely positive quantity.

When $D = Q/2 - 1$ we have $Q(Q - 2 - 2D) = 0$, and so regardless of $\beta$, the estimators $\hat{\beta}_{\mathrm{EGroup}}^{\mathrm{MM}}$ and $\hat{\beta}_{\mathrm{LS}}$ have equal risk, and neither dominates.

Relative performance degrades further in the intermediate regime of $Q/2 - 1 < D < Q - 1$. In this regime, $\mathrm{R}(\beta, \hat{\beta}_{\mathrm{LS}}) - \mathrm{R}(\beta, \hat{\beta}_{\mathrm{EGroup}}^{\mathrm{MM}}) = \sigma^4 Q(Q - 2 - 2D)\mathbb{E}[\|\hat{\beta}_{\mathrm{LS}}^\dagger\|_F^2 \mid \beta]$ may be written as the expectation of an almost surely negative quantity, and so $\hat{\beta}_{\mathrm{EGroup}}^{\mathrm{MM}}$ is dominated by $\hat{\beta}_{\mathrm{LS}}$.

The situation is even worse when $Q - 1 \leq D \leq Q$; appealing again the they symmetry between $\hat{\beta}^{\mathrm{MM}}_{\mathrm{EGroup}}$ and $\hat{\beta}^{\mathrm{MM}}_{\mathrm{ECov}}$, we can see that by Lemma C.6 $\mathrm{R}(\beta, \hat{\beta}^{\mathrm{MM}}_{\mathrm{EGroup}}) = \infty$.

Finally, when $D > Q$ the expression $\hat{\beta}^{\mathrm{MM}}_{\mathrm{EGroup}} = \hat{\beta}_{\mathrm{LS}} - \left[\sigma^{-2}\hat{\Gamma}^{\mathrm{MM}} - I_D\right]^{-1}\hat{\beta}_{\mathrm{LS}}$ involves the inverse of a low rank matrix since under Condition 4.1, $\hat{\Gamma}^{\mathrm{MM}} = Q^{-1}\hat{\beta}_{\mathrm{LS}}\hat{\beta}_{\mathrm{LS}}^{\top} - \sigma^2 I_D$. Accordingly we take as our convention $\|\hat{\beta}^{\mathrm{MM}}_{\mathrm{EGroup}}\| = \infty$, analogously to defining $\frac{1}{0} = \infty$; as a result $\hat{\beta}^{\mathrm{MM}}_{\mathrm{EGroup}}$ has infinite risk in this second regime as well, and we see that this estimator is dominated by $\hat{\beta}_{\mathrm{LS}}$ whenever $D < Q/2 - 1$.

$\square$

With the strong parallels established by Proposition C.1 and Lemma 4.2 under Condition 4.1, we can see that this is not a result of $\hat{\beta}^{\mathrm{MM}}_{\mathrm{EGroup}}$ being singularly bad. Indeed, if we consider the many groups regime with $Q > D$, we can obtain analogous results to demonstrate the superiority of an exchangeability among groups approach.

### C.5 Proof of Lemma 4.4

*Proof.* We first show that under Condition 4.1, $\hat{\Sigma} = U\mathrm{diag}\left[(D^{-1}\lambda - \sigma^2\mathbf{1}_Q)_+\right]U^{\top}$ is the maximum marginal likelihood estimate of $\Sigma$ in Equation (1). Our approach is to first derive a lower bound on the negative log likelihood, and then show that this bound is met with equality by the proposed expression.

For convenience, we consider a scaling of the negative log likelihood,

$$-2D^{-1}\ln p(\hat{\beta}_{\mathrm{LS}}|\Sigma) = \ln|\Sigma + \sigma^2 I_Q| + D^{-1}\mathrm{tr}\left[(\Sigma + \sigma^2 I_Q)^{-1}\hat{\beta}_{\mathrm{LS}}^{\top}\hat{\beta}_{\mathrm{LS}}\right],$$

and are interested in deriving a lower bound on

$$\min_{\Sigma \succeq 0} \ln|\Sigma + \sigma^2 I_Q| + D^{-1}\mathrm{tr}\left[(\Sigma + \sigma^2 I_Q)^{-1}\hat{\beta}_{\mathrm{LS}}^{\top}\hat{\beta}_{\mathrm{LS}}\right],$$

where the notation $\Sigma \succeq 0$ reflects that the minimum is taken over the space of positive semidefinite matrices.

The problem simplifies if we parameterize the minimization with the eigendecomposition $\Sigma = V^{\top}\mathrm{diag}(\nu)V$, where $V$ is a $Q \times Q$ matrix satisfying $V^{\top}V = I_Q$ and $\nu$ is a $Q$-vector of non-negative reals. In particular, if we define $\mathcal{L}(V, \nu) := -2D^{-1}\ln p(\hat{\beta}_{\mathrm{LS}}|\Sigma = V^{\top}\mathrm{diag}(\nu)V)$ then, leaving the constraints on $V$ and $\nu$ implicit, we have

$$\min_{V,\nu}\mathcal{L}(V, \nu) = \min_{V,\nu}\ln|V^{\top}\mathrm{diag}(\nu)V + \sigma^2 I_Q| + D^{-1}\mathrm{tr}\left[(V^{\top}\mathrm{diag}(\nu)V + \sigma^2 I_Q)^{-1}\hat{\beta}_{\mathrm{LS}}^{\top}\hat{\beta}_{\mathrm{LS}}\right]$$

$$= \min_{V,\nu}\ln|V^{\top}\mathrm{diag}(\nu)V + \sigma^2 I_Q| + D^{-1}\mathrm{tr}\left[(\mathrm{diag}(\nu) + \sigma^2 I_Q)^{-1}V\hat{\beta}_{\mathrm{LS}}^{\top}\hat{\beta}_{\mathrm{LS}}V^{\top}\right]$$

$$= \min_{V,\nu}\sum_{q=1}^{Q}\ln(\nu_q + \sigma^2) + D^{-1}\sum_{q=1}^{Q}\frac{1}{\nu_q + \sigma^2}V_q^{\top}\hat{\beta}_{\mathrm{LS}}^{\top}\hat{\beta}_{\mathrm{LS}}V_q$$

$$= \min_{V}\sum_{q=1}^{Q}\min_{\nu_q \geq 0}\ln(\nu_q + \sigma^2) + \frac{D^{-1}V_q^{\top}\hat{\beta}_{\mathrm{LS}}^{\top}\hat{\beta}_{\mathrm{LS}}V_q}{\nu_q + \sigma^2}.$$

Next, Lemma C.7 provides that we may solve the inner optimization problems over $\nu$ in the line above analytically to get $\nu^* := \arg\min_{\nu}\mathcal{L}(V, \nu)$ with entries $\nu_q^* = \max(\sigma^2, D^{-1}V_q^{\top}\hat{\beta}_{\mathrm{LS}}^{\top}\hat{\beta}_{\mathrm{LS}}V_q) - \sigma^2$. Substituting these values in, we obtain

$$\min_{V,\nu}\mathcal{L}(V, \nu) = \min_{V}\sum_{q=1}^{Q}\ln\left[\max(\sigma^2, D^{-1}V_q^{\top}\hat{\beta}_{\mathrm{LS}}^{\top}\hat{\beta}_{\mathrm{LS}}V_q)\right] + \frac{D^{-1}V_q^{\top}\hat{\beta}_{\mathrm{LS}}^{\top}\hat{\beta}_{\mathrm{LS}}V_q}{\max(\sigma^2, D^{-1}V_q^{\top}\hat{\beta}_{\mathrm{LS}}^{\top}\hat{\beta}_{\mathrm{LS}}V_q)}$$

$$= \min_{V}\sum_{q=1}^{Q}\ln\left[\max(\sigma^2, D^{-1}V_q^{\top}\hat{\beta}_{\mathrm{LS}}^{\top}\hat{\beta}_{\mathrm{LS}}V_q)\right] + \sigma^{-2}\min(\sigma^2, D^{-1}V_q^{\top}\hat{\beta}_{\mathrm{LS}}^{\top}\hat{\beta}_{\mathrm{LS}}V_q).$$

We can now further simplify the problem by considering the eigendecomposition of $\hat{\beta}_{\mathrm{LS}}^\top \hat{\beta}_{\mathrm{LS}} = U\mathrm{diag}(\lambda)U^\top$, and recognizing that because $VU$ satisfies $(VU)^\top VU = I_Q$ we may write

$$\min_{V,\nu}\mathcal{L}(V,\nu) = \min_V \sum_{q=1}^Q \ln\left[\max(\sigma^2, D^{-1}V_q^\top\hat{\beta}_{\mathrm{LS}}^\top\hat{\beta}_{\mathrm{LS}}V_q)\right] + \sigma^{-2}\min(\sigma^2, D^{-1}V_q^\top\hat{\beta}_{\mathrm{LS}}^\top\hat{\beta}_{\mathrm{LS}}V_q)$$

$$= \min_V \sum_{q=1}^Q \ln\left[\max\left(\sigma^2, V_q^\top\mathrm{diag}(D^{-1}\lambda)V_q\right)\right] + \sigma^{-2}\min\left[\sigma^2, V_q^\top\mathrm{diag}(D^{-1}\lambda)V_q\right].$$

Finally, we obtain a lower bound by recognizing $\{V_q^\top\mathrm{diag}(D^{-1}\lambda)V_q\}_{q=1}^Q$ as the diagonals of $D^{-1}V\mathrm{diag}(\lambda)V^\top$ and applying Lemma C.8 to obtain that

$$-2D^{-1}\ln p(\hat{\beta}_{\mathrm{LS}}|\Sigma) \geq \sum_{q=1}^Q \ln\left[\max(\sigma^2, D^{-1}\lambda_q)\right] + \sigma^{-2}\min(\sigma^2, D^{-1}\lambda_q)$$

for every $\Sigma \succeq 0$.

We next show that this bound is met with equality by $\hat{\Sigma} = U\mathrm{diag}\left[(D^{-1}\lambda - \sigma^2\mathbf{1}_Q)_+\right]U^\top$, the form given in the statement of Lemma 4.4. Recognize first that $\hat{\Sigma} + \sigma^2 I_Q = U\mathrm{diag}\left[\max(\sigma^2\mathbf{1}_Q, D^{-1}\lambda)\right]U^\top$. Substituting this expression in, we find

$$-2D^{-1}\ln p(\hat{\beta}_{\mathrm{LS}}|\hat{\Sigma}) = \ln|\hat{\Sigma} + \sigma^2 I_Q| + D^{-1}\mathrm{tr}\left[(\hat{\Sigma} + \sigma^2 I_Q)^{-1}\hat{\beta}_{\mathrm{LS}}^\top\hat{\beta}_{\mathrm{LS}}\right]$$

$$= \ln\left|\mathrm{diag}\left[\max(\sigma^2\mathbf{1}_Q, D^{-1}\lambda)\right]\right| + D^{-1}\mathrm{tr}\left[\mathrm{diag}\left[\max(\sigma^2\mathbf{1}_Q, D^{-1}\lambda)\right]^{-1}U^\top\hat{\beta}_{\mathrm{LS}}^\top\hat{\beta}_{\mathrm{LS}}U\right]$$

$$= \sum_{q=1}^Q \ln\left[\max(\sigma^2, D^{-1}\lambda_q)\right] + D^{-1}\lambda_q/\max(\sigma^2, D^{-1}\lambda_q)$$

$$= \sum_{q=1}^Q \ln\left[\max(\sigma^2, D^{-1}\lambda_q)\right] + \sigma^{-2}\min(\sigma^2, D^{-1}\lambda_q),$$

which meets our lower bound. This establishes that the maximum marginal likelihood estimate is $\hat{\Sigma} = U\left[(D^{-1}\lambda - \sigma^2\mathbf{1}_Q)_+\right]U^\top$, as desired.

It now remains to show that, under Condition 4.1, $\hat{\beta}_{\mathrm{ECov}} = V\mathrm{diag}\left[\lambda^{\frac{1}{2}} \odot (\mathbf{1}_Q - \sigma^2 D\lambda^{-1})_+\right]U^\top$. By Lemma C.2, we have that $\hat{\beta}_{\mathrm{ECov}} = \hat{\beta}_{\mathrm{LS}} - \hat{\beta}_{\mathrm{LS}}\left[I_Q + \sigma^{-2}\hat{\Sigma}\right]^{-1}$. Substituting in the analytic expression for $\hat{\Sigma}$, recalling the SVD $\hat{\beta}_{\mathrm{LS}} = V\mathrm{diag}(\lambda^{\frac{1}{2}})U^\top$, and rearranging, we obtain

$$\hat{\beta}_{\mathrm{ECov}} = V\mathrm{diag}(\lambda^{\frac{1}{2}})U^\top - V\mathrm{diag}(\lambda^{\frac{1}{2}})U^\top\left\{I_Q + \sigma^{-2}U\left[(D^{-1}\lambda - \sigma^2\mathbf{1}_Q)_+\right]U^\top\right\}^{-1}$$

$$= V\mathrm{diag}\left\{\lambda^{\frac{1}{2}} - \lambda^{\frac{1}{2}}\left[\mathbf{1}_Q + \sigma^{-2}(D^{-1}\lambda - \sigma^2\mathbf{1}_Q)_+\right]^{-1}\right\}U^\top$$

$$= V\mathrm{diag}\left\{\lambda^{\frac{1}{2}} \odot \left[\mathbf{1}_Q - \left(\mathbf{1}_Q + (\sigma^{-2}D^{-1}\lambda - \mathbf{1}_Q)_+\right)^{-1}\right]\right\}U^\top$$

$$= V\mathrm{diag}\left[\lambda^{\frac{1}{2}} \odot \left(\mathbf{1}_Q - \sigma^2 D\lambda^{-1}\right)_+\right]U^\top,$$

as desired.  □

**Lemma C.7.** *For any $c > 0$,*

$$\nu^* := \arg\min_{\nu \geq 0} \ln(\nu + \sigma^2) + \frac{c}{\nu + \sigma^2}$$

$$= \max(\sigma^2, c) - \sigma^2$$

*Proof.* Define $g(x) := \ln(x + \sigma^2) + c/(x + \sigma^2)$ and $f(x) := g(\sigma^2 x) = \ln(x + 1) + \frac{\sigma^{-2}c}{x+1} + \ln \sigma^2$ to lighten notation. Now $\nu^* = \arg\max_{x \geq 0} g(x) = \sigma^2 \arg\max_{x \geq 0} f(x)$. Denote by $f'$ and $f''$ the first two derivatives of $f$. Notably, $f'(x) = (x + 1)^{-1} \left[ 1 - \sigma^{-2}c/(x + 1) \right]$ and $f''(x) = (x + 1)^{-2} \left[ 2\sigma^{-2}c/(x + 1) - 1 \right]$. The result may be seen by separately considering the cases of $\sigma^{-2}c < 1$ and $\sigma^{-2}c \geq 1$.

If $\sigma^{-2}c < 1$, then $f'$ is positive on $\mathbb{R}_+$, and so $\arg\min_{x \in \mathbb{R}_+} f(x) = 0$. On the other hand, if $\sigma^{-2}c \geq 1$, then $f$ has a local minimum at $x = \sigma^{-2}c - 1$ (note that $f'(\sigma^{-2}c - 1) = 0$, and $f''(\sigma^{-2}c - 1) > 0$)). Since this is the only local minimum on $\mathbb{R}_+$, and with the positive second derivative at the this minimum, we can conclude that in this case $\arg\min_{x \in \mathbb{R}_+} f(x) = \sigma^{-2}c - 1$. In either case, we can write $\arg\min_{x \in \mathbb{R}_+} f(x) = \max(1, \sigma^{-2}c) - 1$. Therefore, as desired, we see that $\arg\min_{x \in \mathbb{R}_+} g(x) = \max(\sigma^2, c) - \sigma^2$. $\qquad\square$

**Lemma C.8.** *Let $A$ be a $Q \times Q$ Hermitian matrix with eigenvalues $\lambda_1, \lambda_2, \ldots, \lambda_Q$. Then*

$$\sum_{q=1}^{Q} \ln \left[ \max(\sigma^2, A_{q,q}) \right] + \sigma^{-2} \min(\sigma^2, A_{q,q}) \geq \sum_{q=1}^{Q} \ln \left[ \max(\sigma^2, \lambda_q) \right] + \sigma^{-2} \min(\sigma^2, \lambda_q).$$

*Proof.* First note that $f(x) = \ln \max(\sigma^2, x) + \min(\sigma^2, x)$ is concave on $\mathbb{R}_+$, and so the vector valued function, $g(x_1, x_2, \ldots, x_N) = \sum_{n=1}^{N} f(x_n)$ is Schur concave. By the Schur-Horn theorem (Theorem D.4) the diagonals of $A$ are majorized by its eigenvalues, when each are sorted in descending order. As such $g\left(\mathrm{diag}(A)\right) \geq g\left(\lambda\right)$, as desired. $\qquad\square$

### C.6 Proof of Theorem 4.5

Our approach to showing dominance of $\hat{\beta}_{\mathrm{ECov}}$ over $\hat{\beta}_{\mathrm{ECov}}^{\mathrm{MM}}$ parallels the classical approach of Baranchik [4], to showing that the positive part James-Stein estimator dominates the original James-Stein estimator. In this case, however, our parameter and estimates are matrix-valued, rather than vector-valued. Additionally, we contend with the added complication that the directions along which we apply shrinkage are random.

*Proof.* To begin, consider again the SVD of the matrix of least squares estimates, $\hat{\beta}_{\mathrm{LS}} = V \mathrm{diag}(\lambda^{\frac{1}{2}}) U^\top$. Recall from Proposition C.1 that $\hat{\beta}_{\mathrm{ECov}}^{\mathrm{MM}} = \hat{\beta}_{\mathrm{LS}} - \sigma^2 D \hat{\beta}_{\mathrm{LS}}^{\dagger\top}$ under Condition 4.1. Because the pseudo-inverse of $\hat{\beta}_{\mathrm{LS}}$ may be written as $\hat{\beta}_{\mathrm{LS}}^{\dagger} = U \mathrm{diag}(\lambda^{-\frac{1}{2}}) V^\top$, we rewrite $\hat{\beta}_{\mathrm{ECov}}^{\mathrm{MM}} = V \mathrm{diag}(\lambda^{\frac{1}{2}} - \sigma^2 D \lambda^{-\frac{1}{2}}) U^\top$. Comparing this estimate to the expression for $\hat{\beta}_{\mathrm{ECov}}$ in Lemma 4.4, $\hat{\beta}_{\mathrm{ECov}} = V \mathrm{diag}\left[ \lambda^{\frac{1}{2}} \odot (1 - \sigma^2 D \lambda^{-1})_+ \right] U^\top$, we see that the two estimates differ only when $\hat{\beta}_{\mathrm{ECov}}^{\mathrm{MM}}$ "flips the direction" of one or more of the singular values of $\hat{\beta}_{\mathrm{LS}}$. Our strategy to proving the theorem is to show that analogously to the "over-shrinking" of the James-Stein estimator relative to the positive part James-Stein estimator, this "over-shrinking" of singular values increases the loss of $\hat{\beta}_{\mathrm{ECov}}^{\mathrm{MM}}$ in expectation.

For convenience, we define $\rho := \lambda^{\frac{1}{2}} \odot (1 - \sigma^2 D \lambda^{-1})$ and $\rho_+ := \lambda^{\frac{1}{2}} \odot (1 - \sigma^2 D \lambda^{-1})_+$ so that $\hat{\beta}_{\mathrm{ECov}}^{\mathrm{MM}} = V \mathrm{diag}(\rho) U^\top$ and $\hat{\beta}_{\mathrm{ECov}} = V \mathrm{diag}(\rho_+) U^\top$.

To show the desired uniform risk improvement we must show that for any $\beta$,

$$\mathbb{E}\left[ \mathrm{L}(\beta, \hat{\beta}_{\mathrm{ECov}}^{\mathrm{MM}}) - \mathrm{L}(\beta, \hat{\beta}_{\mathrm{ECov}}) \right] > 0, \tag{5}$$

where $L(\beta, \hat{\beta}) = \|\hat{\beta} - \beta\|_F^2$ is squared error loss. We can rewrite this difference in loss as

$$
\begin{aligned}
L(\beta, \hat{\beta}_{\text{ECov}}^{\text{MM}}) - L(\beta, \hat{\beta}_{\text{ECov}}) &= \|\hat{\beta}_{\text{ECov}}^{\text{MM}} - \beta\|_F^2 - \|\hat{\beta}_{\text{ECov}} - \beta\|_F^2 \\
&= \|\text{diag}(\rho) - V^\top \beta U\|_F^2 - \|\text{diag}(\rho_+) - V^\top \beta U\|_F^2 \\
&= \sum_{q=1}^Q (\rho_q - V_q^\top \beta U_q)^2 - (\rho_{+q} - V_q^\top \beta U_q)^2 \\
&= \sum_{q=1}^Q \rho_q^2 - \rho_{+q}^2 - 2(V_q^\top \beta U_q)(\rho_q - \rho_{+q}),
\end{aligned}
$$

where we here (and in the proof of this theorem only) write $V_q$ and $U_q$ to denotes columns of $V$ and $U$, rather than rows. Since $\rho_q^2 \overset{a.s.}{\geq} \rho_{+q}^2$, it suffices to show that for any $\beta$ and each $q$,

$$
\mathbb{E}\left[ (V_q^\top \beta U_q)(\rho_q - \rho_{+q}) \right] < 0.
$$

To show this, we again find an even narrower but easier to prove condition will imply the one above; since $\rho_q$ and $\rho_{+q}$ differ only when $\lambda_q < \sigma^2 D$, it is enough to show that for each $0 < c < \sigma^2 D$

$$
\mathbb{E}\left[ (V_q^\top \beta U_q)\rho_q | \lambda_q = c \right] < 0. \tag{6}
$$

If we establish Equation (6), then Equation (5) obtains from the law of iterated expectation. Next, observe that since $\rho_q$ fixed and negative when $\lambda_q = c < \sigma^2 D$, Equation (5) is equivalent to

$$
\mathbb{E}\left[ V_q^\top \beta U_q | \lambda_q = c \right] > 0.
$$

Letting $U_{-q}$ and $V_{-q}$ denote the remaining columns of $U$ and $V$, respectively, we may write

$$
\mathbb{E}\left[ V_q^\top \beta U_q | \lambda_q = c \right] = \mathbb{E}\left[ \mathbb{E}\left[ V_q^\top \beta U_q | \lambda_q = c, U_{-q}, V_{-q} \right] \right]
$$

and, again through the law of iterated expectation, see that it will be sufficient to show for every $U_{-q}$ and $V_{-q}$ that $\mathbb{E}\left[ V_q^\top \beta U_q | \lambda_q = c, U_{-q}, V_{-q} \right] > 0$.

With all but one column of each of $U$ and $V$ fixed, $U_q$ and $V_q$ are determined up to signs, as unit vectors in the one dimensional subspaces orthogonal to $[\{U^{q'}\}_{q' \neq q}]$ and $[\{V^d\}_{d \neq q}]$. As such, we need only to show

$$
\mathbb{P}\left[ V_q^\top \beta U_q > 0 | U_{-q}, V_{-q}, \lambda_q = c \right] > \mathbb{P}\left[ V_q^\top \beta U_q < 0 | U_{-q}, V_{-q}, \lambda_q = c \right], \tag{7}
$$

since

$$
\begin{aligned}
&\mathbb{E}\left[ V_q^\top \beta U_q | \lambda_q, U_{-q}, V_{-q} \right] \\
&= |V_q^\top \beta U_q| \left\{ \mathbb{P}\left[ V_q^\top \beta U_q > 0 | \lambda_q, U_{-q}, V_{-q} \right] - \mathbb{P}\left[ V_q^\top \beta U_q < 0 | \lambda_q, U_{-q}, V_{-q} \right] \right\},
\end{aligned}
$$

where, in an abuse of notation, we have moved $|V_q^\top \beta U_q|$ outside the expectation since it is deterministic once we have observed $V_{-q}$ and $U_{-q}$.

That Equation (7) holds may be seen from considering the conditional probability densities for $U_q$ and $V_q$, and noting that the density is larger for $V_q$ and $U_q$ such that $V_q^\top \beta U_q$ is positive. In particular, we have that

$$
\begin{aligned}
\ln p(\hat{\beta}_{\text{LS}} | \beta, U_{-q}, V_{-q}, \lambda) &= -\frac{1}{2}\|\beta - \hat{\beta}\|_F^2 + h \\
&= -\frac{1}{2}\|V^\top \beta U - \text{diag}(\lambda^{\frac{1}{2}})\|_F^2 + h \\
&= -\frac{1}{2}(\lambda_q^{\frac{1}{2}} - V_q^\top \beta U_q)^2 + h'
\end{aligned}
$$

where $h$ and $h'$ are constants that do not depend on the signs of $U_q$ and $V_q$. Since $\lambda_q^{\frac{1}{2}}$ is positive with probability one, the conditional probability that $V_q^\top \beta U_q$ is positive is greater than that it is negative. Accordingly, we see that Equation (6) does in fact hold, and the result obtains. $\qquad \square$

# D  Gains from ECov in the high-dimensional limit – supplementary proofs

## D.1  Proof of Lemma 5.2

From the sequence of datasets, $\{\mathcal{D}_D\}_{D=1}^{\infty}$, we obtain sequences of estimates. To make explicit the dimension dependence, we denote these as explicit functions of the data, e.g. $\{\hat{\beta}_{\text{ECov}}(\mathcal{D}_D)\}_{D=1}^{\infty}$ where $\hat{\beta}_{\text{ECov}}(\mathcal{D}_D)$ denotes $\hat{\beta}_{\text{ECov}}$ in Equation (1) applied to $\mathcal{D}_D$. Furthermore, we consider the entire sequence of datasets and estimates as existing in a single probability space.

We note that Lemma D.1 establishes that $\hat{\beta}_{\text{ECov}}(\mathcal{D}_D)$ and $\hat{\beta}_{\text{ECov}}^{\text{MM}}(\mathcal{D}_D)$ coincide almost surely in the high-dimensional limit. As such, the squared error loss of these two estimates coincide almost surely in the limit, and we may write

$$
\begin{aligned}
\lim_{D\to\infty} D^{-1}\text{R}_\pi^D(\hat{\beta}_{\text{ECov}}(\mathcal{D}_D)) &= \lim_{D\to\infty} D^{-1}\mathbb{E}\left[\mathbb{E}[\|\hat{\beta}_{\text{ECov}}(\mathcal{D}_D) - \beta\|_F^2 \mid \beta]\right] \\
&= \lim_{D\to\infty} D^{-1}\mathbb{E}\big[\mathbb{E}[\|\hat{\beta}_{\text{ECov}}^{\text{MM}}(\mathcal{D}_D) - \beta\|_F^2 + \|\hat{\beta}_{\text{ECov}}(\mathcal{D}_D) - \hat{\beta}_{\text{ECov}}^{\text{MM}}(\mathcal{D}_D)\|_F^2 + \\
&\quad 2\text{tr}((\hat{\beta}_{\text{ECov}}(\mathcal{D}_D) - \hat{\beta}_{\text{ECov}}^{\text{MM}}(\mathcal{D}_D))^\top(\hat{\beta}_{\text{ECov}}^{\text{MM}}(\mathcal{D}_D) - \beta)) \mid \beta]\big] \\
&= \lim_{D\to\infty} \mathbb{E}\left[D^{-1}\mathbb{E}[\|\hat{\beta}_{\text{ECov}}^{\text{MM}}(\mathcal{D}_D) - \beta\|_F^2 \mid \beta]\right] \\
&= \lim_{D\to\infty} \mathbb{E}\left[\sigma^2 Q - \sigma^4(D - 2Q - 2)\mathbb{E}[\|\hat{\beta}_{\text{LS}}(\mathcal{D}_D)^\dagger\|_F^2 \mid \beta]\right] \\
&= \sigma^2 Q - \sigma^4 \lim_{D\to\infty} \mathbb{E}[(D - 2Q - 2)\|\hat{\beta}_{\text{LS}}(\mathcal{D}_D)^\dagger\|_F^2] \\
&= \sigma^2 Q - \sigma^4 \lim_{D\to\infty} \mathbb{E}[\text{tr}[(\tilde{\Sigma} + \sigma^2 I_Q)^{-1}] + o(1)] \\
&= \sigma^2 Q - \sigma^4 \text{tr}[(\tilde{\Sigma} + \sigma^2 I_Q)^{-1}].
\end{aligned}
$$

The third line comes from linearity of expectation and that $\|\hat{\beta}_{\text{ECov}} - \hat{\beta}_{\text{ECov}}^{\text{MM}}\| \overset{a.s.}{\to} 0$. The fourth line comes from Lemma 4.2. The second to last line comes from Lemma D.2.

We next recognize that $\text{tr}[(\tilde{\Sigma} + \sigma^2 I_Q)^{-1}] = \sum_{q=1}^Q (\lambda_q + \sigma^2)^{-1}$, where $\lambda_1, \ldots, \lambda_Q$ are the eigenvalues of $\tilde{\Sigma}$. Accordingly we may write,

$$
\lim_{D\to\infty} D^{-1}\text{R}_\pi^D(\hat{\beta}_{\text{ECov}}(\mathcal{D}_D)) = \sigma^2 Q - \sigma^4 \sum_{q=1}^Q (\lambda_q + \sigma^2)^{-1}.
$$

Furthermore since we obtain $\hat{\beta}_{\text{ID}}(\mathcal{D}_D)$ by applying $\hat{\beta}_{\text{ECov}}(\mathcal{D}_D)$ independently to the data in each group, we analogously obtain

$$
\lim_{D\to\infty} D^{-1}\text{R}_\pi^D(\hat{\beta}_{\text{ID}}(\mathcal{D}_D)) = \sigma^2 Q - \sigma^4 \sum_{q=1}^Q (\tilde{\Sigma}_{q,q} + \sigma^2)^{-1}.
$$

Putting these expressions together, we obtain

$$
\lim_{D\to\infty} D^{-1}\left[\text{R}_\pi^D(\hat{\beta}_{\text{ID}}(\mathcal{D}_D)) - \text{R}_\pi^D(\hat{\beta}_{\text{ECov}}(\mathcal{D}_D))\right] = \sigma^4\left[\sum_{q=1}^Q (\lambda_q + \sigma^2)^{-1} - \sum_{q=1}^Q (\tilde{\Sigma}_{q,q} + \sigma^2)^{-1}\right].
$$

Finally, including the additional scaling by $\sigma^{-2}Q^{-1}$ we obtain

$$
\text{Gain}(\pi, \sigma^2) = \sigma^2 Q^{-1}\left[\sum_{q=1}^Q (\lambda_q + \sigma^2)^{-1} - \sum_{q=1}^Q (\tilde{\Sigma}_{q,q} + \sigma^2)^{-1}\right]
$$

as desired.

**Lemma D.1.** *Under the conditions of Lemma 5.2,* $\lim_{D\to\infty} \|\hat{\beta}_{\text{ECov}}(\mathcal{D}_D) - \hat{\beta}_{\text{ECov}}^{\text{MM}}(\mathcal{D}_D)\|_F = 0$ *almost surely.*

*Proof.* Note that under the conditions of Lemma 5.2, Lemma 4.4 provides that $\hat{\beta}_{\mathrm{ECov}}(\mathcal{D}_D)$ and $\hat{\beta}_{\mathrm{ECov}}^{\mathrm{MM}}(\mathcal{D}_D)$ differ only when $\hat{\Sigma}^{\mathrm{MM}}$ is not positive definite; otherwise $\hat{\Sigma}^{\mathrm{MM}} = \hat{\Sigma}$. Since $\hat{\Sigma}^{\mathrm{MM}} = D^{-1}\hat{\beta}_{\mathrm{LS}}(\mathcal{D}_D)^\top \hat{\beta}_{\mathrm{LS}}(\mathcal{D}_D) - \sigma^2 I_Q$, by Lemma D.3 $\hat{\Sigma}^{\mathrm{MM}}$ will be positive definite for all $D$ above some $D'$ almost surely, and so $\hat{\beta}_{\mathrm{ECov}}(\mathcal{D}_D)$ and $\hat{\beta}_{\mathrm{ECov}}^{\mathrm{MM}}(\mathcal{D}_D)$ become equal for all $D$ large enough, implying strong convergence. $\qquad\square$

**Lemma D.2.** *Under the conditions of Lemma 5.2, $\lim_{D\to\infty} D\|\hat{\beta}_{\mathrm{LS}}(\mathcal{D}_D)^\dagger\|_F^2 = \mathrm{tr}[(\tilde{\Sigma} + \sigma^2 I_Q)^{-1}]$ almost surely.*

*Proof.* Recall that $\|\hat{\beta}_{\mathrm{LS}}(\mathcal{D}_D)^\dagger\|_F^2 = \mathrm{tr}[(\hat{\beta}_{\mathrm{LS}}(\mathcal{D}_D)^\top \hat{\beta}_{\mathrm{LS}}(\mathcal{D}_D))^{-1}]$. As such, we may write $D\|\hat{\beta}_{\mathrm{LS}}(\mathcal{D}_D)^\dagger\|_F^2 = \mathrm{tr}[(D^{-1}\hat{\beta}_{\mathrm{LS}}(\mathcal{D}_D)^\top \hat{\beta}_{\mathrm{LS}}(\mathcal{D}_D))^{-1}]$. By Lemma D.3 $D^{-1}\hat{\beta}_{\mathrm{LS}}(\mathcal{D}_D)^\top \hat{\beta}_{\mathrm{LS}}(\mathcal{D}_D) \overset{a.s.}{\to} \tilde{\Sigma} + \sigma^2 I_Q$, and so we can see that $D\|\hat{\beta}_{\mathrm{LS}}(\mathcal{D}_D)^\dagger\|_F^2 \overset{a.s.}{\to} \mathrm{tr}[(\tilde{\Sigma} + \sigma^2 I_Q)^{-1}]$ as desired. $\qquad\square$

**Lemma D.3.** *Under the conditions of Lemma 5.2 $\lim_{D\to\infty} D^{-1}\hat{\beta}_{\mathrm{LS}}(\mathcal{D}_D)^\top \hat{\beta}_{\mathrm{LS}}(\mathcal{D}_D) = \tilde{\Sigma} + \sigma^2 I_Q$ almost surely.*

*Proof.* It suffices to show strong convergence element wise, as this implies strong convergence in all other relevant norms. For convenience, let $C^{(D)} := D^{-1}\hat{\beta}_{\mathrm{LS}}(\mathcal{D}_D)^\top \hat{\beta}_{\mathrm{LS}}(\mathcal{D}_D)$. Note that we may write each entry $C_{q,q'}^{(D)} = \sum_{d=1}^D D^{-1}\hat{\beta}_{\mathrm{LS}}(\mathcal{D}_D)_d^q \hat{\beta}_{\mathrm{LS}}(\mathcal{D}_D)_d^{q'}$ as a sum of $D$ i.i.d. terms. Notably, each term $\hat{\beta}_{\mathrm{LS}}(\mathcal{D}_D)_d^q \cdot \hat{\beta}_{\mathrm{LS}}(\mathcal{D}_D)_d^{q'}$ is a product of two Gaussian random variables and is therefore sub-exponential with some non-negative parameters $(\nu, \alpha)$ (see e.g. Wainwright [63, Definition 2.7]). As a result, $C^{(D)}$ is then sub-exponential with parameters $(D^{-\frac{1}{2}}\nu, D^{-1}\alpha)$. Therefore, for any constant $b$ satisfying $0 < b < \nu^2/\alpha$, by Wainwright [63, Proposition 2.9] we have that

$$\mathbb{P}\left[\left|C_{q,q'}^{(D)} - \mathbb{E}[C_{q,q'}^{(D)}]\right| \ge b\right] \le 2\exp\{-\frac{D}{2}b^2/\nu^2\}.$$

This rapid, exponential decay in tail probability with $D$ implies that for small $b$,

$$\sum_{D=1}^\infty \mathbb{P}\left[\left|C_{q,q'}^{(D)} - \mathbb{E}[C_{q,q'}^{(D)}]\right| \ge b\right] \le \infty.$$

Therefore, by the Borel-Cantelli lemma we see that $|C_{q,q'}^{(D)} - \mathbb{E}[C_{q,q'}^{(D)}]| \overset{a.s.}{\to} 0$. Since $\mathbb{E}[C^{(D)}] = \tilde{\Sigma} + \sigma^2 I_Q$ for each $D$, this implies that $\lim_{D\to\infty} D^{-1}\hat{\beta}_{\mathrm{LS}}(\mathcal{D}_D)^\top \hat{\beta}_{\mathrm{LS}}(\mathcal{D}_D) = \tilde{\Sigma} + \sigma^2 I_Q$ almost surely. $\qquad\square$

### D.2 Further discussion of Theorem 5.3

We here give further detail related to the proof of Theorem 5.3 and introduce additional notation used in the remainder of the section. Recall from Lemma 5.2 that $\mathrm{Gain}(\pi, \sigma^2) = \sigma^2 Q^{-1}[\sum_{q=1}^Q (\lambda_q + \sigma^2)^{-1} - \sum_{q=1}^Q (\tilde{\Sigma}_{q,q} + \sigma^2)^{-1}]$. For convenience, we will use $\ell := \mathrm{diag}(\tilde{\Sigma})^\downarrow$ to denote the $Q$-vector of diagonal entries of $\tilde{\Sigma}$ sorted in descending order. Similarly, we take $\lambda$ to be the $Q$-vector of eigenvalues of $\tilde{\Sigma}$, again sorted in descending order. Next, it is useful to rewrite

$$\mathrm{Gain}(\pi, \sigma^2) = \sigma^2 Q^{-1}\left[\vec{f}(\lambda) - \vec{f}(\ell)\right]$$

where $\vec{f}(x) := \sum_{q=1}^Q f(x_q) = \sum_{q=1}^Q (\sigma^2 + x_q)^{-1}$ (where $f(x) := (\sigma^2 + x)^{-1}$).

The key theoretical tool used in establishing Theorem 5.3 is the Schur-Horn theorem. We state this result below, adapted from Horn [32, Theorem 5]. The Schur-Horn theorem guarantees that $\lambda$ majorizes $\ell$. In particular, an $N$-vector $a$ is said to majorize a second $N$-vector $b$ if $\sum_{n=1}^N a_n = \sum_{n=1}^N b_n$ and for all $N' \le N$,

$$\sum_{n=1}^{N'} a_n^\downarrow \ge \sum_{n=1}^{N'} b_n^\downarrow,$$

where for a vector $v$, we use $v^{\downarrow}$ to denote the vector with the same components as $v$, sorted in descending order. As captured by Theorem 5.3, we can therefore see that $\mathrm{Gain}(\pi, \sigma^2)$ is non-negative for any $\tilde{\Sigma}$ by observing that $\vec{f}$ is Schur-convex (since $f$ is convex).

**Theorem D.4** (Schur-Horn). *A vector $\ell$ can be the diagonal of a Hermitian matrix with (repeated) eigenvalues $\lambda$ if and only if $\lambda$ majorizes $\ell$.*

## D.3 Proof of Theorem 5.4

We here show that $\mathrm{Gain}(\pi, \sigma^2)$ is upper bounded as

$$\mathrm{Gain}(\pi, \sigma^2) \leq \sigma^2 Q^{-1} f''(\lambda_{\min}) \|\lambda\|_2 \|\lambda - \ell\|_2$$
$$= 2\sigma^2 Q^{-1} \|\lambda\|_2 \|\lambda - \ell\|_2 / (\sigma^2 + \lambda_{\min})^3,$$

and lower bounded as

$$\mathrm{Gain}(\pi, \sigma^2) \geq \frac{1}{2}\sigma^2 Q^{-1} f''(\lambda_{\max}) \|\lambda - \ell\|^2$$
$$= \sigma^2 Q^{-1} \|\lambda - \ell\|^2 / (\sigma^2 + \lambda_{\max})^3,$$

where $f''(x) := \frac{d^2}{dx^2} f(x)$ where $f$ is as defined in Appendix D.2.

We obtain both bounds with quadratic approximations to $f$. In particular, we define $g_\alpha$ as the 2$^{\mathrm{nd}}$ order Taylor approximation of $f$ expanded at $\alpha$,

$$g_\alpha(x) := f(\alpha) + f'(\alpha)(x - \alpha) + \frac{1}{2}f''(\alpha)(x - \alpha)^2,$$

and note that by Lemma D.5

$$\vec{g}_{\lambda_{\max}}(\lambda) - \vec{g}_{\lambda_{\max}}(\ell) \leq \vec{f}(\lambda) - \vec{f}(\ell) \leq \vec{g}_{\lambda_{\min}}(\lambda) - \vec{g}_{\lambda_{\min}}(\ell), \tag{8}$$

where $\vec{g}_\alpha(x) := \sum_{q=1}^{Q} g_\alpha(x_q)$.

**Proof of upper bound.** We obtain the desired upper bound as follows.

Equation (8) and Lemma D.6 allow us to see

$$\mathrm{Gain}(\pi, \sigma^2) \leq \sigma^2 Q^{-1} \left[ \vec{g}_{\lambda_{\min}}(\lambda) - \vec{g}_{\lambda_{\min}}(\ell) \right]$$
$$= \frac{1}{2}\sigma^2 Q^{-1} f''(\lambda_{\min})(\|\lambda\|^2 - \|\ell\|^2). \tag{9}$$

Since $f''$ is positive on $\mathbb{R}_+$, the problem reduces to upper bounding $\|\lambda\|^2 - \|\ell\|^2$.

In particular, we find

$$\|\lambda\|^2 - \|\ell\|^2 = \langle \lambda + \ell, \lambda - \ell \rangle \tag{10}$$
$$\leq \|\lambda + \ell\| \|\lambda - \ell\| \qquad \text{// by Cauchy-Schwarz} \tag{11}$$
$$= \sqrt{\|\lambda\|^2 + 2\langle \lambda, \ell \rangle + \|\ell\|^2} \, \|\lambda - \ell\| \tag{12}$$
$$\leq \sqrt{\|\lambda\|^2 + 2\|\lambda\|\|\ell\| + \|\ell\|^2} \, \|\lambda - \ell\| \quad \text{// by Cauchy-Schwarz} \tag{13}$$
$$\leq 2\|\lambda\| \|\lambda - \ell\| \qquad \text{// Since } \|\lambda\| \geq \|\ell\|, \tag{14}$$

where we can see that $\|\lambda\| \geq \|\ell\|$ by noting that $\|\cdot\|^2$ is Schur convex, and again appealing to the Schur-Horn Theorem. The desired upper bound obtains by combining Equations (9) and (10).

**Proof of lower bound.** We begin as we did for the upper bound. Equation (8) and Lemma D.6 allow us to see

$$\mathrm{Gain}(\pi, \sigma^2) \geq \sigma^2 Q^{-1} \left[ \vec{g}_{\lambda_{\max}}(\lambda) - \vec{g}_{\lambda_{\max}}(\ell) \right]$$
$$= \frac{1}{2}\sigma^2 Q^{-1} f''(\lambda_{\max})(\|\lambda\|^2 - \|\ell\|^2). \tag{15}$$

Since, again, $f''$ is positive on $\mathbb{R}_+$, the problem reduces to lower bounding $\|\lambda\|^2 - \|\ell\|^2$.

In particular, we would like to show $\|\lambda\|^2 - \|\ell\|^2 \geq \|\lambda - \ell\|^2$. We can arrive at this bound with a particular expansion of $\|\lambda - \ell\|^2$ and using Lemma D.7, which again leverages the fact that $\lambda$ majorizes $\ell$. Specifically, we write

$$
\begin{aligned}
\|\lambda - \ell\|^2 &= \langle \lambda - \ell, \lambda \rangle - \langle \lambda - \ell, \ell \rangle \\
&= \|\lambda\|^2 - \left[ \langle \lambda, \ell \rangle + \langle \lambda - \ell, \ell \rangle \right] \\
&= \|\lambda\|^2 - \|\ell\|^2 - \left[ \langle \lambda, \ell \rangle - \langle \ell, \ell \rangle + \langle \lambda - \ell, \ell \rangle \right] \\
&= \|\lambda\|^2 - \|\ell\|^2 - 2\langle \lambda - \ell, \ell \rangle \\
&\leq \|\lambda\|^2 - \|\ell\|^2
\end{aligned}
\tag{16}
$$

where the last line follows from Lemma D.7, which provides that $\langle \lambda - \ell, \ell \rangle \geq 0$ since, from the Schur-Horn theorem for any $Q' \leq Q$ $\sum_{q=1}^{Q'} \lambda_q - \ell_q \geq 0$, and $\ell$ has non-negative, non-increasing entries. We obtain the desired lower bound by combining Equations (15) and (16).

**Lemma D.5.** *Let $\lambda$ and $\ell$ be Q-vectors of non-negative reals with non-increasing entries, and let $\lambda$ majorize $\ell$. Consider $\vec{f} : \mathbb{R}^Q \to \mathbb{R}, x \mapsto \sum_{q=1}^Q f(x_q) = \sum_{q=1}^Q (\sigma^2 + x_q)^{-1}$ (where $f(v) := (\sigma^2 + v)^{-1}$) for any $\sigma^2 > 0$, and define $g_\alpha$ to be the 2^{nd} order Taylor approximation of $f$ expanded at $\alpha$,*

$$
g_\alpha(x) := f(\alpha) + f'(\alpha)(x - \alpha) + \frac{1}{2} f''(\alpha)(x - \alpha)^2.
$$

*Then*

$$
\vec{g}_{\lambda_{\max}}(\lambda) - \vec{g}_{\lambda_{\max}}(\ell) \leq \vec{f}(\lambda) - \vec{f}(\ell) \leq \vec{g}_{\lambda_{\min}}(\lambda) - \vec{g}_{\lambda_{\min}}(\ell),
$$

*where $\vec{g}_\alpha(x) := \sum_{q=1}^Q g_\alpha(x_q)$ and $\lambda_{\max} = \lambda_1$ and $\lambda_{\min} = \lambda_Q$ are the largest and smallest entries of $\lambda$, respectively.*

*Proof.* If there are indices $q$ for which $\lambda_q = \ell_q$, remove them (they do not affect $\vec{f}(\ell) - \vec{f}(\lambda)$). If all are equal, $\lambda = d$ and so the result is trivial, otherwise we have $Q \geq 2$ entries with $\lambda_q \neq \ell_q$.

We begin with the lower bound; the upper bound follows similarly. For this, it suffices to show $\vec{f}(\lambda) - \vec{f}(\ell) - \left( \vec{g}_{\lambda_{\max}}(\lambda) - \vec{g}_{\lambda_{\max}}(\ell) \right) \geq 0$.

We first express this difference as an inner product

$$
\begin{aligned}
\vec{f}(\lambda) - \vec{f}(\ell) - \left( \vec{g}_{\lambda_{\max}}(\lambda) - \vec{g}_{\lambda_{\max}}(\ell) \right) &= \sum_{q=1}^Q \left[ (f - g_{\lambda_{\max}})(\lambda_q) - (f - g_{\lambda_{\max}})(\ell_q) \right] \\
&= \sum_{q=1}^Q (\lambda_q - \ell_q) \left[ \frac{(f - g_{\lambda_{\max}})(\lambda_q) - (f - g_{\lambda_{\max}})(\ell_q)}{\lambda_q - \ell_q} \right] \\
&\text{// defining each } h_q := \frac{(f - g_{\lambda_{\max}})(\lambda_q) - (f - g_{\lambda_{\max}})(\ell_q)}{\lambda_q - \ell_q} \\
&= \sum_{q=1}^Q (\lambda_q - \ell_q) h_q \\
&= \langle \lambda - \ell, h \rangle
\end{aligned}
$$

where $h = [h_1, h_2, \ldots, h_Q]^\top$.

We will complete our proof by leveraging Lemma D.7, which provides that $\langle a, b \rangle \geq 0$ for any $Q$-vector $a$ satisfying $\sum_{q=1}^Q a_q = 0$ and $\sum_{q=1}^{Q'} a_q \geq 0$ for every $Q' \leq Q$, and $Q$-vector $b$ with non-increasing entries.

It therefore remains only to show that $\lambda - \ell$ and $h$ satisfy the conditions of Lemma D.7. Since the entries of $\lambda$ and $\ell$ are taken to be in descending order, the condition that $\sum_{q=1}^{Q'} (\lambda - \ell)_q \geq 0$ for any $Q' \leq Q$, follows from the Schur-Horn theorem. Likewise, this theorem provides that $\sum_{q=1}^Q \lambda_q = \sum_{q=1}^Q \ell_q$, and therefore that $\sum_{q=1}^Q (\lambda - \ell)_q = 0$, so that $\lambda - \ell$ meets condition (2) of the lemma.

We next confirm that $h$ has non-increasing entries by considering an expansion of the expressions for each $h_q$. In particular, observe that

$$h_q = \frac{(f - g_{\lambda_{\max}})(\lambda_q) - (f - g_{\lambda_{\max}})(\ell_q)}{\lambda_q - \ell_q}$$

$$= (\lambda_q - \ell_q)^{-1} \left\{ f(\lambda_q) - f(\ell_q) - \left[ g_{\lambda_{\max}}(\lambda_q) - g_{\lambda_{\max}}(\ell_q) \right] \right\}$$

$$= (\lambda_q - \ell_q)^{-1} \Big\{ \frac{(\sigma^2 + \ell_q) - (\sigma^2 + \lambda_q)}{(\sigma^2 + \ell_q)(\sigma^2 + \lambda_q)} -$$

$$\left[ (\lambda_q - \ell_q) f'(\lambda_{\max}) + \frac{1}{2}((\lambda_q - \lambda_{\max})^2 - (\ell_q - \lambda_{\max})^2) f''(\lambda_{\max}) \right] \Big\}$$

$$= (\sigma^2 + \lambda_{\max})^{-2} - (\sigma^2 + \ell_q)^{-1}(\sigma^2 + \lambda_q)^{-1} - \frac{1}{2}(\lambda_q - \ell_q)^{-1}(\sigma^2 + \lambda_{\max})^{-3} \left[ \lambda_q^2 - \ell_q^2 - 2\lambda_{\max}(\lambda_q - \ell_q) \right]$$

$$= (\sigma^2 + \lambda_{\max})^{-2} - (\sigma^2 + \ell_q)^{-1}(\sigma^2 + \lambda_q)^{-1} - \frac{1}{2}(\sigma^2 + \lambda_{\max})^{-3} \left[ \lambda_q + \ell_q - 2\lambda_{\max} \right].$$

Next define $\phi(a, b) = (\sigma^2 + \lambda_{\max})^{-2} - (\sigma^2 + a)^{-1}(\sigma^2 + b)^{-1} - \frac{1}{2}(\sigma^2 + \lambda_{\max})^{-3} [b + a - 2\lambda_{\max}]$, so that for each $q$, $h_q = \phi(\ell_q, \lambda_q)$. Now, for $q' > q$, we may write

$$h_{q'} - h_q = \phi(\ell_{q'}, \lambda_{q'}) - \phi(\ell_q, \lambda_q)$$

$$= \int_{\ell_q}^{\ell_{q'}} \frac{\partial}{\partial a} \phi(a, \lambda_q) da + \int_{\lambda_q}^{\lambda_{q'}} \frac{\partial}{\partial b} \phi(\ell_{q'}, b) db. \tag{17}$$

Next note that

$$\frac{\partial}{\partial a} \phi(a, b) = (\sigma^2 + a)^{-2}(\sigma^2 + b)^{-1} - \frac{1}{2}(\sigma^2 + \lambda_{\max})^{-3}$$

and

$$\frac{\partial}{\partial b} \phi(a, b) = (\sigma^2 + a)^{-1}(\sigma^2 + b)^{-2} - \frac{1}{2}(\sigma^2 + \lambda_{\max})^{-3}$$

from which we can see that $\frac{\partial}{\partial a}\phi(a, b)$ and $\frac{\partial}{\partial b}\phi(a, b)$ are positive for $a, b \in [\lambda_{\min}, \lambda_{\max}]$. Accordingly, Equation (17) provides that $h_{q'} - h_q \leq 0$, since $\ell_{q'} \leq \ell_q$ and $\lambda_{q'} \leq \lambda_q$ for $q' > q$, because the entries of $\ell$ and $\lambda$ are non-increasing. Therefore $h_{q'} \leq h_q$, completing the proof. $\square$

**Lemma D.6.** *Consider the quadratic function $\vec{h}(x) = \sum_{q=1}^{Q}(ax_q^2 + bx_q + c)$. Let $\lambda, \ell \in \mathbb{R}^Q$ satisfy $\sum_{q=1}^{Q} \lambda_q = \sum_{q=1}^{Q} \ell_q$. Then*

$$\vec{h}(\ell) - \vec{h}(\lambda) = a(\|\ell\|^2 - \|\lambda\|^2).$$

*Proof.* The result follows from the simple algebraic rearrangement below,

$$\vec{h}(\ell) - \vec{h}(\lambda) = \sum_{q=1}^{Q}(a\ell_q^2 + b\ell_q + c) - (a\lambda_q^2 + b\lambda_q + c)$$

$$= \sum_{q=1}^{Q} a\ell_q^2 - a\lambda_q^2$$

$$= a(\|\ell\|^2 - \|\lambda\|^2).$$

$\square$

**Lemma D.7.** *Let $x$ be a $Q$-vector satisfying for each $Q' \leq Q$, $\sum_{q=1}^{Q'} x_q \geq 0$, and let $y$ be a $Q$-vector with non-increasing entries. If additionally either (1) $y$ has non-negative entries or (2) $\sum_{q=1}^{Q} x_q = 0$ then $\langle x, y \rangle \geq y_Q \sum_{q=1}^{Q} x_q \geq 0$.*

*Proof.* We first prove the lemma under condition (1) by induction. The base case of $Q = 1$ is trivial; $\langle x, y \rangle = x_1 y_1$ and under (1) $x_1$ and $y_1$ are non-negative and under (2) $x_1 = 0$.

Assume the result holds for $Q - 1$. Then

$$\langle x, y \rangle = y_Q x_Q + \langle x_{1:Q-1}, y_{1:Q-1} \rangle \tag{18}$$

$$\geq y_Q x_Q + y_{Q-1} \sum_{q=1}^{Q-1} x_q \quad \text{// by the inductive hypothesis} \tag{19}$$

$$\geq y_Q x_Q + y_Q \sum_{q=1}^{Q-1} x_q \quad \text{// since } y_{Q-1} \geq y_Q \text{ and } \sum_{q=1}^{Q-1} x_q \geq 0 \tag{20}$$

$$= y_Q \sum_{q=1}^{Q} x_q \geq 0 \quad \text{// since } y_Q \text{ and } \sum_{q=1}^{Q} x_q \text{ are non-negative.} \tag{21}$$

This provides the desired inductive step, completing the proof under condition (1).

Under condition (2), consider $y' = y - \min_q y_q \mathbf{1}_Q$. Then

$$\langle x, y \rangle = \langle x, y' \rangle + \min_q y_q \langle x, \mathbf{1}_Q \rangle$$

$$= \langle x, y' \rangle.$$

Since $y'$ now has non-negative entries, condition (1) is satisfied and the result follows. $\qquad \square$

### D.4  Proof of Corollary 5.5

We establish the corollary with a brief sequence of upper bounds following from our initial upper bound in Theorem 5.3. In particular, the theorem provides

$$\text{Gain}(\pi, \sigma^2) \leq 2\sigma^2 Q^{-1} \|\lambda^{\downarrow}\| \|\ell^{\downarrow} - \lambda^{\downarrow}\| / (\sigma^2 + \lambda_{\min})^3.$$

We begin by simplifying this upper bound. As a first step, note that

$$\|\ell^{\downarrow} - \lambda^{\downarrow}\|^2 = \|\ell\|^2 + \|\lambda\|^2 - 2\langle \ell^{\downarrow}, \lambda^{\downarrow} \rangle$$

$$\leq 2\|\lambda\|^2.$$

As such, we can simplify our upper bound as

$$\text{Gain}(\pi, \sigma^2) \leq 2\sigma^2 Q^{-1} \|\lambda\| \|\ell^{\downarrow} - \lambda^{\downarrow}\| / (\sigma^2 + \lambda_{\min})^3$$

$$\leq 4\sigma^2 Q^{-1} \|\lambda\|^2 / (\sigma^2 + \lambda_{\min})^3 \tag{22}$$

$$\leq 4\kappa^2 \lambda_{\min}^2 \sigma^2 / (\sigma^2 + \lambda_{\min})^3$$

where $\kappa := \lambda_{\max} / \lambda_{\min}$ is the condition number of $\tilde{\Sigma}$.

We then obtain the first bound by noting that

$$\lambda_{\min}^2 \sigma^2 / (\sigma^2 + \lambda_{\min})^3 \leq \lambda_{\min}^2 \sigma^2 / (\sigma^2)^2 / \lambda_{\min}$$

$$\leq \lambda_{\min} / \sigma^2$$

and the second by noting that

$$\lambda_{\min}^2 \sigma^2 / (\sigma^2 + \lambda_{\min})^3 \leq \lambda_{\min}^2 \sigma^2 / (\lambda_{\min})^3$$

$$\leq \sigma^2 / \lambda_{\min}.$$

Substituting these expressions into Equation (22) provides the desired expressions in Corollary 5.5.

### D.5  Extensions to random design matrices

The asymptotic formulation in Section 5 may allow us to relax Condition 4.1. In particular, Theorem 5.3 and Theorem 5.4 depend on this condition only through Lemma 5.2, which provides an analytic expression for the asymptotic gain. We conjecture that this condition may be satisfied for certain sequences of datasets with random design matrices of increasing dimension. For example if

for each group $q$, the number of data points $N_D^q$ grows as $\omega(D^2)$ and if each of the covariates are each distributed as $X_{n,d}^q \overset{i.i.d.}{\sim} \mathcal{N}(0, \sigma_q^2/(\sigma^2 I_D^2))$, then an asymptotic analogue of Condition 4.1 will be satisfied in the sense that $\|\sigma_q^{-2} X^{q\top} X^q - \sigma^2 I_D\|_2$ will be $o(1/\sqrt{D})$ (see e.g. Wainwright [63, Theorem 6.5]). As a result, we can expect the sequence of estimates $\hat{\beta}_{\mathrm{ECov}}$ to converge to estimates with the simplified form utilized in the proof of Lemma 5.2 fast enough that the asymptotic gains are equal in these two cases.

Making this argument rigorous, however, requires contending with convergence of sequences of random variables of changing dimension (recall that we consider $D \to \infty$). This technical aspect complicates the required theoretical analysis because common tools (e.g. continuous mapping theorems) do not apply in this setting. We leave further analysis of $\hat{\beta}_{\mathrm{ECov}}$ with random design matrices to future work.

# E  Experiments Supplementary Results and Details

## E.1  Simulations additional details

We here describe the details of the simulated datasets discussed in Section 6. For each of the dimensions $D$ and each of the 20 replicates we first generated covariate effects for all $Q = 10$ groups. To do this, we began by setting $\Sigma$; for the correlated covariate effects experiments (Figure 1 Left) we generating a random $Q \times Q$ matrix of orthonormal vectors $U$ and set $\Sigma = U\mathrm{diag}([2^0, 2^{-1}, \ldots, 2^{Q-1}]^\top)U^\top$, and for independent effects (Figure 1 Right) we set $\Sigma = I_Q$. We then simulated covariate effects as $\beta_d \overset{i.i.d.}{\sim} \mathcal{N}(0, \Sigma)$.

We next simulated the design matrices. For each group $q$, we chose a random number of data points $N^q \sim \mathrm{Pois}(\lambda = 1000)$, and for each data point $n = 1, \ldots, N^q$ sampled $X_n^q \sim \mathcal{N}(0, (1/1000)I_D)$ so that for each group $\mathbb{E}[X^{q\top} X^q] = I_D$. Finally, we generated each response as $Y_n^q \overset{indep}{\sim} \mathcal{N}(X_n^{q\top}\beta^q, 1)$.

For $\hat{\beta}_{\mathrm{EGroup}}$, we estimated the $D \times D$ covariance $\Gamma$ by maximum marginal likelihood. We did this with an EM algorithm closely related to Algorithm 1. See e.g. Gelman et al. [24, Chapter 15 sections 4-5] for an explanation of the relevant conjugacy calculations in a more general case that includes a hyper-prior on $\Gamma$.

## E.2  Practical moment estimation for poorly conditioned problems

The moment based estimator (using $\hat{\Sigma}^{\mathrm{MM}}$ in Section 4) is unstable in the two real data applications discussed in Section 6 due to poor conditioning of the design matrices leading $\hat{\beta}_{\mathrm{LS}}$ to have high variance. To overcome this limitation, we instead used an adapted moment estimation procedure which is less sensitive to this poor conditioning. While, in agreement with Theorem 4.5, this approach performs worse than $\hat{\beta}_{\mathrm{ECov}}$ (see Figure 3) we report it nonetheless because it has lower computational cost and may be appealing for larger scale applications. We describe this approach here. We note however that moment based estimates of the sort we consider here do not naturally extend to logistic regression and so are not reported for our application to CIFAR10.

We first introduce some additional notation. For each group $q$ consider the reduced singular value decomposition $X^q = S^q\mathrm{diag}(\omega^q)R^{q\top}$, where $S^q$ and $R^q$ are $N^q \times D$ and $D \times D$ matrices with orthonormal columns and $\omega^q$ is a $D$-vector of non-negative singular values. Next define for each group $W^q := S^{q\top} X^q$ and $Z^q := S^{q\top} Y^q$, which we may interpret as a $D \times D$ matrix of pseudo-covariates and $D$-vector of pseudo-responses, respectively. Next define $\Omega$ to be the $Q \times Q$ matrix with entries $\Omega_{q,q'} := \mathrm{tr}(W^{q\top} W^{q'})^{-1}$ and $\vec{\sigma}^2 := [\sigma_1^2, \sigma_2^2, \ldots, \sigma_Q^2]^\top$. Lastly, let $Z = [Z^1, Z^2, \ldots, Z^Q]$ be the $D \times Q$ matrix of all pseudo-responses. Our new moment estimator is

$$\hat{\Sigma}^{\mathrm{MM}} := [Z^\top Z - D\mathrm{diag}(\vec{\sigma}^2)] \odot \Omega.$$

We next show hat $\mathbb{E}[\hat{\Sigma}^{\mathrm{MM}}] = \Sigma$ under correct prior and likelihood specification. Note first that if $\delta$ is a $D \times Q$ matrix with i.i.d. standard normal entries we may write

$$Z \overset{d}{=} [W^1\beta^1, W^2\beta^2, \ldots, W^Q\beta^Q] + \delta\mathrm{diag}(\vec{\sigma}^2).$$

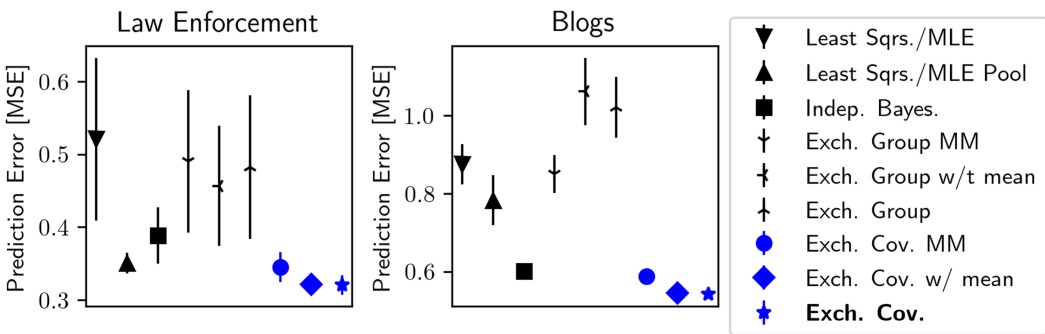

Figure 3: Performances of additional methods on the law enforcement and blog datasets. Uncertainty intervals are $\pm 1\text{SEM}$.

As such, for each $q$ and $q'$, we have that

$$
\begin{aligned}
\mathbb{E}[(Z^\top Z)_{q,q'}] &= \mathbb{E}[Z^{q\top} Z^q] \\
&= \mathbb{E}[\beta^{q\top} W^{q\top} W^{q'} \beta^{q'}] + \mathbb{I}[q = q']\sigma_q^2 D \\
&= \text{tr}(W^{q\top} W^{q'} \mathbb{E}[\beta^{q'} \beta^{q\top}]) + \mathbb{I}[q = q']\sigma_q^2 D \\
&= \Omega_{q,q'}^{-1} \Sigma_{q,q'} + \mathbb{I}[q = q']\sigma_q^2 D.
\end{aligned}
$$

Accordingly, we can see that each entry of $\hat\Sigma^{\text{MM}}$ has expectation $\mathbb{E}[\hat\Sigma_{q,q'}^{\text{MM}}] = \Sigma_{q,q'}$, which establishes unbiasedness.

However, this moment estimate still has the limitation that it evaluates to a non positive semidefinite matrix with positive probability. Under the expectation that, in line with Theorem 4.3 the very small and negative eigenvalues of $\hat\Sigma^{\text{MM}}$ might lead to over-shrinking, we performed an additional step of clipping these eigenvalues to force the resulting estimate to be reasonably well conditioned. In particular, if our initial estimate had eigendecomposition $\hat\Sigma^{\text{MM}} = U\text{diag}(\lambda)U^\top$, we instead used $\hat\Sigma^{\text{MM}} = U\text{diag}(\tilde\lambda)U^\top$, where for each $q$, we have $\tilde\lambda_q = \max(\lambda_q, \lambda_{\max}/100)$ so that the condition number of the modified estimate was at most 100. Though we did not find the performance of the resulting estimates to be very sensitive to this cutoff, we view requirement for these partly subjective implementation choices required to make the $\hat\beta_{\text{ECov}}^{\text{MM}}$ effective in practice to be a downside of the approach as compared to $\hat\beta_{\text{ECov}}$, which avoids such choices by estimating $\Sigma$ by maximum marginal likelihood.

Compared to the iterative EM algorithms, which rely on matrix inversions at each iteration, computation of $\hat\Sigma^{\text{MM}}$ is much faster. In each of our experiments, computing it requires less than one second.

### E.3 Allowing for non-zero means a priori in hierarchical Bayesian estimates

In the development of our approach in Section 2 we imposed the restriction that $\mathbb{E}[\beta_d] = 0$ a priori. Though in general one might prefer to let $\beta$ have some nontrivial mean (as Lindley and Smith [44] do in the context of exchangeability of effects across groups) this assumption simplifies the resulting estimators, theory, and notation. When $\beta$ is permitted to have a non-zero mean, conjugacy maintains and the methodology presented in Section 3 may be updated to accommodate the change. While we omit a full explanation of the tedious details of this variation, we include its implementation in our code and the performance of the resulting empirical Bayesian estimators in Figures 3 to 5. From these empirical results we see that removing this restriction has little impact on the performance of the resulting estimators. Notably, our results in these figures reveal that the same is true for choosing to include or exclude a prior mean for the exchangeability of effects across groups prior.

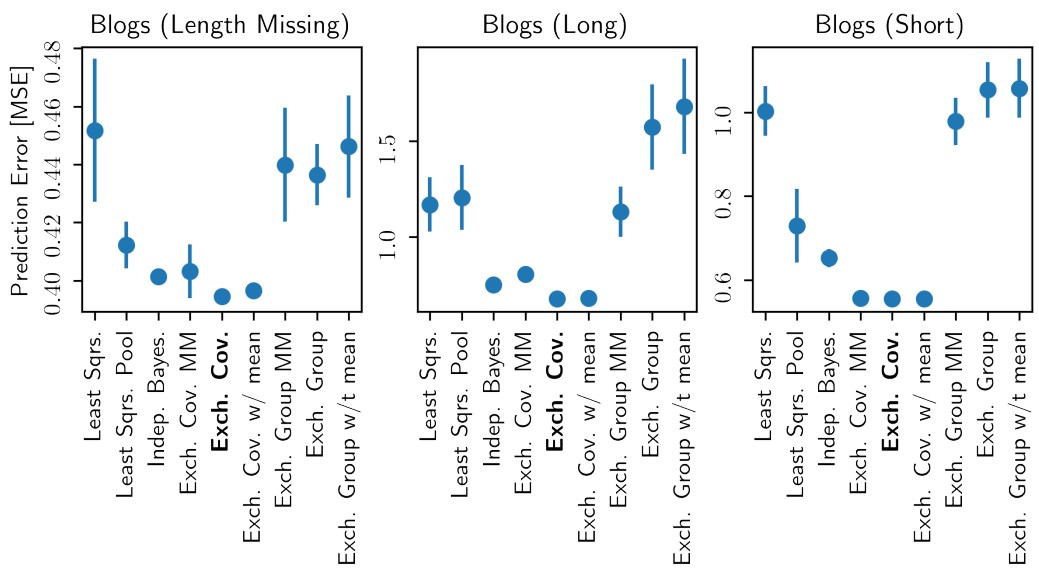

Figure 4: Performances of methods on the blog dataset, segmented by post type. Uncertainty intervals are ±1SEM.

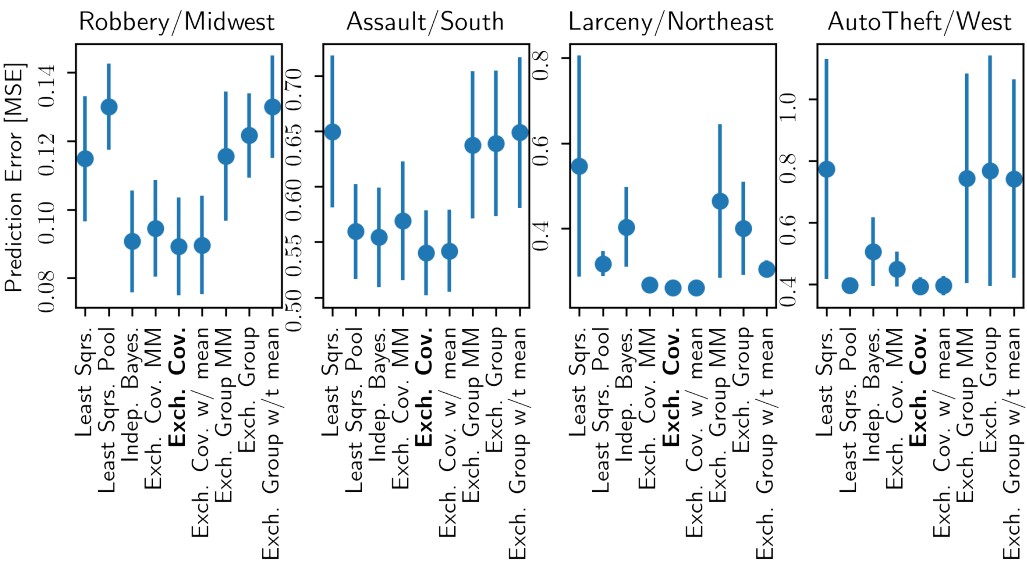

Figure 5: Performances of methods on the law enforcement dataset, segmented by region and recorded offense categorization. Uncertainty intervals are ±1SEM.

### E.4 Additional details on datasets

In each of the two regression applications, for each component dataset, we mean centered and variance-normalized the responses. Additionally, we Winsorized the responses by group; in particular, we clipped values more than 2 standard deviations from the mean.

**BlogFeedback Data Set details**   Given the nature of the features included in the blog dataset used in the main text (which are summarizing characteristics rather than readable text), we believe it may be possible to find the blog post that corresponds to a particular data point. But we believe it is unlikely that the dataset directly contains any personally identifiable information. The blog information was obtained by web-crawling on publicly posted pages, so it is unlikely that consent for inclusion of the content into this dataset was obtained.

**Communities and Crime Dataset details**   All data in this dataset was obtained through official channels. This dataset is composed of statistics aggregated at the community level, so it is less likely (though not impossible) to contain personally identifiable information. Since it contains demographic, census, and crime data, it is unlikely to contain offensive content.

**CIFAR10 details.**   For the tasks `car vs. cat`, `car vs. dog`, `truck vs. cat`, and `truck vs. dog` we used $N^q = 100$ data points. For the tasks `car vs. deer`, `car vs. horse`, `truck vs. deer`, and `truck vs. horse` we used $N^q = 1000$ data points.

We generated the pre-trained neural network embeddings using a variational auto-encoder (VAE) [36]. We adapted our VAE implementation from `ALIBI DETECT` [60], here. See also `notebooks/2021_05_12_CIFAR10_VAE_embeddings.ipynb` for details.

CIFAR10 is composed from a subset of the 80 million tiny images dataset. As is currently acknowledged on the 80 million tiny images website, this larger dataset is known to contain offensive images and images obtained without consent (`https://groups.csail.mit.edu/vision/TinyImages/`). However, given the benign nature of the 10 image classes in CIFAR10, we expect it does not contain offensive or personally identifiable content. These data were also obtained by web-crawling, so it is unlikely that consent for inclusion of the content into this dataset was obtained.

### E.5 Software Licenses

We here report the software used to generate our results and their associated licenses.

All of our experiments were implemented in `python`, which is licensed under the PSF license. For ease of reproducibility, ran our experiments and generated our plots IPython in Jupyter notebooks; this software is covered by a modified BSD license.

For our application to transfer learning using CIFAR10, we used a variational auto-encoder implementation adapted from `ALIBI DETECT` [60], which uses the Apache licence. Our implementation of our EM algorithm uses `TensorFlow` [1], which is licensed under the MIT license.

We made frequent use of python packages `numpy` and `scipy` and `matplotlib`. These are large libraries with components covered different licenses. See github.com/scipy/scipy/blob/master/LICENSES_bundled.txt for `scipy`, github.com/numpy/numpy/blob/main/LICENSES_bundled.txt for `numpy`, and github.com/matplotlib/matplotlib/tree/master/LICENSE for `matplotlib`.

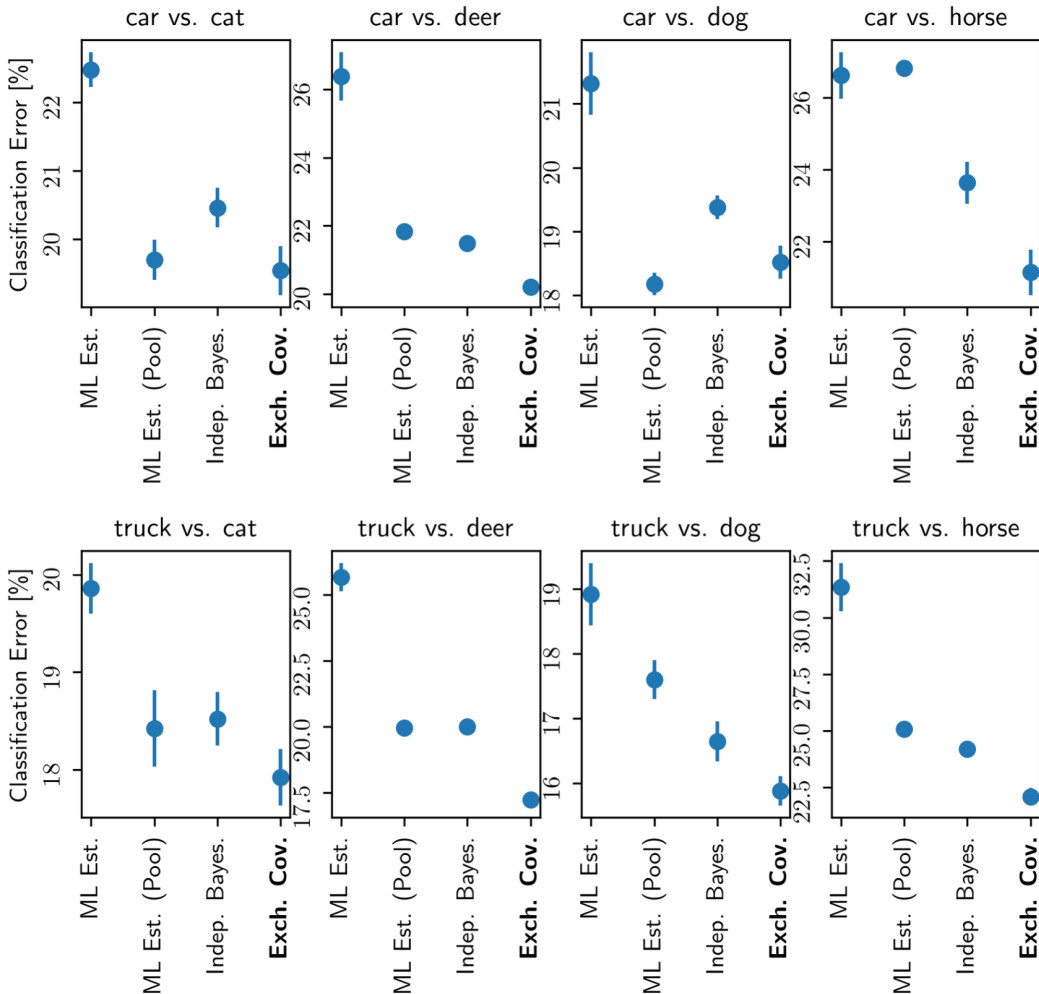

Figure 6: Performances of methods on CIFAR10 segmented by binary classification task. Uncertainty intervals are ±1SEM.