# OpenReview forum: "For high-dimensional hierarchical models, consider exchangeability of effects across covariates instead of across datasets"
_NeurIPS.cc/2021/Conference — NeurIPS 2021 Poster_

### Official Review · Reviewer_g5kM · 2021-07-16

**Rating:** 7
**Confidence:** 3

**Summary:**

The authors of this paper study exchangeable prior distribution for regression coefficients in high dimensional regression problems, where the number of covariates is typically much larger than the number of groups.  They argue that when the number of covariates are large, it is preferable to model the covariates as exchangeable rather than the datasets. The authors provide theoretical results by comparing frequentist risk of different empirical Bayes estimators and demonstrate the value of their approach empirically.

**Limitations And Societal Impact:**

The authors largely address limitations, though I'd like to see more discussion about possible extensnions as mentioned in my review:
1) structured (low rank) covariance matrices
2) full Bayesian inference (e.g. a hyperprior on Sigma)
3) more transparent simulations

**Main Review:**

Overall, I think this is an interesting, well written paper.  That said, I’m not really familiar enough with this literature to know for sure how impactful and insightful their results really are.  My gut instinct is that, while interesting, the results are not particular profound, if only because they are quite intuitive: in high dimension, low sample size settings covariance estimation is known to be quite hard, so using an empirical Bayes procedure to infer the covariance of Beta (across covariates) might be quite bad.

Setting that aside, I have another main “complaint” which doesn’t really affect my overall assessment of the quality of the paper: I find it strange that this paper was submitted to neurips.  In my opinion, this paper is much more appropriate for a statistics journal.  I say this for two reasons: 1) As somebody who rarely cares about this distinction, I still found myself thinking that this paper is very clearly a statistics paper dealing with canonical statistical issues around normal means estimation, James Stein, regression etc.  2) this is a 9 page paper with a 40 page supplement— much of the supplement is explanatory text (not just proofs and additional figures).  It seems like this might be better in a longer journal paper.  I would encourage the authors to think about whether they should consider a more appropriate venue.

Regarding the comment about statistical power at the top of page 3, is this the main driver for improvements in the ECov over EData? Since Sigma is shared across all D covariates, as D->infinity you can consistently estimate Sigma, whereas with EData D > Q implies a very poor empirical Bayes estimator of Gamma.

Relatedly, I’m curious about an estimator in which you learn an approximately low rank covariance matrix under EData, e.g. with Gamma = AA^T + sigma^2I which be a better competitor? For example, perhaps A is an D x k matrix k <= Q, so that the number of parameters in the covariance matrices to learn in EData and much smaller than the general exchangneable model. I think there is something to be learned here from the high dimensional covariance estimation literature.  In short, I think to make this really useful I’d like to know if I wouldn’t be better off just modeling exchangeability among datasets under a structured prior for the covariance (e.g low rank or sparse).  That may solve some of the problems encountered in the large D, small Q case.

Section 6.1.  It seems like a truly fair comparison would include data generated under the EData assumption, e.g. correlation between covariates within datasets, Beta_q ~ N(xi, Gamma). I’d be curious how ECov performs when D < Q and when Q > D for this assumption.  ECov may still be better when D >> Q even if the “right” prior is exchangeability across covariates just do to statistical power issues.

Minor comments:
- Notation: it would be much more intuitive if there were D datasets and Q covariates (D for dataset), rather than the reverse.  This confused me many times.
- Relatedly, I think of hierarchical modeling as models for data which can be naturally categorized into groups rather than separate datasets.  In my opinion, it would probably be much more natural to refer to Q groups rather than Q datasets.
- Is Beta_ID just iid priors on all coefficients for all datasets across all covariates?
- I’m curious about doubly exchangeable prior distributions? E.g. exchangeable across datasets and covariates.  Have you considered that at all?
- Have you looked at full Bayes instead of empirical Bayes? May not be viable in very high dimensions but I’d be curious about the performance here.  The algorithms should be much simpler and likely could be implemented entirely in something like Stan.
- The lemma about shrinkage along singular vector directions is interesting.  One comment here is that the eigenvalues of BetaBeta’ and Beta’Beta are connected via the singular vectors of Beta which relates the two approaches.  This might provide another interesting direction for thinking about this problem.  There’s also question about whether both data and covariates are correlated.  Efron comments about this in a paper about detecting row and column correlations in microarray data.  See also: “Inference with transposable data: modelling the effects of row and column correlations” (Allen and Tibshirani, 2012).


**Time Spent Reviewing:**

5

---

> ### Author Response · Authors · 2021-08-09
> **Reply to Reviewer**
>
> We thank the reviewer for their thoughtful and insightful comments, and are glad that they found the paper to be interesting and well written. Please see our top-level remark to all reviewers as well as our individual responses below.
>
> **On impact, profundity, and intuitiveness:** The reviewer is concerned about impact, insight, and profundity of our results “if only because they are quite intuitive.” A fascinating phenomenon that a number of writers have observed is that many useful results may appear obvious in retrospect [1,2,3]; one author even suggests profundity is exactly the quality of being obvious in retrospect [4]. Indeed, in computer science we suspect from "P vs NP" that not every problem with a quickly-verified solution could have been quickly solved. So we would argue that the bar for a useful paper should not be whether or not it is obvious in retrospect, but instead whether the paper is useful and novel.
>
> Arguments for usefulness and novelty appear in our opening comment to all reviewers, but we elaborate further here. **Usefulness:** high-dimensional data sets are increasingly common and learning effects in these scenarios is of fundamental importance in applications. We agree with the reviewer that our new paradigm and method are intuitive, and we see this as a positive sign for wide adoption and substantial practical impact. **Novelty:** The problem of how to share signal across multiple datasets in the high-dimensional setting is an unsolved question. Again, the standard package lme4 simply fails when there are more dimensions than datasets. We introduce a new paradigm: to share statistical strength across groups in the high-dimensional setting, assume exchangeability across covariates instead of across data sets, and improve inference by modeling correlations in effects across datasets. The only previous work we know of that falls within this paradigm are (a) a model for genetic data that can be seen as a special case of this framework, and (b) theoretical work from the 1980s on a different special case. In our manuscript, we introduce exchangeability across covariates as a general paradigm; we present an algorithm that enables tractable computation; and we demonstrate the advantages of our new approach with theoretical results, simulations, and analysis of several real data sets. Our work sets the stage for widespread adoption across many fields, while establishing a foundation for further exploration and development of both theory and empirics.
>
> **Driver for improvements in ECov over EData.** We agree with the reviewer that having insufficient statistical power to estimate a high dimensional covariance matrix is a primary mechanism by which we understand the failure of EData in high dimensions and will highlight this important point in our revised introduction.
>
> **On instability of covariance estimates, and regularized/Bayesian alternatives.**
> We appreciate the reviewer’s suggestion to consider regularized (e.g. sparse or low-rank) covariance estimates. Indeed, even when Q>D, the instability of (restricted) maximum likelihood estimates for EData as implemented in lme4 leads to well known pathologies [5-6]. We have experimented with regularization via MAP estimates (for both ECov and EData) under Wishart priors. However, we observed similar performance to the type II MLE approach we describe in our submission, and so presented only maximum likelihood for simplicity and for consistency with lme4.
>
> We agree as well that fully Bayesian inference is an exciting direction worth pursuing. While we expect Stan will work well for low-dimensional problems, it is insufficient for problems as high dimensional as those we consider here; the covariance matrices in the EData model in our real data applications have between 6,783 and 38,781 degrees of freedom. Addressing the computational burden of Bayesian inference over covariance matrices of this size is beyond the scope of the paper.
>
> **Existing work in high dimensional covariance estimation:** We agree that there is much to gain from this literature, and will note this future direction in our discussion. For simplicity, we narrowed focus to two simple covariance estimates (moment matching and maximum likelihood), but we have been looking into other techniques as well (for example, we have found [7] to be particularly relevant). We are excited to better understand how to take advantage of this literature in future work.
>
> **On relevance to the ML / NeurIPS community:** We thank the authors for recognizing the depth of our paper, and for the suggestion to consider statistics journals instead. However, we do believe our work is (also) relevant to the machine learning community in general, and to the NeurIPS community in particular. The level of engagement in our reviews (yours and the others) is already strong evidence of community interest. Two out of three of the papers reviewer rxbW brings up in relation to our paper are from NeurIPS (and the other is a tech report). And NeurIPS features nearly annual workshops in Bayesian and Bayesian nonparametric methods, separate from an annual Bayesian deep learning workshop. Even beyond Bayesian effect estimation in social science & the sciences & engineering, we believe our work has applications such as transfer learning with neural network embeddings.
>
> **A more fair comparison to EData:** We appreciate this great suggestion and will be sure to implement it. We detail a concrete plan, and its motivation, next. We first observe that our current simulations explore how relative performances change with increasing covariate dimension D. We cannot trivially increase D for EData since EData requires a sequence of covariance matrices of increasing dimension; it is not obvious to us how to choose a sensible sequence or how to interpret results in such a case. Instead, to address the reviewer’s concern in our revision, we will add a simulation plot analogous to Figure 1 but with increasing Q for a fixed D.
>
> We welcome further suggestions on possible simulation setups.
>
> **Notation:** We thank the reviewer for their suggestion on notation and will use Q "groups" rather than "datasets" throughout.
>
> **Our approach applied to each dataset independently:** Yes, this is exactly what betaID is, and we will clarify this in our revision. See our reply to reviewer ZNBH for additional details.
>
> **Doubly exchangeable prior distributions:** We think this idea is interesting precisely because it illustrates some further intuition regarding the ECov and EData paradigms. In particular, note that the dimension for which we posit exchangeability is the dimension where we are positing ignorance about the relation among units (either datasets/groups or covariates); the other (non-exchangeable) dimension is the one where we enable learning of similarities between units (via covariance structure). By modeling this covariance structure, we can share strength across units. As such, when we use exchangeability across both datasets and covariates simultaneously, de Finetti representation theorems do not provide a route to sharing strength across either dimension. An alternative approach is to instead model covariances across both datasets/groups and covariates simultaneously. Such approaches do not leverage exchangeability, but have previously been studied; for details, see our discussions in Appendix A, around lines 605 and 633.
>
> **Shrinkage along singular vectors.** We are glad to hear that the reviewer found our lemma interesting. Efron’s work in this area is quite nice. We thank the reviewer for the additional reference to Allen and Tibshirani (2012). We recommend Tsukama's fascinating and technically impressive work [8] fleshing out related connections to shrinkage estimators for matrix-valued parameters that may be understood as operating on the SVD, if the reviewer is interested in reading further into this area.
>
> References
> [1] https://www.psychologytoday.com/us/blog/close-encounters/201604/social-psychology-it-s-obvious-or-it-s-false
> [2] https://link.springer.com/article/10.1245/s10434-021-09849-4
> [3] https://mathoverflow.net/questions/338148/on-math-looking-obvious-in-retrospect
> [4] https://www.robinstewart.com/blog/2010/03/obvious-only-in-retrospect/
> [5] Eager, Christopher, and Joseph Roy. "Mixed effects models are sometimes terrible." arXiv preprint arXiv:1701.04858 (2017).
> [6] Chung, Yeojin, et al. "Avoiding boundary estimates in linear mixed models through weakly informative priors." (2012).
> [7] Tsukama and Kubokawa (2020) Shrinkage Estimation for Mean and Covariance Matrices
> [8] Tsukuma. Admissibility and minimaxity of Bayes estimators for a normal mean matrix. Journal of Multivariate Analysis, 99(10):2251–2264, 2008

---

### Official Review · Reviewer_rxbW · 2021-07-17

**Rating:** 6
**Confidence:** 3

**Summary:**

This paper studies a setting in which we analyze $Q$ datasets with $D$ covariates. The design matrix for the $q$-th dataset is denoted by $X^q$, a $N_q \times D$-dimensional matrix and the response is a $N_q$ dimensional vector denoted by $Y^q$. The question this paper addresses is the following. How can one share information across datasets and covariates to improve the predictive performance for each individual regression? The paper focuses mostly on the case of linear as well as logistic regression.

The motivation for the paper is that most existing approaches to the above problem posit exchangeability across datasets, for example, that the coefficient vector $\\beta^q \\in \\mathbb R^D$ in the $q$-th dataset, is drawn from the same prior as $\\beta^{q'}$ for another dataset $q'$. In contrast, the main suggestion in this paper is to assume exchangeability across covariates and that $(\\beta^{1}_j, ..., \\beta^{Q}_j) \\in \\mathbb R^Q$ (the vector that collects the coefficient of covariate $j$ across all datasets) is drawn from the same prior distribution for all covariates $j=1,...,D$. It is claimed, that the second approach may be preferable when the number of covariates is larger than the number of datasets.

To implement the abstract proposal above, a multivariate normal prior is assumed for  $(\\beta^{1}_j, ..., \\beta^{Q}_j)$ and an empirical Bayes approach is developed and the prior covariance matrix is estimated by marginal maximum likelihood through the EM algorithm.

There are real data analyses that demonstrate that the approach performs well. There is also theoretical justification. In one part of the theory, James-Stein-type dominance results are established (in a frequentist setting where the true coefficients are deterministic), and in the second part, some results are established in a Bayesian setting (where a prior exists and is well-specified) with asymptotics as $D \to \infty$.

**Limitations And Societal Impact:**

Yes

**Main Review:**


The paper is well-written. I liked the James-Stein type results and the applications of the new method. As the authors argue, their approach is particularly well-suited to situations where it is important to develop interpretable models. Furthermore, the new approach is effective at sharing information. Below are my main comments on the manuscript.


1) I do not mind the assumption of orthonormal sample covariance in Section 4, because the results derived are strong (James-Stein-type results). However, for Section 5, where an asymptotic approach is pursued, I feel the assumption is too strong. Instead, a Random Matrix theoretic approach (taking the covariates as being random, with suitable rescaling, and perhaps also letting $N_q$ grow as $D \to \infty$) it should be possible to provide sharp results without the very stringent Condition 4.2.


2) The main motivation for the new approach does not seem very to be sufficiently justified. In particular, I am not convinced by the introductory message that everyone in machine learning (in contrast to genetics) is following the Lindley-Smith approach of exchangeability across datasets. For example, I think it would be important that the authors provide references to previous NeurIPS papers (or machine learning papers more generally) that pursue the approach against which they argue. In fact, the basic idea of exchangeability across covariates, appears to have been embraced by the machine learning community previously. For example, a common approach for multitask learning is to seek the same sparsity pattern across datasets (for example, to use a group-lasso penalty on the coefficients of each covariate across all datasets, enforcing that some covariates will be zeroed out across all datasets). Such approaches also have a formal Bayes justification (as MAP estimators with priors that posit exchangeability across covariates). Some example references are the following:

Obozinski, Guillaume, Ben Taskar, and Michael Jordan. "Multi-task feature selection." Statistics Department, UC Berkeley, Tech. Rep 2.2.2 (2006): 2.

Yang, X., Kim, S. and Xing, E., 2009. Heterogeneous multitask learning with joint sparsity constraints. Advances in neural information processing systems, 22, pp.2151-2159.

Lee, S., Zhu, J. and Xing, E., 2010. Adaptive Multi-Task Lasso: with Application to eQTL Detection. Advances in Neural Information Processing Systems, 23, pp.1306-1314.'



3) The claim "it is used by default in the mixed modeling package lme4 [5], which has over 13 million downloads at the time of writing" needs some clarification. What is meant by "default" here?

4) I would be interested in seeing the covariance matrices estimated in the real data examples.

5)   Supplement D4, line 1113, there seems to be a typo in the proof (first equality).

6) This is very minor, but I found the notation $Q$ for number of datasets and $D$ for number of covariates a bit confusing and I kept needing to remind myself that $D$ is not the number of datasets.

*Update*: I would like to thank the authors for a very well-thought and detailed response. Correspondingly I have raised my score.


**Time Spent Reviewing:**

4

---

> ### Author Response · Authors · 2021-08-09
> **Reply to Reviewer**
>
> We thank the reviewer for their thoughtful and detailed comments. We are glad they found the paper well-written, and that they appreciated our results. We ask, however, that they reconsider their assessment of the novelty of our approach; please see our overall comment to reviewers at the top level and more detail below.
>
> **Novelty:** We respectfully disagree with the reviewer's claim that the idea of exchangeability across covariates has been "embraced by the ML community previously," and stand strongly by our assertion that our work is the first to leverage exchangeability to nontrivially share strength across datasets. Nevertheless, we appreciate the reviewer’s suggestion to relate our contribution to this prior work and will do so in our revision.
>
> In greater detail, we agree that a common approach is to use shared sparsity patterns across datasets (see e.g. our discussion and references on line 699 in appendix A4) but do not believe these approaches connect directly to exchangeability across covariates. For example, the prior to which the shrinkage estimate in Lee et al. (2010) corresponds is not exchangeable across covariates, but is actually exchangeable across datasets/groups, because the scaling parameters ($\delta_j$ and $\rho_j$ in equations 2, 3 and 4) are defined for each covariate. Accordingly, the associated priors place a different variance on the effects for each covariate and so violate ECov. By contrast, for any setting of beta, the regularization penalty is invariant to permutation of the K groups/datasets, and so reflects a priori exchangeability across datasets (i.e. EData).
>
> Moreover, none of the cited papers mention exchangeability explicitly, or discuss the idea of sharing strength and improving inferences by modeling correlation of effects across datasets as we have proposed (e.g. in our abstract on line 9). Our revised introduction will clarify these connections.
>
> **Random design asymptotics as a replacement for Condition 4.1:** We appreciate the reviewer’s suggestion to consider this asymptotic regime, and have already made significant progress towards formalizing it. Indeed, we agree that condition 4.1 is very stringent; we had loosely justified it to ourselves under the assumption that one is often interested in situations where each $N^q$ is much larger than D, and so each $X^{q T} X^q$ may be well conditioned. Joint asymptotics in N and D is a promising way to formalize this perspective! We agree that sharp results with this approach are possible; our existing lemmas already overcome the technical hurdle of proving almost sure convergence of the sequence of covariance estimates without appealing to a strong law of large numbers. **Our revision will take the suggested approach to show our theorem holds under the weaker condition that the sequence of designs satisfies an analogous condition asymptotically, and will add a proposition noting (via random matrix theory) when this weaker condition is satisfied.**
>
> **Lme4 & groups vs datasets:** We will clarify the relation to lme4 in our revision. In brief, what we call "datasets" lme4 calls "groups." lme4 models correlations across random effects within groups but not across groups [see equations 2 and 3 of the lme4 paper, Bates et al. (2015)]. By "default" we mean that when one specifies multiple (i.e. >1D) random effects in multiple (i.e. Q>1) groups, lme4 implements this correlation structure within groups. While our intent in our initial submission was to simplify the exposition by describing only the case of disjoint datasets (as in the 1972 Lindley and Smith paper) and avoiding the terminology of "groups" (since lme4 allows specification of multiple overlapping “groups”), based on these comments (and those of reviewer g5kM) we realize that this decision causes confusion instead. Our revision will address this by replacing “dataset” with "group" throughout, and will rename “EData” to “EGroup”.
>
> **Choice of notation:** We appreciate the reviewer’s comment on notation. We use Q to agree with prior work [e.g. Brown and Zidek (1980) and Haitovsky (1987)]. We believe our change from “dataset” to "group" throughout (see above) will remedy the confusion.
>
> Reporting estimated covariances: We like this suggestion and will add the estimated covariances as supplementary figures in our resubmission for the interested reader.
>
> **Typo:** Great catch! Indeed, we missed a factor of 2 in front of $<l, \lambda>$. We sincerely appreciate the reviewer's close reading and have corrected this error.

---

> > ### Comment · Reviewer_rxbW · 2021-08-25
> > **Clarification on asymptotics**
> >
> > I would like to thank the authors for their very thoughtful response. I agree that the wording in my review ("embraced by the ML community") was too strong. I am glad that the authors plan to include more connections to existing related ideas in the ML community (e.g. relating to shared sparsity across datasets) and I also appreciate the clarification regarding lme4 and groups/datasets.
> >
> > I have one question regarding the random design asymptotics. What would be the formal statement of the result in the revised manuscript?

---

> > > ### Author Response · Authors · 2021-08-25
> > > **Formal statement for joint asymptotics**
> > >
> > > Thank you for your reply.
> > >
> > > In order to formalize this random matrix approach we consider joint asymptotics in both covariate dimension and sample size.
> > > This requires some additional notation, which we introduce before providing the formal statement.
> > > As we described earlier, for clarity we now refer to the hierarchical regression problem as over $G$ "groups", rather than $Q$ "datasets".
> > > $G$ remains fixed.
> > >
> > >
> > > **Notation:** Consider a sequence of multiple regression datasets (each divided into $G$ groups) with an increasing number of covariates $D, (\mathcal{D}_1, \mathcal{D}_2, \dots ).$
> > >
> > > For each covariate dimensionality $D\in\mathbb{N}$ we write $\mathcal{D}_D= [(X_D^1, Y_D^1), (X_D^2, Y_D^2), \dots, (X_D^G, Y_D^G)  ],$
> > >
> > > where for each group $g,$ we have $X_{D}^g\in \mathbb{R}^{{N_D^g},D}$ and $Y_D^g\in \mathbb{R}^{N_D^g}$
> > > where each $N_D^g$ is the number of datapoints in group $g$ within dataset $D.$
> > > We take $\sigma_g^2$ to be the noise variance of observations in group $g$.
> > >
> > > As described in our initial response, our approach will be to replace Condition 4.1 with the following new and less stringent condition.
> > >
> > > **Condition 5.1:** For each group $g, \|\sigma_g^{-2} X_D^{g\top} X_D^g  - \sigma^{-2} I_D\|_2 \rightarrow 0$ for some shared variance $\sigma^2$ as $D\rightarrow \infty.$
> > >
> > >
> > > We then update the theory in Section 5 by replacing "Condition $4.1$" with "Condition $5.1$" in Definition 5.1,
> > > but leave the subsequent theorem and lemma statements unchanged (besides the change of terminology to "groups").
> > > As mentioned in our initial response, our current proofs go through without substantive changes.
> > >
> > > Finally, we include the following proposition.
> > >
> > > **Proposition:** For each group $g,$ assume $N_D^g$ is $\omega(D)$ (i.e. the number of datapoints in each group grows super-linearly in the covariate dimension).
> > > Additionally assume that for each group $g$ and dataset $D$
> > > the scaled entries of the design matrix $X_D^g$ are i.i.d. with mean $0.$ and variance $\sigma_g^2/(\sigma^2 N_D^g).$
> > > Lastly, assume there exists a constant $c,$ such that the entries of each design matrix, once scaled as
> > > $\sqrt{N_D^g} X_{D}^g,$ are $c$-sub-Gaussian.
> > > Then Condition 5.1 is satisfied with probability 1.
> > >
> > >
> > > One approach to proving the proposition is to use a non-asymptotic concentration inequality on the eigenvalues of
> > > each $X_D^{g\top} X_D^g$ (e.g. Wainwright, 2019; Corollary 6.20).
> > > The significance of this proposition is that it demonstrates that our results in Section 5 can hold without the stringent Condition 4.1.
> > >
> > > We welcome any additional suggestions in this direction, or clarifications of the approach the reviewer had in mind in their initial review.
> > >
> > > Reference:
> > > Wainwright, Martin J. "High-dimensional statistics: A non-asymptotic viewpoint." Vol. 48. Cambridge University Press, 2019.

---

### Official Review · Reviewer_ZNBH · 2021-07-19

**Rating:** 5
**Confidence:** 2

**Summary:**

This paper argues that when the number of covariates exceeds the number of datasets, it is more natural to model effects as exchangeable across covariates than across datasets. The paper illustrates the setting using a microcredit example.   It presents a hierarchical model to formalize the perspective and derive an efficient algorithm. The paper evaluates the model using three real-world datasets.

**Limitations And Societal Impact:**

see above

**Main Review:**

pros
- the idea is interesting: why shouldn't we treat the effect as exchangeable among covariates?
- The paper establishes that theoretically, Ecov may have better frequentist risk than Edata. Further, it uses Bayes risk to show that Ecov may improve the estimation quality in the high dimensional settings.
- The proposed method, Ecov, is evaluated on a range of different real-world applications from image classification to social science study.

My main concerns are about (1) the motivation of the method and (2) the writing.

(1) The core of the paper is arguing that one should treat the effect to be exchangeable among covariates.
In various parts of the paper, e.g., 7, 42, 100, the authors claim that Ecov is more “natural”. However, it’s not clear what is meant here. In line 78, the authors explain the intuition using a micro-credit dataset. However, in the experiment, there was no intuition or justification for the method. A priori, shouldn’t it be the case that what is exchangeable is dependent on the question at hand? This issue was only raised briefly in the discussion.  It should be discussed in detail in the introduction and throughout the paper.

(2) The overall writing is not clear. Notably, section 5 is supposed to justify how Ecov leverages similarities across datasets in a meaningful way. However, the writing is not precise and is difficult to follow. For example, in line 222, it is unclear what “our method applied to each dataset independently” means.

Other comments
- Please use a reference package.

**Time Spent Reviewing:**

5

---

> ### Author Response · Authors · 2021-08-09
> **Reply to Reviewer**
>
> We thank the reviewer for their thoughtful review and for raising valid concerns. We expect that these concerns will be decisively resolved in our revision, and urge the reviewer to reconsider their score with only the concerns they expect to go unresolved in mind. Please see our comment to all reviewers at the top level; we respond to individual concerns below.
>
> **The motivation of ECov as “natural”:** We agree with the reviewer that our use of the word “natural” was too vague. As we discuss next, we had meant to say that we see ECov as often more consistent with prior beliefs than EData in high dimensions; we will be sure to eliminate the word “natural” from the paper and elaborate on our precise meaning.
>
> In more detail, we view assumptions of exchangeability as assumptions of ignorance. Following from Lindley and Smith (1972), we believe that exchangeability and de Finetti representation theorems can be effectively used precisely when there is no good answer to the rhetorical question you posed; i.e. “ why shouldn't we [...]?”. One’s prior beliefs are exchangeable when they have no prior information to contradict this assumption. Our contention is that it is typically easier to reason about non-exchangeable prior beliefs about small numbers of things. For example, consider our law enforcement analysis, using data from four regions (NorthEast, South, Midwest, and West) and >100 demographic characteristics of communities in those regions. When we have many covariates (as in this case), we find it harder to imagine having or incorporating non-exchangeable information about this collection. By contrast, reasoning about which regions might be more similar to one another seems more reasonable to us a priori. As another example, in the application to CIFAR10, the covariates are dimensions of the latent space of a variational auto-encoder, so we have no information to distinguish the dimensions; any set of parameter values would seem equally reasonable upon a relabeling of the covariate dimensions.
>
> **The meaning of "our method applied to each dataset independently”:** We apologize for accidentally deleting a more precise explanation in our submission and will be sure to include it in a revision. In brief, the ECov approach is formulated for a general number of datasets (Q), including the special case of a single dataset (Q=1). This baseline approach is defined by considering this special case; our estimate for each $\beta^q$ is what we would obtain from ECov when it is provided with the $q^{th}$ dataset alone.
>
> **Reference package:** Would the reviewer clarify what they mean by “Please use a reference package”? We haved used natbib, to which we passed the options ‘{numbers,compress}’. We welcome suggestions for alternative tools or styles for references and bibliography management.

---

> > ### Comment · Reviewer_ZNBH · 2021-08-22
> > **response**
> >
> > Hi, thank you for the response. After reading the other reviews and responses, I realize that I did not fully understand the paper. I have updated my score and confidence level to reflect that.
> >
> > However, I agree with Review rxbW that the motivation is not sufficiently justified. Given that the NEURIPS audience is mostly ML researchers, there should be a more thorough connection to the existing ML literature.
> >
> > I recommend using the hyperref package. The cross-referenced elements can become hyperlinked.

---

> > > ### Author Response · Authors · 2021-08-25
> > > **reply to response**
> > >
> > > Thank you for reconsidering your initial assessment.
> > >
> > > We hope you will confer on the extent of justification with the other reviewers before finalizing your decision to mark us down on this point.
> > >  We expect that our response to reviewer rxbW (and the discussion we describe adding) will largely resolve their concerns (see our response to rxbW under "Novelty").
> > >
> > > We believe our approach is a simple solution to the problem we address.  Reviewer g5km, for example, describes our approach as "intuitive" (even to a fault).
> > >
> > > We used hyperref; hyperlinks may have broken when splitting main and supplementary files.  Our revision will contain working hyperlinks.

---

### Official Review · Reviewer_RrAL · 2021-07-27

**Rating:** 6
**Confidence:** 4

**Summary:**

Paper contains a simple and effective result on analyzing datasets shared across a domain or in other words multiple sources with promising theoretical results supported by some experiments.

**Ethical Concerns:**

None.

**Limitations And Societal Impact:**

Addressed appropriately.

**Main Review:**

Strength: + The main contribution of the paper is an Alternating minimization procedure. To describe it formally, in each step a covariance type matrix is estimated for each dimension of the covariate based on regressors for that dimension -- importantly, these are estimated with data from all the sources. The paper proposes to use Conjugate Gradient (CG) algorithm algorithm to solve the inner least squares (or logistic) regression problems.
+ Explains how block diagonal structure can be exploited when implementing CG algorithm. Experimental results with three different datasets (in addition to simulations) compared with show that the proposed method

Weakness: - The theoretical justification presented in Sections 4 and 5 for the proposed algorithm seems correct, although it is a bit out of place. At a high level, Section 4 in the paper argues by associativity that the proposed estimator using ECov dominates the naive estimator EData by exploiting the closed form solution available for Least Squares problem. However, it is not clear whether the results presented can be adapted for classification setups. In Section 5, near Line 226 the paper mentions that risk depending on non-central inverse Wishart matrix is challenging, and that is the main motivation to use average case risk as done in Bayesian analysis. However, the results presented in Lemma and Theorem both use eigenvalues in the Gain. It would have been nice to read why we do not face the challenge anymore. The paper presents some simulations to support the theory presented but it falls on the short side -- for example, it is not clear whether any of the benefits of Theorem 5.3 can be realized in practice since only simulation results are shown.
- Somewhat related to the previous point, the paper has no discussion on PAC-Bayes methods which are also a standard way to perform Bayesian generalization analysis.
- Only preliminary experiments are presented paper which is surprising given that CG algorithm is indeed scalable, especially in the sparse settings considered here.

**Time Spent Reviewing:**

5

---

> ### Author Response · Authors · 2021-08-09
> **Reply to Review**
>
> We are excited to hear the reviewer found our new method simple and effective, and we thank the reviewer for their thoughtful comments. Please see our comment to all reviewers at the top level; we respond to individual concerns below.
>
> **Extension to classification problems:** We agree that our current theory applies to regression but not to classification, which is certainly an important use case. We think that the extension of our theory to classification should be straightforward using standard techniques -- such as linearization via a first-order Taylor expansion. But actually working these details out in full would require substantial additional space, and another reviewer has already balked at the length of our appendix. We hope to balance showing the promise of our approach with length considerations by focusing on experimental evidence. Namely, our existing classification experiments on real data (Figure 2) show substantial accuracy improvement in our method over baselines. We have also already performed simulations for classification though we did not add these to our original submission; we will include them in a revision.
>
> **Overcoming intractability of non-central Wisharts:** We appreciate the reviewer’s interest in gaining intuition for how Section 5 overcomes this intractability. We will be sure to clarify this point in the main text in a revision. In brief (cf. line 228), we regain tractability by appealing to asymptotics (in addition to average case analysis). Specifically, lemma D3 (p. 31 of the supplement) uses what in effect amounts to a strong law of large numbers to show that once scaled, the problematic Wishart matrix converges strongly to its expectation. The eigenvalues of this expectation appear in our expression for the asymptotic risk -- and in the gain of joint modeling.
>
> **Evidence of practical benefit:** The reviewer writes that “it is not clear whether any of the benefits of Theorem 5.3 can be realized in practice since only simulation results are shown.” We weren’t sure how to interpret this comment, so we propose some interpretations next; if we were off the mark, we would appreciate any clarification. If the reviewer is concerned that we do not analyze real data at all, we respectfully point the reviewer to our extensive evaluation of different methods on the following real datasets: “Blog Popularity”, “Law Enforcement”, and “CIFAR10”. Figure 2 showcases the benefits of our approach in real data. If the reviewer is concerned that we focus on prediction rather than estimation in our real data analysis, we note that our choice is one of necessity; while our paper is concerned with estimation performance, there is no ground truth for estimation available in real data -- so we turn to prediction as a convenient proxy (see line 329). If the reviewer meant to suggest we should have examined the eigenvalues in our real data sets and discussed how they relate to performance, we note that our theory refers to population eigenvalues, which we do not have access to in real data. But we are happy to display the learned covariance matrices (a point we discuss in more detail below) to aid in intuition.
>
> **Connections to PAC-Bayes:** The reviewer observes that we have no discussion of PAC-Bayesian methods. However, as the reviewer notes, PAC-Bayesian methods are used to provide bounds on generalization error, which describes prediction performance on test data. While generalization is an important area of research, it is distinct from our present objective, which is accurate estimation of effects without regard to any test set or predictive objective. Recall that, in our real-data analyses, we consider prediction error on held-out data only as a (standard) proxy because true parameters are not observed.
>
> **“Preliminary” experiments:** The reviewer describes our experiments as “preliminary,” but we respectfully disagree with this characterization. Our present work is concerned with any case where the number of parameters/covariates exceeds the number of datasets. Within those cases, we see three regimes of potential examples: (i) hundreds to low thousands of parameters, (ii) thousands to tens of thousands of parameters, and (iii) tens of thousands of parameters and beyond. We found it straightforward to locate real datasets of interest in set (i), and indeed we believe a great many problems have this form, are of practical importance, and are still overlooked by existing software (e.g., lme4 does not even run). We could use genomics datasets in sets (ii) or (iii), but we wanted to demonstrate the usefulness of our method beyond genomics. We are certainly open to suggestions for other datasets where effect estimation would be of interest, though.
>
> While we dispute that our experiments are preliminary, we do agree that e.g. for genomics, it will be important to scale to (ii) or (iii). In these cases, there are additional practical considerations that we think will be interesting avenues for future work. Note that for our EM algorithm, we need both the posterior mean (which is indeed quite scalable) and the posterior variance. The latter is a bottleneck; namely, we need to invert a QD-sized matrix many times. We can run our existing methods for (ii) though slowly; for (iii) our EM algorithm becomes prohibitive, so we would need to make further algorithmic additions (which we again worry might bog down the present paper).
>
> **Sparsity:** We would appreciate any additional clarity about the reviewer’s note of computational efficiency “especially in the sparse settings considered here.” While we think it would be interesting to consider potential computational gains in the case of sparse data, our present paper does not consider sparsity in the models, simulations, or applications.

---

> > ### Comment · Reviewer_RrAL · 2021-09-01
> > **Response clarifies my concerns**
> >
> > Thanks for clarifying my concerns. I will keep the rating as is since there is some amount of rewriting to be done as indicated in your response.

---

### Author Response · Authors · 2021-08-09
**Overall Reply to All Reviewers**

We are grateful to the reviewers for their very thoughtful and thorough comments. We are excited to hear that multiple reviewers found our ideas intuitive and interesting and found the paper well-written. We are certain that our paper will be greatly improved by addressing reviewers’ remarks.

Before responding to specific points, we recap the novelty and significance of our paper. Hierarchical regression problems are ubiquitous in the sciences, engineering, and social sciences as evidenced by the (now) over 15 million downloads of the lme4 package. But lme4 fails to even run when the number of covariates exceeds the number of datasets. And in general, the problem of how to share signal across multiple datasets in the high-dimensional setting is an unsolved question of fundamental importance. Whereas lme4 uses the standard paradigm of “exchangeability across datasets,” we introduce an analogous and intuitive paradigm of “exchangeability across covariates” for high-dimensional problems. We develop and analyze a new algorithm to make this paradigm practical.

---

### Decision · Program_Chairs · 2021-09-27

**Decision:**

Accept (Poster)

**Comment:**

The reviewers were initially concerned that this paper was not well-suited for NeurIPS, but better suited to a statistics journal. However, during the discussion phase the authors were able to convince the reviewers that they were able to make the necessary changes to make this right for NeurIPS. Therefore I move to accept. The authors should make sure to make the serious changes suggested by the authors and then this paper will be a nice contribution to the conference.